# EQUIVARIANT NEURAL FUNCTIONAL NETWORKS FOR TRANSFORMERS

**Viet-Hoang Tran**[1]* **Thieu N. Vo**[1]* **An Nguyen The**[2]* **Tho Tran Huu**[1]
**Minh-Khoi Nguyen-Nhat**[2] **Thanh Tran**[3] **Duy-Tung Pham**[2] **Tan M. Nguyen**[1]
[1]National University of Singapore  [2]FPT Software AI Center, Vietnam
[3]VinUniversity, Vietnam
{hoang.tranviet, thotranhuu}@u.nus.edu, {thieuvo, tanmn}@nus.edu.sg
{annt68, khoinnm1, tungpd10}@fpt.com, 21thanh.tq@vinuni.edu.vn

## ABSTRACT

This paper systematically explores neural functional networks (NFN) for transformer architectures. NFN are specialized neural networks that treat the weights, gradients, or sparsity patterns of a deep neural network (DNN) as input data and have proven valuable for tasks like learnable optimizers, implicit data representations, and weight editing. While NFN have been extensively developed for MLP and CNN, no prior work has addressed their design for transformers, despite their importance in modern deep learning. This paper aims to address this gap by systematically studying NFN for transformers. We first determine the maximal symmetric group of the weights in a multi-head attention module and a necessary and sufficient condition under which two sets of hyperparameters of the module define the same function. We then define the weight space of transformer architectures and its associated group action, leading to design principles for NFN in transformers. Based on these, we introduce Transformer-NFN, an NFN equivariant under this group action. Additionally, we release a dataset of over 125,000 Transformers model checkpoints trained on two datasets with two tasks, providing a benchmark for evaluating Transformer-NFN and encouraging further research on transformer training and performance. The code is publicly available at https://github.com/MathematicalAI-NUS/Transformer-NFN.

## 1 INTRODUCTION

Deep neural networks (DNNs) have become highly flexible and adaptable modeling tools, widely employed in domains such as natural language processing (Rumelhart et al., 1986; Hochreiter & Schmidhuber, 1997; Vaswani et al., 2017; Devlin et al., 2019; Nielsen et al., 2025b; Gu & Dao, 2024; Vo et al., 2025), computer vision (He et al., 2015; Szegedy et al., 2015; Krizhevsky et al., 2012; Huang et al., 2020; Nguyen et al., 2023a), and natural science (Raissi et al., 2019; Jumper et al., 2021; Portwood et al., 2019). Increasing attention is being given to constructing specialized neural networks that treat the weights, gradients, or sparsity patterns of DNNs as inputs. These specialized networks are referred to as neural functional networks (NFNs) (Zhou et al., 2024b). NFNs have proven useful in various applications, such as designing learnable optimizers for neural network training (Bengio et al., 2013; Runarsson & Jonsson, 2000; Andrychowicz et al., 2016; Metz et al., 2022; Knyazev et al., 2024), extracting information from implicit representations of data (Stanley, 2007; Mildenhall et al., 2021; Runarsson & Jonsson, 2000), performing targeted weight editing (Sinitsin et al., 2020; Cao et al., 2021; Mitchell et al., 2022), evaluating policies (Harb et al., 2020), and applying Bayesian inference with neural networks (Sokota et al., 2021).

The construction of NFNs presents significant challenges due to the complexity and high dimensionality of their structures. Earlier approaches sought to address this by constraining the training process, thereby restricting the weight space (Dupont et al., 2022b; Bauer et al., 2023; Luigi et al.). More recently, research has shifted towards the development of permutation-equivariant NFNs, capable of operating on neural network weights without such constraints (Navon et al., 2023; Zhou et al., 2024b; Kofinas et al., 2024; Zhou et al., 2024c). These advancements have resulted in NFNs

---

*Equal contribution. Correspondence to: hoang.tranviet@u.nus.edu and tanmn@nus.edu.sg

that exhibit equivariance to weight permutations, which correspond to neuron rearrangements within hidden layers. Additionally, NFNs that are equivariant to both permutations and operations such as scaling or sign-flipping have been proposed. These include graph-based message-passing methods as described in (Kalogeropoulos et al., 2024) and parameter sharing techniques in (Tran et al., 2024a). However, most of the recent neural functional networks are designed for Multilayer Perceptrons (MLPs) and Convolutional Neural Networks (CNNs).

Transformers have achieved remarkable success in the field of natural language processing (NLP) (Vaswani et al., 2017; Bahdanau et al., 2015; Nguyen et al., 2023b; Nielsen et al., 2025a), powering large-scale language models. These models highlight the transformer architecture's capacity to perform a broad range of NLP-related tasks and its applicability to real-world scenarios. The influence of Transformers extends beyond NLP, finding applications in areas such as computer vision through Vision Transformers (Dosovitskiy et al., 2020), reinforcement learning (Parisotto et al., 2020; Chen et al., 2021), audio processing (Radford et al., 2023), multi-modal data integration (Radford et al., 2021), and robotics (Monastirsky et al., 2022). Despite the significance of Transformers, no systematic development of NFNs for Transformers has been realized to date. Addressing the design principle of NFNs for Transformers, as well as constructing equivariant NFNs capable of processing transformer models, has thus emerged as a critical research problem.

## 1.1 BACKGROUND

Self-attention is the core component of a transformer block (Cho et al.; Lin et al., 2017; Parikh et al., 2016; Teo & Nguyen, 2025). Self-attention learns to align tokens within an input sequence by assessing the relative importance of each token in relation to all other tokens. It then transforms each token into a weighted average of the feature representations of other tokens, with the weights determined by importance scores between token pairs. These importance scores allow each token to attend to others in the sequence, effectively capturing contextual representations (Bahdanau et al., 2015; Kim et al., 2017; Vaswani et al., 2017; Nguyen et al., 2022; Abdullaev & Nguyen, 2025).

**Self-attention.** Let $X = [X_1, \ldots, X_L]^\top \in \mathbb{R}^{L \times D}$ be an input sequence of $L$ tokens with $D$ features. Each token is a vector $X_i \in \mathbb{R}^D$. The self-attention, denoted by Head, transforms $X$ into the output sequence $\text{Head}(X) \in \mathbb{R}^{L \times D_v}$ defined as

$$\text{Head}(X; W^{(Q)}, W^{(K)}, W^{(V)}) = \text{softmax}\left(\frac{(XW^{(Q)})(XW^{(K)})^\top}{\sqrt{D_k}}\right) XW^{(V)}, \qquad (1)$$

where the parameters $W^{(Q)}, W^{(K)} \in \mathbb{R}^{D \times D_k}$ and $W^{(V)} \in \mathbb{R}^{D \times D_v}$ are called the query, key and value matrices. Here, $D_k$ and $D_v$ are given positive integers.

**Multihead attention.** To jointly attend to information from different representation subspaces at different positions, a multihead attention is used. Let $h$ be a positive integer which represents the number of head. The multihead attention transforms the input sequence $X \in \mathbb{R}^{L \times D}$ to an output sequence in $\mathbb{R}^{L \times D}$ defined by:

$$\text{MultiHead}\left(X; W^{(O)}, \{W^{(Q,i)}, W^{(K,i)}, W^{(V,i)}\}_{i=1}^h\right)$$
$$= \left(\bigoplus_{i=1}^h \text{Head}\left(X; W^{(Q,i)}, W^{(K,i)}, W^{(V,i)}\right)\right) W^{(O)}, \qquad (2)$$

where each head is a self-attention defined in equation 1, and $\bigoplus$ is the concatenation operator. Here, the matrices $W^{(Q,i)}, W^{(K,i)} \in \mathbb{R}^{D \times D_k}$, $W^{(V,i)} \in \mathbb{R}^{D \times D_v}$ and $W^{(O)} \in \mathbb{R}^{hD_v \times D}$ are learnable from input data for some postive integers $D_k$ and $D_v$.

## 1.2 CONTRIBUTION

This paper provides a systematic study on the development of a neural functional network (NFN) for transformer architectures. To achieve this, we present three essential components for the study: (1) a design principle of NFNs for Transformers that incorporates the maximal symmetric group for the multi-head attention module, (2) an equivariant NFN for Transformers, which we will call Transformer-NFN, and (3) a benchmark dataset for testing the applicability and efficiency of Transformer-NFN. In particular, our contributions are four-fold:

1. We determine the maximal symmetric group of the weights in a multi-head attention module, establishing the necessary and sufficient conditions under which two sets of hyperparameters for a multi-head attention module define the same function.

2. We formally define the weight space of a transformer architecture, along with a group action on this weight space. In conjunction with the maximal symmetric group of the weights in multi-head attention modules, we characterize the design principles for NFNs in transformer architectures.

3. We introduce Transformer-NFN, an NFN for transformer architectures that is equivariant under the specified group action. The main building block of Transformer-NFN is an equivariant polynomial layer derived from a parameter-sharing strategy.

4. We release Small Transformer Zoo dataset, which consists of more than 125,000 Transformers model checkpoints trained on two different tasks: digit image classification on MNIST and text topic classificaction on AGNews. To our knowledge, this the first dataset of its kind. This dataset serves as a benchmark for testing the applicability and efficiency of our Transformer-NFN, while also encouraging further research in this field to gain a deeper understanding of transformer network training and performance.

We empirically demonstrate that Transformer-NFN consistently outperforms other baseline models on our constructed datasets. Through comprehensive ablation studies, we emphasize Transformer-NFN's ability to effectively capture information within the transformer block, establishing it as a robust predictor of model generalization.

**Organization.** Following a review of related works in Section 2, we derive the maximal symmetric group of multihead attention in Section 3. In Section 4, we construct the weight space of a standard transformer block and define a corresponding group action. Section 5 introduces a family of equivariant NFNs for Transformers, referred to as Transformer-NFNs. We then present the setting and details of the Small Transformer Zoo dataset in Section 6. Finally, in Section 7, we evaluate the applicability and efficiency of Transformer-NFNs on this dataset.

## 2 RELATED WORK

**Symmetries of weight spaces.** The exploration of symmetries within the weight spaces of neural networks, which relates to assessing the functional equivalence of these networks, is a well-established field of study (Allen-Zhu et al., 2019; Du et al., 2019; Frankle & Carbin; Belkin et al., 2019; Novak et al., 2018). This topic was first introduced by Hecht-Nielsen in (Hecht-Nielsen, 1990). Following this, numerous studies have yielded insights for various network architectures, as detailed in (Chen et al., 1993; Fefferman & Markel, 1993; Kurkova & Kainen, 1994; Albertini & Sontag, 1993b;a; Bui Thi Mai & Lampert, 2020).

**Neural functional networks.** Recent studies have aimed to develop representations for trained classifiers to evaluate their generalization performance and understand neural network behavior (Baker et al., 2018; Eilertsen et al.; Unterthiner et al., 2020; Schürholt et al., 2021; 2022a;b). Notably, low-dimensional encodings for Implicit Neural Representations (INRs) have been created for various downstream tasks (Dupont et al., 2022a; Luigi et al.). Schürholt et al. (2021) proposed neuron permutation augmentations, and other research has focused on encoding and decoding neural network parameters for reconstruction and generative purposes (Peebles et al., 2022; Ashkenazi et al., 2023; Knyazev et al., 2021; Erkoç et al., 2023).

**Equivariant NFNs for MLPs and CNNs.** Recent advancements have made considerable strides in addressing the limitations of permutation equivariant neural networks through the introduction of permutation equivariant layers. These layers employ intricate weight-sharing patterns (Navon et al., 2023; Zhou et al., 2024b; Kofinas et al., 2024; Zhou et al., 2024c), as well as set-based (Andreis et al., 2023) and graph-based structures of the network's weights (Lim et al., 2024; Kofinas et al., 2024; Zhou et al., 2024a), to maintain equivariance. Moreover, monomial equivariant NFNs, which are equivariant to both permutations and scaling, have been proposed in (Kalogeropoulos et al., 2024) utilizing a graph-based message-passing mechanism and in (Tran et al., 2024a; Vo et al., 2024) through a parameter sharing mechanism.

## 3 MAXIMAL SYMMETRIC GROUP OF A MULTI-HEAD ATTENTION

As the first step in characterizing a principal design of NFNs for Transformers, we need to decide when two tuples of matrices with appropriate sizes, say $\left(\left\{W^{(Q,i)}, W^{(K,i)}, W^{(V,i)}\right\}_{i=1}^{h}, W^{(O)}\right)$ and $\left(\left\{\overline{W}^{(Q,i)}, \overline{W}^{(K,i)}, \overline{W}^{(V,i)}\right\}_{i=1}^{h}, \overline{W}^{(O)}\right)$, define the same Multihead map. We provide a complete answer for this step in this section, thus characterizing the maximal symmetric group of the weights of Multihead.

Let us consider the MultiHead map with $h$ heads defined in Equation (2). We can rewrite MultiHead as:

$$
\begin{aligned}
\text{MultiHead} & \left(X; W^{(O)}, \left\{W^{(Q,i)}, W^{(K,i)}, W^{(V,i)}\right\}_{i=1}^{h}\right) \\
& = \left(\bigoplus_{i=1}^{h} \text{Head}\left(X; W^{(Q,i)}, W^{(K,i)}, W^{(V,i)}\right)\right) W^{(O)} \\
& = \sum_{i=1}^{h} \text{Head}\left(X; W^{(Q,i)}, W^{(K,i)}, W^{(V,i)}\right) W^{(O,i)} \\
& = \sum_{i=1}^{h} \text{softmax}\left(X \cdot \left(\frac{W^{(Q,i)} \cdot \left(W^{(K,i)}\right)^{\top}}{\sqrt{D_k}}\right) \cdot X^{\top}\right) \cdot X \cdot \left(W^{(V,i)} \cdot W^{(O,i)}\right),
\end{aligned}
$$

where $W^{(O)} = \left(W^{(O,1)}, \ldots, W^{(O,h)}\right)$ with each $W^{(O,i)} \in \mathbb{R}^{D_v \times D}$. Here, the matrices $\left(W^{(Q)}\right) \cdot \left(W^{(K)}\right)^{\top}/\sqrt{D_k}$ and $W^{(V,i)} \cdot W^{(O,i)}$ have the same size $D \times D$. Based on this observation, we define for each positive integer $h$ and each sequence of matrices $\{A_i, B_i\}_{i=1}^{h}$ in $\mathbb{R}^{D \times D}$ a map

$$
F\left(\cdot; \{A_i, B_i\}_{i=1}^{h}\right) \ : \ \bigsqcup_{l>0} \mathbb{R}^{l \times D} \to \bigsqcup_{l>0} \mathbb{R}^{l \times D},
$$

defined by

$$
F(X; \{A_i, B_i\}_{i=1}^{h}) := \sum_{i=1}^{h} \text{softmax}\left(X \cdot A_i \cdot X^{\top}\right) \cdot X \cdot B_i,
$$

for each $X \in \mathbb{R}^{l \times D}$. It is noted that, for each integer $l > 0$, the image of $\mathbb{R}^{l \times D}$ via $F$ is contained in $\mathbb{R}^{l \times D_v}$ and $F$ can be viewed as a generalization of MultiHead. With this setting at hand, we proved that:

**Theorem 3.1** (Independence of heads in multi-head attention). *Let $D$ be a positive integer. Assume that for a positive integer $h$, matrices $A_1, A_2, \ldots, A_h \in \mathbb{R}^{D \times D}$ and $B_1, B_2, \ldots, B_h \in \mathbb{R}^{D \times D}$, we have*

$$
F\left(X; \{A_i, B_i\}_{i=1}^{h}\right) = 0, \tag{3}
$$

*for all positive integers $L$ and $X \in \mathbb{R}^{L \times D}$. Then, if $A_1, A_2, \ldots, A_h$ are pairwise distinct, then*

$$
B_1 = \ldots = B_h = 0.
$$

*Remark* 1. Roughly speaking, the above theorem says that, in the multi-head attention, each individual head plays its own unique role.

Based on the above theorem, we characterize the maximal symmetric group of the weights of MultiHead in the following theorem.

**Theorem 3.2** (Maximal symmetric group of multi-head attentions). *Let $h, D, D_k, D_v$ be positive integers. Let $\left(W^{(Q,i)}, W^{(K,i)}, W^{(V,i)}, W^{(O,i)}\right)$ and $\left(\overline{W}^{(Q,i)}, \overline{W}^{(K,i)}, \overline{W}^{(V,i)}, \overline{W}^{(O,i)}\right)$ be arbitrary elements of $\mathbb{R}^{D \times D_k} \times \mathbb{R}^{D \times D_k} \times \mathbb{R}^{D \times D_v} \times \mathbb{R}^{D_v \times D}$ with $i = 1, \ldots, h$. Assume that*

*(a)* $\max(D_k, D_v) \leqslant D$,

(b) the matrices $W^{(Q,i)} \cdot \left(W^{(K,i)}\right)^{\top}, \overline{W}^{(Q,i)} \cdot \left(\overline{W}^{(K,i)}\right)^{\top}, W^{(V,i)} \cdot W^{(O,i)}$, and $\overline{W}^{(V,i)} \cdot \overline{W}^{(O,i)}$ are of full rank,

(c) the matrices $W^{(Q,i)} \cdot \left(W^{(K,i)}\right)^{\top}$ with $i = 1, \ldots, h$ are pairwise distinct,

(d) the matrices $\overline{W}^{(Q,i)} \cdot \left(\overline{W}^{(K,i)}\right)^{\top}$ with $i = 1, \ldots, h$ are pairwise distinct.

*Then the following are equivalent:*

1. *For every positive integer $L$ and every $X \in \mathbb{R}^{L \times D}$, we always have*

$$\text{MultiHead}\left(X; \left\{W^{(Q,i)}, W^{(K,i)}, W^{(V,i)}, W^{(O,i)}\right\}_{i=1}^{h}\right)$$
$$= \text{MultiHead}\left(X; \left\{\overline{W}^{(Q,i)}, \overline{W}^{(K,i)}, \overline{W}^{(V,i)}, \overline{W}^{(O,i)}\right\}_{i=1}^{h}\right). \tag{4}$$

2. *There exist matrices $M^{(i)} \in \text{GL}_{D_k}(\mathbb{R})$ and $N^{(i)} \in \text{GL}_{D_v}(\mathbb{R})$ for each $i = 1, \ldots, h$, as well as a permutation $\tau \in \mathcal{S}_h$, such that*

$$\left(\overline{W}^{(Q,\tau(i))}, \overline{W}^{(K,\tau(i))}, \overline{W}^{(V,\tau(i))}, \overline{W}^{(O,\tau(i))}\right)$$
$$= \left(W^{(Q,i)} \cdot (M^{(i)})^{\top}, W^{(K,i)} \cdot (M^{(i)})^{-1}, W^{(V,i)} \cdot N^{(i)}, (N^{(i)})^{-1} \cdot W^{(O,i)}\right). \tag{5}$$

*Remark* 2 (Necessity of the assumptions $(a)$, $(b)$, $(c)$, and $(d)$). The four assumptions in Theorem 3.2 are necessary for the implication $(1.) \Rightarrow (2.)$. Indeed, the assumptions $(c)$ and $(d)$ are needed to utilize Theorem 3.1. In addition, it follows from Theorem 3.1 and item $(1.)$ that

$$W^{(Q,i)} \cdot (W^{(K,i)})^{\top} = \overline{W}^{(Q,\tau(i))} \cdot \left(\overline{W}^{(K,\tau(i))}\right)^{\top} = A_i, \text{ and}$$
$$W^{(V,i)} \cdot W^{(O,i)} = \overline{W}^{(V,\tau(i))} \cdot \overline{W}^{(O,\tau(i))} = B_i$$

for some matrices $A_i, B_i \in \mathbb{R}^{D \times D}$. At this point, the assumptions $(a)$ and $(b)$ are necessary to utilize the rank decomposition of $A_i$ and $B_i$ to obtain the symmetries of the factors (Piziak & Odell, 1999). It worth noting that these assumptions are generically satisfied in practical implementations since the parameters of MultiHead are randomly initialized with dimensions $D_k \leqslant D$ and $D_v \leqslant D$.

*Remark* 3 (Maximal symmetric group of multi-head attentions). Theorem 3.2 suggests that a multi-head attention in practice is characterized by the products $W^{(Q,i)} \cdot \left(W^{(K,i)}\right)^{\top}$ and $W^{(V,i)} \cdot W^{(O,i)}$ at each head. As a consequence, we can consider the group $\mathcal{S}_h \times \text{GL}_{D_k}(\mathbb{R})^h \times \text{GL}_{D_v}(\mathbb{R})^h$ to be the maximal symmetric group of the MultiHead map.

## 4 WEIGHT SPACE OF A TRANSFORMER BLOCK AND GROUP ACTION

In this section, we construct the weight space of a standard transformer block and define a group action on it. Let us first recall necessary notations from permutation matrices before defining the weight space in Section 4.2 and the group action in Section 4.3.

### 4.1 PERMUTATION MATRICES

**Definition 4.1.** Let $n$ be a positive integer. A matrix of size $n \times n$ is called a *permutation matrix* if it has exactly one entry equal to $1$ in each row and each column, and zeros elsewhere. We denote by $\mathcal{P}_n$ the set of such all permutation matrices.

*Remark* 4 (Permutation matrix vs. permutation). For every permutation matrix $P \in \mathcal{P}_n$, there exists a unique permutation $\pi \in \mathcal{S}_n$ such that $P$ is obtained by permuting the $n$ columns of the identity matrix $I_n$ according to $\pi$. In this case, we write $P := P_\pi$ and call it the *permutation matrix* corresponding to $\pi$. Here, $\mathcal{S}_n$ is the group of all permutations of the set $\{1, 2, \ldots, n\}$. In particular, for permutation matrices $P_\pi \in \mathcal{P}_n$ and $P_\sigma \in \mathcal{P}_m$, we have

$$P_\pi \cdot \mathbf{x} = \left(x_{\pi^{-1}(1)}, x_{\pi^{-1}(2)}, \ldots, x_{\pi^{-1}(n)}\right)^{\top}, \tag{6}$$

and

$$\left(P_\pi \cdot A \cdot P_\sigma\right)_{ij} = A_{\pi^{-1}(i)\sigma(j)}, \tag{7}$$

for all vector $\mathbf{x} \in \mathbb{R}^n$ and matrix $A \in \mathbb{R}^{n \times m}$.

## 4.2 WEIGHT SPACE

A standard transformer block, which will be denoted by $\mathrm{Attn}$ from now on, contains a multi-head attention and a two-linear-layer ReLU MLP as well as a layer normalization. Formally, $\mathrm{Attn}$ transforms an input sequence $X \in \mathbb{R}^{L \times D}$ to an output sequence $\mathrm{Attn}(X) \in \mathbb{R}^{L \times D}$ defined as follows:

$$\mathrm{Attn}(X) = \mathrm{LayerNorm}\left(\mathrm{ReLU}\left(\hat{X} \cdot W^{(A)} + \mathbf{1}_L \cdot b^{(A)}\right) \cdot W^{(B)} + \mathbf{1}_L \cdot b^{(B)}\right), \qquad (8)$$

where

$$\hat{X} = \mathrm{LayerNorm}\left(\mathrm{MultiHead}\left(X; \{W^{(Q,i)}, W^{(K,i)}, W^{(V,i)}, W^{(O,i)}\}_{i=1}^h\right)\right). \qquad (9)$$

with $\mathbf{1}_L = [1, \ldots, 1]^\top \in \mathbb{R}^{L \times 1}$. This transformer block is parametrized by the following matrices:

- $\left(W^{(Q,i)}, W^{(K,i)}, W^{(V,i)}, W^{(O,i)}\right)_{i=1,\ldots,h} \in \left(\mathbb{R}^{D \times D_k} \times \mathbb{R}^{D \times D_k} \times \mathbb{R}^{D \times D_v} \times \mathbb{R}^{D_v \times D}\right)^h$ inside the multi-head attention, and
- weights $(W^{(A)}, W^{(B)}) \in \mathbb{R}^{D \times D_A} \times \mathbb{R}^{D_A \times D}$ and biases $(b^{(A)}, b^{(B)}) \in \mathbb{R}^{1 \times D_A} \times \mathbb{R}^{1 \times D}$ inside the two-linear-layers ReLU MLP.

The weight space of this transformer block, say $\mathcal{U}$, is the vector space:

$$\mathcal{U} = \left(\mathbb{R}^{D \times D_k} \times \mathbb{R}^{D \times D_k} \times \mathbb{R}^{D \times D_v} \times \mathbb{R}^{D_v \times D}\right)^h \times \left(\mathbb{R}^{D \times D_A} \times \mathbb{R}^{D_A \times D}\right) \times \left(\mathbb{R}^{1 \times D_A} \times \mathbb{R}^{1 \times D}\right). \tag{10}$$

We write each element $U \in \mathcal{U}$ in the form

$$U = \left(\left([W]^{(Q,i)}, [W]^{(K,i)}, [W]^{(V,i)}, [W]^{(O,i)}\right)_{i=1,\ldots,h}, \left([W]^{(A)}, [W]^{(B)}\right), \left([b]^{(A)}, [b]^{(B)}\right)\right). \tag{11}$$

To emphasize the weights of each $\mathrm{Attn}$ map, we will write $\mathrm{Attn}(X; U)$ instead of $\mathrm{Attn}(X)$. In particular, each element $U \in \mathcal{U}$ will define a map $\mathrm{Attn}(\cdot; U) \colon \mathbb{R}^{L \times D} \to \mathbb{R}^{L \times D}$, which maps a sequence $X \in \mathbb{R}^{L \times D}$ to a sequence $\mathrm{Attn}(X; U) \in \mathbb{R}^{L \times D}$ defined by equation 8 and equation 9. In the next section, we will find a sufficient condition for a pair $U$ and $U'$ in $\mathcal{U}$ such that $\mathrm{Attn}(\cdot; U) = \mathrm{Attn}(\cdot; U')$.

## 4.3 GROUP ACTION ON WEIGHT SPACE

Let us consider the weight space $\mathcal{U}$ of a standard transformer block defined in Equation (10) and Equation (11). We consider the symmetries of $\mathcal{U}$ which arise from the following two sources:

- the maximal symmetric group of the multi-head attention, which are characterized in Theorem 3.2;
- the permutation symmetries represented by permutation matrices, which arise from the unordered structure of neurons in the two-layer ReLU MLP.

Based on these observations, we consider the group

$$\mathcal{G}_{\mathcal{U}} = \mathcal{S}_h \times \mathrm{GL}_{D_k}(\mathbb{R})^h \times \mathrm{GL}_{D_v}(\mathbb{R})^h \times \mathcal{P}_D \times \mathcal{P}_{D_A}. \tag{12}$$

This group will serve as the symmetric group of the standard attention block. Each element $g \in \mathcal{G}_{\mathcal{U}}$ has the form

$$g = \left(\tau, (M_i)_{i=1,\ldots,h}, (N_i)_{i=1,\ldots,h}, P_{\pi_O}, P_{\pi_A}\right), \tag{13}$$

where $\tau \in \mathcal{S}_h$, $\pi_O \in \mathcal{S}_D$, $\pi_A \in \mathcal{S}_{D_A}$ are permutations and $M_i, N_i$ are invertible matrices of appropriate sizes. The action of $\mathcal{G}_{\mathcal{U}}$ on $\mathcal{U}$ is defined formally as follows:

**Definition 4.2** (Group action). With notation as above, the action of $\mathcal{G}_{\mathcal{U}}$ on $\mathcal{U}$ is defined to be a map $\mathcal{G}_{\mathcal{U}} \times \mathcal{U} \to \mathcal{U}$ which maps an element $(g, U) \in \mathcal{G}_{\mathcal{U}} \times \mathcal{U}$ with $U$ and $g$ given in equation 11 and equation 13 to the element

$$gU = \left(\left([gW]^{(Q,i)}, [gW]^{(K,i)}, [gW]^{(V,i)}, [gW]^{(O,i)}\right)_{i=1,\ldots,h}, \right.$$
$$\left. \left([gW]^{(A)}, [gW]^{(B)}\right), \left([gb]^{(A)}, [gb]^{(B)}\right)\right), \qquad (14)$$

where

$$[gW]^{(Q,i)} = [W]^{(Q,\tau(i))} \cdot \left(M^{(\tau(i))}\right)^{\top}, \quad [gW]^{(K,i)} = [W]^{(K,\tau(i))} \cdot \left(M^{(\tau(i))}\right)^{-1},$$

$$[gW]^{(V,i)} = [W]^{(V,\tau(i))} \cdot N^{(\tau(i))}, \quad [gW]^{(O,i)} = \left(N^{(\tau(i))}\right)^{-1} \cdot [W]^{(O,\tau(i))} \cdot P_{\pi_O},$$

$$[gW]^{(A)} = P_{\pi_O}^{-1} \cdot [W]^{(A)} \cdot P_{\pi_A}, \quad [gW]^{(B)} = P_{\pi_A}^{-1} \cdot [W]^{(B)},$$

$$[gb]^{(A)} = [b]^{(A)} \cdot P_{\pi_A}, \quad [gb]^{(B)} = [b]^{(B)}.$$

We conclude the construction of the group action on $\mathcal{U}$ in the following theorem which results in a design principle for transformer blocks.

**Theorem 4.3** (Invariance of $\mathrm{Attn}$ under the action of $\mathcal{G}_{\mathcal{U}}$). *With notations as above, we have*

$$\mathrm{Attn}(X; gU) = \mathrm{Attn}(X; U) \tag{15}$$

*for all $U \in \mathcal{U}$, $g \in \mathcal{G}$, and $X \in \mathbb{R}^{L \times D}$.*

*Remark* 5 (Other types of symmetries). The group $\mathcal{G}_{\mathcal{U}}$ defined above does not cover all symmetries of the weight space $\mathcal{U}$. Indeed, there are symmetries of scaling type arising from the internal architecture of the ReLU MLP (see (Godfrey et al., 2022; Tran et al., 2024a)). In addition, layer normalization also offers additional scaling and sign-flipping symmetries for $\mathcal{U}$ (see Appendix B). We leave designing the maximal symmetric group of a transformer block for future study.

## 5 EQUIVARIANT POLYNOMIAL NFNS FOR TRANSFORMERS

In this section, we introduce a family of NFNs specifically designed for Transformers, referred to as Transformer-NFNs, which are equivariant to the group $\mathcal{G}_{\mathcal{U}}$ as described in equation 12. The main building blocks of Transformer-NFNs consist of $\mathcal{G}_{\mathcal{U}}$-equivariant and $\mathcal{G}_{\mathcal{U}}$-invariant polynomial layers. We will outline the construction of a class of $\mathcal{G}_{\mathcal{U}}$-invariant polynomial layers below. The detailed construction of the $\mathcal{G}_{\mathcal{U}}$-equivariant polynomial layers, which follows a similar approach, will be thoroughly derived in Appendix D.

In concrete, we define a polynomial map $I \colon \mathcal{U} \to \mathbb{R}^{D'}$ with maps each element $U \in \mathcal{U}$ to the vector $I(U) \in \mathbb{R}^{D'}$ of the form

$$
\begin{aligned}
I(U) = {} & \sum_{s=1}^{h} \sum_{p=1}^{D} \sum_{q=1}^{D} \Phi_{(QK,s):p,q} \cdot [WW]_{p,q}^{(QK,s)} + \sum_{s=1}^{h} \sum_{p=1}^{D} \sum_{q=1}^{D} \Phi_{(VO,s):p,q} \cdot [WW]_{p,q}^{(VO,s)} \\
& + \sum_{s=1}^{h} \sum_{p=1}^{D} \sum_{q=1}^{D_k} \Phi_{(Q,s):p,q} \cdot [W]_{p,q}^{(Q,s)} + \sum_{s=1}^{h} \sum_{p=1}^{D} \sum_{q=1}^{D_k} \Phi_{(K,s):p,q} \cdot [W]_{p,q}^{(K,s)} \\
& + \sum_{s=1}^{h} \sum_{p=1}^{D} \sum_{q=1}^{D_v} \Phi_{(V,s):p,q} \cdot [W]_{p,q}^{(V,s)} + \sum_{s=1}^{h} \sum_{p=1}^{D_v} \sum_{q=1}^{D} \Phi_{(O,s):p,q} \cdot [W]_{p,q}^{(O,s)} \\
& + \sum_{p=1}^{D} \sum_{q=1}^{D_A} \Phi_{(A):p,q} \cdot [W]_{p,q}^{(A)} + \sum_{p=1}^{D_A} \sum_{q=1}^{D} \Phi_{(B):p,q} \cdot [W]_{p,q}^{(B)} \\
& + \sum_{q=1}^{D_A} \Phi_{(A):q} \cdot [b]_{q}^{(A)} + \sum_{q=1}^{D} \Phi_{(B):q} \cdot [b]_{q}^{(B)} + \Phi_1,
\end{aligned}
\tag{16}
$$

where

$$[WW]^{(QK,s)} := [W]^{(Q,s)} \cdot \left([W]^{(K,s)}\right)^{\top}, \quad \text{and} \quad [WW]^{(VO,s)} := [W]^{(V,s)} \cdot [W]^{(O,s)}, \tag{17}$$

and the coefficients $\Phi_{\_}$s are matrices of size $D' \times 1$.

*Remark* 6 ($I(U)$ as a quadratic polynomial). Intuitively speaking, $I(U)$ is a linear combination of all entries of the matrices $[W]^{(Q,s)}$, $[W]^{(K,s)}$, $[W]^{(V,s)}$, $[W]^{(O,s)}$, $[W]^{(A)}$, $[W]^{(B)}$, $[b]^{(A)}$, and $[b]^{(B)}$ in $U$, as well as all entries of the additional matrices $[WW]^{(QK,s)}$ and $[WW]^{(VO,s)}$ defined in Equation (17). Since the entries of the matrices $[WW]^{(QK,s)}$ and $[WW]^{(VO,s)}$ are polynomials of degree 2, the map $I(U)$ is indeed a quadratic polynomial in the entries of $U$. These additional quadratic terms help us incorporate more relations between weights inside the input multi-head attention block, thus allowing Transformer-NFN to maintain its expressivity.

The above formula for $I(U)$ is irredundant in the following sense:

**Proposition 5.1.** *With notation as above, if $I(U) = 0$ for all $U \in \mathcal{U}$, then $\Phi_- = 0$ for all coefficients $\Phi_-$.*

To make $I$ to be $\mathcal{G}_\mathcal{U}$-invariant, the parameters $\Phi_-$ must satisfy a system of constraints (usually called *parameter sharing*), which are induced from the condition $I(gU) = I(U)$ for all $g \in \mathcal{G}_\mathcal{U}$ and $U \in \mathcal{U}$. We show in details what are these constraints and how to derive the concrete formula of $I$ in Appendix E. The formula of $I$ is then determined by

$$
\begin{aligned}
I(U) = \sum_{p=1}^{D}\sum_{q=1}^{D} \Phi_{(QK,\bullet):p,q} \cdot \left( \sum_{s=1}^{h} [WW]_{p,q}^{(QK,s)} \right) + \sum_{p=1}^{D}\sum_{q=1}^{D} \Phi_{(VO,\bullet):p,\bullet} \cdot \left( \sum_{s=1}^{h} [WW]_{p,q}^{(VO,s)} \right) \\
+ \sum_{p=1}^{D}\sum_{q=1}^{D_A} \Phi_{(A):\bullet,\bullet} \cdot [W]_{p,q}^{(A)} + \sum_{p=1}^{D_A}\sum_{q=1}^{D} \Phi_{(B):\bullet,q} \cdot [W]_{p,q}^{(B)} \\
+ \sum_{q=1}^{D_A} \Phi_{(A):\bullet} \cdot [b]_q^{(A)} + \sum_{q=1}^{D} \Phi_{(B):q} \cdot [b]_q^{(B)} + \Phi_1
\end{aligned}
\tag{18}
$$

In the above formula, the bullet $\bullet$ indicates that the value of the corresponding coefficient is independent of the index at the bullet.

**Theorem 5.2.** *With notation as above, the polynomial map $I : \mathcal{U} \to \mathbb{R}^{D'}$ defined by Equation (18) is $\mathcal{G}_\mathcal{U}$-invariant. Moreover, if a map given in Equation (16) is $\mathcal{G}_\mathcal{U}$-invariant, then it has the form given in Equation (18).*

The concrete formula for the polynomial $\mathcal{G}_\mathcal{U}$-equivariant layer is presented in detail in Appendix D.

## 6 THE SMALL TRANSFORMER ZOO DATASET

Large-scale empirical studies have produced datasets of trained classification models with varied hyperparameters (Eilertsen et al.; Unterthiner et al., 2020), enabling data-driven approaches to modeling generalization. However, a dataset of Transformer-based classification models is absent. Motivated by this shortage, we introduce the Small Transformer Zoo dataset to experimentally demonstrate the efficacy of our proposed method. This dataset contains the weights of a fixed Transformer architecture trained on two distinct datasets, spanning both vision and language tasks. Each entry in the dataset includes a checkpoint weight along with its corresponding accuracy metrics and hyperparameter configurations.

**General settings.** We focus on two prevalent deep learning tasks: *digit image classification* using the MNIST dataset (LeCun & Cortes, 2005) for vision, and *text topic classification* using AGNews (Zhang et al., 2015) for natural language processing. This selection covers two primary data modalities in deep learning—image and text—while addressing classification tasks, which are among the most common and fundamental in the field. Our model architecture contains three components: an embedding layer, an encoder, and a classifier. The embedding layer processes raw input data to produce initial token embeddings, which the encoder then transforms through self-attention mechanisms to capture contextual relationships. Finally, the classifier generates the classification output based on these enriched embeddings. While we adapt the embedding and classifier components to each specific task, we maintain a consistent encoder architecture across both tasks, consisting of two stacked two-head transformer blocks as defined in Equation (8).

Our resulting datasets, named MNIST-Transformers and AGNews-Transformers, consist of 62756 and 63796 model checkpoints, respectively. These models were generated by varying key hyperparameters such as the optimizer, learning rate, weight regularization, weight initialization, and dropout, as detailed in Appendix G. By making these datasets available, we aim to facilitate and inspire further research into the inner workings and behavior of Transformer models.

## 7 EXPERIMENTAL RESULTS

We empirically evaluate the performance of the proposed Transformer-NFN model on two datasets: MNIST-Transformers and AGNews-Transformers. Additionally, we conduct ablation studies to examine the contribution of each component within the Transformer architecture in predicting network

Table 1: Performance measured by Kendall's $\tau$ of all models on MNIST-Transformers dataset. Uncertainties indicate standard error over 5 runs.

| | Accuracy threshold | | | | |
| | No threshold | 20% | 40% | 60% | 80% |
|---|---|---|---|---|---|
| MLP | $0.866 \pm 0.002$ | $0.873 \pm 0.001$ | $0.874 \pm 0.003$ | $0.874 \pm 0.006$ | $0.873 \pm 0.007$ |
| STATNN (Unterthiner et al., 2020) | $0.881 \pm 0.001$ | $0.872 \pm 0.001$ | $0.868 \pm 0.001$ | $0.86 \pm 0.001$ | $0.856 \pm 0.001$ |
| XGBoost (Chen & Guestrin, 2016) | $0.860 \pm 0.002$ | $0.839 \pm 0.004$ | $0.869 \pm 0.003$ | $0.846 \pm 0.001$ | $0.884 \pm 0.001$ |
| LightGBM (Ke et al., 2017) | $0.858 \pm 0.002$ | $0.835 \pm 0.001$ | $0.847 \pm 0.001$ | $0.822 \pm 0.001$ | $0.830 \pm 0.001$ |
| Random Forest (Breiman, 2001) | $0.772 \pm 0.002$ | $0.758 \pm 0.004$ | $0.769 \pm 0.001$ | $0.752 \pm 0.001$ | $0.759 \pm 0.001$ |
| Transformer-NFN (ours) | $\mathbf{0.905 \pm 0.002}$ | $\mathbf{0.899 \pm 0.001}$ | $\mathbf{0.895 \pm 0.001}$ | $\mathbf{0.895 \pm 0.002}$ | $\mathbf{0.888 \pm 0.002}$ |

Table 2: Performance measured by Kendall's $\tau$ of all models on AGNews-Transformers dataset. Uncertainties indicate standard error over 5 runs.

| | Accuracy threshold | | | | |
| | No threshold | 20% | 40% | 60% | 80% |
|---|---|---|---|---|---|
| MLP | $0.879 \pm 0.006$ | $0.875 \pm 0.001$ | $0.841 \pm 0.012$ | $0.842 \pm 0.001$ | $0.862 \pm 0.006$ |
| STATNN (Unterthiner et al., 2020) | $0.841 \pm 0.002$ | $0.839 \pm 0.003$ | $0.812 \pm 0.001$ | $0.813 \pm 0.001$ | $0.812 \pm 0.001$ |
| XGBoost (Chen & Guestrin, 2016) | $0.859 \pm 0.001$ | $0.852 \pm 0.002$ | $0.872 \pm 0.002$ | $0.874 \pm 0.001$ | $0.872 \pm 0.001$ |
| LightGBM (Ke et al., 2017) | $0.835 \pm 0.001$ | $0.845 \pm 0.001$ | $0.837 \pm 0.001$ | $0.835 \pm 0.001$ | $0.820 \pm 0.001$ |
| Random Forest (Breiman, 2001) | $0.774 \pm 0.003$ | $0.801 \pm 0.001$ | $0.797 \pm 0.001$ | $0.798 \pm 0.002$ | $0.773 \pm 0.001$ |
| Transformer-NFN (ours) | $\mathbf{0.910 \pm 0.001}$ | $\mathbf{0.908 \pm 0.001}$ | $\mathbf{0.897 \pm 0.001}$ | $\mathbf{0.896 \pm 0.001}$ | $\mathbf{0.890 \pm 0.001}$ |

generalization, and investigate the impact of varying the Transformer-NFN dimension and the number of layers on the overall performance. Our analysis yields three key findings: (1) Transformer-NFN, with its enhanced layers for processing transformer block parameters, outperforms existing baselines in both vision and NLP Transformers datasets, maintains consistent performance across different accuracy thresholds of the data (2) the information embedded in the weights of Transformer blocks provides a strong predictor for the performance of transformer model, and (3) good performance for Transformer-NFN can be obtained with a compact setting.

We use Kendall's $\tau$ rank correlation (Kendall, 1938), ranging from $[-1, 1]$, as the evaluation metric to assess how closely predicted accuracy rankings align with ground truth accuracy rankings. A value near 1 indicates strong agreement, as shown in the scatterplot in Figure 2 (in Appendix H). All results in this section are averaged over 5 runs with different random seeds, with details on hyperparameters and training settings provided in Appendix H.

### 7.1 PREDICTING VISION TRANSFORMERS GENERALIZATION FROM PRETRAINED WEIGHTS

**Experiment Setup.** In this experiment, we focus on predicting the test accuracy of pretrained Vision Transformer models using only their weights, without access to the test set. To perform this task, we utilize our MNIST-Transformers dataset. We evaluate our model against 5 models: MLP, STATNN (Unterthiner et al., 2020), XGBoost (Chen & Guestrin, 2016), LightGBM (Ke et al., 2017), and Random Forest (Breiman, 2001). As shown in Figure 1 (in Appendix H), the accuracy distribution of the MNIST-Transformers dataset is highly skewed (notice the log scale on the y-axis). Therefore, we evaluate each model's prediction performance not only on the entire dataset but also on four smaller subsets, each filtered by accuracy thresholds of 20%, 40%, 60%, and 80%. As a significant portion of pretrained models in the dataset exhibit higher accuracy, achieving a strong Kendall's $\tau$ correlation becomes increasingly challenging as the accuracy thresholds increase.

**Results.** Table 1 illustrates the results of all models when predicting generalization of Vision Transformer networks trained on MNIST-Transformer dataset. As expected, Kendall's $\tau$ generally decreases as the accuracy threshold increases. In addition, our Transformer-NFN consistently outperforms all four baseline models across all dataset settings with performance gap ranging from 0.004 to 0.026, demonstrating the effectiveness of our model's design in capturing the information within each transformer block.

### 7.2 PREDICTING TEXT CLASSIFICATION TRANSFORMERS GENERALIZATION

**Experiment Setup.** In this experiment, we utilize the AGNews-Transformers dataset to predict the performance of pretrained transformer models in text classification. The goal is to evaluate the effectiveness of Transformer-NFN in predicting the performance of pretrained models specifically trained on language tasks. Similar to Experiment 7.1, we assess our model's capabilities across different dataset configurations by using five subsets: the entire dataset without any accuracy threshold, and four subsets with accuracy thresholds of 20%, 40%, 60%, and 80%, respectively.

Table 3: Ablation study the important of each component of the input networks on predicting generalization of the input network, the metric being used is Kendall's $\tau$.

| No. of components | Components | MNIST-Transformers | AGNews-Transformers |
|---|---|---|---|
| 1 | Encoder | **0.902 ± 0.001** | **0.909 ± 0.001** |
| | Embedding | 0.424 ± 0.002 | - |
| | Classifier | 0.847 ± 0.001 | 0.795 ± 0.008 |
| 2 | Embedding + Classifier | 0.857 ± 0.003 | - |
| | Encoder + Classifier | **0.904 ± 0.001** | **0.910 ± 0.001** |
| | Encoder + Embedding | 0.903 ± 0.001 | - |
| 3 | Encoder + Embedding + Classifier | **0.905 ± 0.002** | - |

**Results.** Table 2 shows that our model consistently outperforms all baselines across all text classification dataset configurations. Compared to MNIST-Transformers, the performance gain is even more pronounced, with Kendall's $\tau$ gaps ranging from 0.018 to 0.033. As language tasks involve more complex syntax, the transformer encoder captures richer information, allowing Transformer-NFN to predict NLP performance more effectively than in vision tasks. Figure 2 (Appendix H) further illustrates its superiority, particularly in generalization for low-accuracy networks.

### 7.3 IMPORTANCE OF ENCODER IN PREDICTING THE GENERALIZATION

A interesting question arises is how much information about the network generalization ability is embedded in each component of the transformer model. To investigate this, we restrict our Transformer-NFNs to access only certain subsets of the Transformer's components and train the model on MNIST-Transformers dataset and AGNews-Transformers dataset. Our goal is to determine the importance of each component, both individually and in combination, for predicting generalization of the input network.

Table 3 shows that for both MNIST-Transformers and AGNews-Transformers, transformer blocks alone provide strong performance predictions. The classifier is the second most important component, followed by the embedding. Even with only transformer block weights, our model achieves a Kendall's $\tau$ score nearly identical to using all components: 0.902 vs. 0.905 for MNIST-Transformers and 0.909 vs. 0.91 for AGNews-Transformers.

### 7.4 ABALATION STUDY ON VARYING THE DIMENSION AND NUMBER OF LAYERS

In this section, we conduct an ablation study to explore the impact of varying the hidden dimension and the number of equivariant layers in Transformer-NFN. We evaluate different configurations of hidden dimensions and layer counts on the AGNews-Transformer dataset, with dimensions $\in [3, 5, 10, 15]$, and number of layer $\in [1, 2]$. These configurations allow us to assess how changes in model size affect performance and efficiency.

Table 5 (in Appendix H) demonstrates that strong performance can be achieved with a relatively small dimension and few parameters. For example, with dimension of 15 and a single layer, the model reaches a Kendall's $\tau$ score of 0.913, matching the best performance across all settings. These results suggest that good performance for our Transformer-NFN can be obtained with a compact setting.

## 8 CONCLUSION

In this work, we made significant contributions to the understanding and application of NFNs in transformer architectures. We determined the maximal symmetric group of the weights in a multi-head attention module. We also formally defined the weight space of a transformer architecture and introduced a group action on this weight space, thereby characterizing the design principles for NFNs. Additionally, we presented Transformer-NFN, an NFN designed for transformer architectures that is equivariant under the specified group action. Finally, we released a dataset of more than 125,000 transformers model checkpoints trained on two datasets with two different tasks, marking a significant resource for benchmarking the applicability and efficiency of Transformer-NFN and promoting further research to enhance our understanding of transformer network training and performance.

ACKNOWLEDGMENTS

This research / project is supported by the National Research Foundation Singapore under the AI Singapore Programme (AISG Award No: AISG2-TC-2023-012-SGIL). This research / project is supported by the Ministry of Education, Singapore, under the Academic Research Fund Tier 1 (FY2023) (A-8002040-00-00, A-8002039-00-00). This research / project is supported by the NUS Presidential Young Professorship Award (A-0009807-01-00) and the NUS Artificial Intelligence Institute–Seed Funding (A-8003062-00-00).

Thieu N. Vo is supported by the Singapore National Academy of Science under the SASEA Fellowship Programme (Award No: NRF-MP-2025-0001).

Thanh Tran acknowledges support from the Application Driven Mathematics Program funded and organized by the Vingroup Innovation Fund and VinBigData. Thanh Tran acknowledges support from the VinUni's Student Research Grant Program AY24-25.

**Ethics Statement.** Given the nature of the work, we do not foresee any negative societal and ethical impacts of our work.

**Reproducibility Statement.** Source codes for our experiments are provided in the supplementary materials of the paper. The details of our experimental settings are given in Section 7 and the Appendix H. All datasets used in this paper are publicly available through an anonymous link provided in the README file of the supplementary material.

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

# Supplement to "Equivariant Neural Functional Networks for Transformers"

**Table of Contents**

# A   MAXIMAL SYMMETRIC GROUP OF MultiHead

This section aims to provide a complete answer to the question of when two sets of parameters of a multi-head attention mechanism will define the same function. The answer to this question will serve as the core of the design principle of NFNs for Transformers. We recall the formulation of Head and MultiHead attention maps in Section A.1. Then, the maximal symmetric group of Head is derived in Section A.2.

## A.1   MULTI-HEAD ATTENTION

Let $D$ be a positive integer. Recall the notion of the parameterized map Head as follows: For a positive integer $L$ and $X \in \mathbb{R}^{L \times D}$, we have

$$\text{Head}\left(X; W^{(Q)}, W^{(K)}, W^{(V)}\right) = \text{softmax}\left(\frac{\left(XW^{(Q)}\right) \cdot \left(XW^{(K)}\right)^{\top}}{\sqrt{D_k}}\right) \cdot \left(XW^{(V)}\right),$$

where $W^{(Q)}, W^{(K)} \in \mathbb{R}^{D \times D_k}$ and $W^{(V)} \in \mathbb{R}^{D \times D_v}$. By definition, we have

$$\text{Head}\left(\cdot; W^{(Q)}, W^{(K)}, W^{(V)}\right) : \bigsqcup_{l>0} \mathbb{R}^{l \times D} \to \bigsqcup_{l>0} \mathbb{R}^{l \times D_v},$$

and for all $l > 0$, the image of $\mathbb{R}^{l \times D}$ is contained in $\mathbb{R}^{l \times D_v}$. By combining $\left(W^{(Q)}\right) \cdot \left(W^{(K)}\right)^{\top}/\sqrt{D_k} \in \mathbb{R}^{D \times D}$, we can rewrite the map Head as follows

$$\text{Head}\left(X; W^{(Q)}, W^{(K)}, W^{(V)}\right) = \text{softmax}\left(X \cdot \frac{\left(W^{(Q)}\right) \cdot \left(W^{(K)}\right)^{\top}}{\sqrt{D_k}} \cdot X^{\top}\right) \cdot \left(XW^{(V)}\right),$$

By this observation, we define a class of parameterized maps as follows: For $X \in \mathbb{R}^{L \times D}$, we have

$$f(X; A) := \text{softmax}\left(XAX^{\top}\right) \cdot X,$$

where $A \in \mathbb{R}^{D \times D}$. Similarly, for the MultiHead map, we have

$$\text{MultiHead}\left(X; W^{(O)}, \left\{W^{(Q,i)}, W^{(K,i)}, W^{(V,i)}\right\}_{i=1}^{h}\right)$$

$$= \left(\bigoplus_{i=1}^{h} \text{Head}\left(X; W^{(Q,i)}, W^{(K,i)}, W^{(V,i)}\right)\right) W^{(O)}$$

$$= \sum_{i=1}^{h} \text{Head}\left(X; W^{(Q,i)}, W^{(K,i)}, W^{(V,i)}\right) W^{(O,i)}$$

$$= \sum_{i=1}^{h} \text{softmax} \left( \frac{\left( XW^{(Q,i)} \right) \cdot \left( XW^{(K,i)} \right)^{\top}}{\sqrt{D_k}} \right) \cdot X \cdot \left( W^{(V,i)} \cdot W^{(O,i)} \right),$$

where $h$ is a positive integer, $W^{(O)} = \left( W^{(O:1)}, \ldots, W^{(O:h)} \right)$ with each $W^{(O:i)} \in \mathbb{R}^{D_v \times D}$. Considering $W^{(V,i)} \cdot W^{(O,i)}$ as a matrix $B_i \in \mathbb{R}^{D \times D}$, we define a new class of parameterized maps as follows: For $X \in \mathbb{R}^{L \times D}$, we have

$$F \left( X; \{A_i, B_i\}_{i=1}^{h} \right) = \sum_{i=1}^{h} f(X; A_i) \cdot B_i,$$

where $h$ is a positive integer and $A_i, B_i \in \mathbb{R}^{D \times D}$. Note that,

$$\text{rank} \left( \left( W^{(Q)} \right) \cdot \left( W^{(K)} \right)^{\top} / \sqrt{D_k} \right) \leqslant \min(D, D_k) \leqslant D$$

$$\text{rank} \left( W^{(V,i)} \cdot W^{(O,i)} \right) \leqslant \min(D, D_v) \leqslant D$$

So, in general, the new class of $F$ maps contains the class of MultiHead maps. Note that, $F$ simply is a weighted summation of some $f$ maps and is linear with respect to $\{B_i\}_{i=1}^{h}$.

*Remark* 7. We can see, in $f$, the matrix $A$ plays the role as a core matrix that defines $f$. Similarly, in $F$, each $A_i$ defines each $f_i$, and $B_i$ is the weight of each component contributes to $F$.

## A.2 MAXIMAL SYMMETRIC GROUP OF MULTI-HEAD ATTENTION

In this section, we present a theoretical result that shows the following: Roughly speaking, in the multi-head scenario, each individual head plays its own unique role. The main results are Theorems A.1 (which corresponds to Theorem 3.1 in the main text) on the unique role of each head and Theorem A.7 (which corresponds to Theorem 3.2 in the main text) on the maximal symmetric group of multi-head attention.

**Theorem A.1.** *Let $D$ be a positive integer. Assume that for a positive integer $h$, matrices $A_1, A_2, \ldots, A_h \in \mathbb{R}^{D \times D}$ and $B_1, B_2, \ldots, B_h \in \mathbb{R}^{D \times D}$, we have*

$$F \left( X; \{A_i, B_i\}_{i=1}^{h} \right) = \sum_{i=1}^{h} f(X; A_i) \cdot B_i = 0 \tag{19}$$

*for all positive integers $L$ and $X \in \mathbb{R}^{L \times D}$. Then, if $A_1, A_2, \ldots, A_h$ are pairwise distinct, then*

$$B_1 = \ldots = B_h = 0.$$

*Proof.* From Equation (19), we have

$$\sum_{i=1}^{h} \text{softmax} \left( X A_i X^{\top} \right) \cdot X \cdot B_i = 0 \tag{20}$$

Consider $L > 1$. We write $X = (x, y, y, \ldots, y)^{\top} \in \mathbb{R}^{L \times D}$ where $x = (x_1, \ldots, x_D)$ and $y = (y_1, \ldots, y_D)$ in $\mathbb{R}^{1 \times D}$ (So in $X$, $y$ appears $L - 1$ times). We will consider the entry in the first row and first column of both sides of Equation (20). Let $b_1, b_2, \ldots, b_h \in \mathbb{R}^{D \times 1}$ be the first column of matrices $B_1, B_2, \ldots, B_h$. From Equation (20), we have

$$\sum_{i=1}^{h} \left( \frac{e^{x A_i x^{\top}}}{e^{x A_i x^{\top}} + L e^{x A_i y^{\top}}} \cdot x + \frac{L e^{x A_i y^{\top}}}{e^{x A_i x^{\top}} + L e^{x A_i y^{\top}}} \cdot y \right) \cdot b_i = 0 \tag{21}$$

By substituting $x = y = (1, 0, \ldots, 0) \in \mathbb{R}^D$ to Equation (21), we have the sum of all first entries of $b_1, \ldots, b_h$ is equal to 0. Similarly, for every $j = 1, \ldots, D$, the sum of $j^{\text{th}}$ entries of $b_1, \ldots, b_h$ is equal to 0. It shows that

$$b_1 + b_2 + \ldots + b_h = 0. \tag{22}$$

From Equation (21), we have

$$0 = \sum_{i=1}^{h} \left( \frac{e^{x A_i x^{\top}}}{e^{x A_i x^{\top}} + L e^{x A_i y^{\top}}} \cdot x + \frac{L e^{x A_i y^{\top}}}{e^{x A_i x^{\top}} + L e^{x A_i y^{\top}}} \cdot y \right) \cdot b_i \tag{23}$$

$$= \sum_{i=1}^{h} \left( \frac{e^{xA_ix^\top} + Le^{xA_iy^\top}}{e^{xA_ix^\top} + Le^{xA_iy^\top}} \cdot x + \frac{Le^{xA_iy^\top}}{e^{xA_ix^\top} + Le^{xA_iy^\top}} \cdot (y - x) \right) \cdot b_i \tag{24}$$

$$= \sum_{i=1}^{h} \left( x + \frac{Le^{xA_iy^\top}}{e^{xA_ix^\top} + Le^{xA_iy^\top}} \cdot (y - x) \right) \cdot b_i \tag{25}$$

$$= \sum_{i=1}^{h} x \cdot b_i + \sum_{i=1}^{h} \frac{Le^{xA_iy^\top}}{e^{xA_ix^\top} + Le^{xA_iy^\top}} \cdot (y - x) \cdot b_i \tag{26}$$

$$= x \cdot \left( \sum_{i=1}^{h} b_i \right) + (y - x) \cdot \left( \sum_{i=1}^{h} \frac{Le^{xA_iy^\top}}{e^{xA_ix^\top} + Le^{xA_iy^\top}} \cdot b_i \right) \tag{27}$$

$$= (y - x) \cdot \left( \sum_{i=1}^{h} \frac{Le^{xA_iy^\top}}{e^{xA_ix^\top} + Le^{xA_iy^\top}} \cdot b_i \right) \tag{28}$$

$$= (y - x) \cdot \left( \sum_{i=1}^{h} \frac{L}{e^{xA_ix^\top - xA_iy^\top} + L} \cdot b_i \right) \tag{29}$$

$$= (y - x) \cdot \left( \sum_{i=1}^{h} \frac{L}{e^{xA_i(x-y)^\top} + L} \cdot b_i \right) \tag{30}$$

By let $z = x - y$, from the above equations, we have

$$z \cdot \left( \sum_{i=1}^{h} \frac{L}{e^{xA_iz^\top} + L} \cdot b_i \right) = 0, \tag{31}$$

or

$$z \cdot \left( \sum_{i=1}^{h} \frac{1}{e^{xA_iz^\top} + L} \cdot b_i \right) = 0, \tag{32}$$

for all $x = (x_1, \ldots, x_D)$ and $z = (z_1, \ldots, z_D)$ in $\mathbb{R}^D$. Now, each $xA_iz^\top$ can be viewed as a polynomial in $2D$ indeterminates $x_1, \ldots, x_D$ and $z_1, \ldots, z_D$ as follows

$$xA_iz^\top = \sum_{p=1}^{D} \sum_{q=1}^{D} (A_i)_{p,q} \cdot x_p z_q. \tag{33}$$

Since $A_1, \ldots, A_h$ are pairwise distinct, so $xA_1z^\top, \ldots, xA_hz^\top$ are pairwise distinct polynomials. By Lemma A.3, there exists $u \in \mathbb{R}^D$ and a non-empty open set $V \in \mathbb{R}^D$, such that for all $z \in V$, we have $uA_1z^\top, \ldots, uA_hz^\top$ are pairwise distinct real numbers. Obviously $0 \notin V$. Now fix an $v \in V$ and let $z = t \cdot v$ for $t \in \mathbb{R}$. Denote $uA_iv^\top = s_i$, from Equation (33), we have

$$t \cdot v \cdot \left( \sum_{i=1}^{h} \frac{1}{e^{t \cdot s_i} + L} \cdot b_i \right) = 0 \tag{34}$$

for all $t \in \mathbb{R}$, or

$$v \cdot \left( \sum_{i=1}^{h} \frac{1}{e^{t \cdot s_i} + L} \cdot b_i \right) = 0 \tag{35}$$

for all $t \in \mathbb{R} \setminus \{0\}$. By continuity, this still holds for $t = 0$. So we have

$$v \cdot \left( \sum_{i=1}^{h} \frac{1}{e^{t \cdot s_i} + L} \cdot b_i \right) = 0 \tag{36}$$

for all $t \in \mathbb{R}$. Now, consider the set

$$S := \left\{ \left( \frac{1}{e^{t \cdot s_1} + L}, \frac{1}{e^{t \cdot s_2} + L}, \ldots, \frac{1}{e^{t \cdot s_h} + L} \right) \in \mathbb{R}^h : t \in \mathbb{R} \right\}. \tag{37}$$

From Equation (36), we have the linear span of $S$, i.e. $\mathrm{span}(S)$, satisfies that: For all $s = (s_1, s_2, \ldots, s_h) \in \mathrm{span}(S)$, we have

$$v \cdot \left( \sum_{i=1}^{h} s_i \cdot b_i \right) = 0. \tag{38}$$

In other words, $v$ and $\sum_{i=1}^{h} s_i \cdot b_i$ are orthogonal to each other as two vectors in $\mathbb{R}^D$. Since $s_1, \ldots, s_h \in \mathbb{R}$ are pairwise distinct, by Lemma A.4, there exist $t_1, \ldots, t_h \in \mathbb{R}$ such that their $h$ corresponding vectors in $S$ form a basis of $\mathbb{R}^h$. This implies $\text{span}(S) = \mathbb{R}^h$. We have the set

$$\text{span}\left(\{b_1, \ldots, b_h\}\right) = \left\{\sum_{i=1}^{h} s_i \cdot b_i \ : \ s = (s_1, \ldots, s_h) \in \mathbb{R}^h \right\} \tag{39}$$

$$= \left\{\sum_{i=1}^{h} s_i \cdot b_i \ : \ s = (s_1, \ldots, s_h) \in \text{span}(S) \right\}. \tag{40}$$

By the previous observation, it implies that $v$ is orthogonal to every vectors in $\text{span}\left(\{b_1, \ldots, b_h\}\right)$. In other words, the orthogonal complement of $\text{span}\left(\{b_1, \ldots, b_h\}\right)$, which is denoted by $\text{span}\left(\{b_1, \ldots, b_h\}\right)^{\perp}$, contains $v$. This holds for every vectors $v$ in the non-empty open set $V$, so

$$V \subset \text{span}\left(\{b_1, \ldots, b_h\}\right)^{\perp}. \tag{41}$$

Since $\text{span}\left(\{b_1, \ldots, b_h\}\right)^{\perp}$ is a linear subspace of $\mathbb{R}^D$ that contains a non-empty open set of $\mathbb{R}^d$, so by Lemma A.2, we have $\text{span}\left(\{b_1, \ldots, b_h\}\right)^{\perp} = \mathbb{R}^D$. This implies $\text{span}\left(\{b_1, \ldots, b_h\}\right) = 0$, which means $b_1 = \ldots = b_h = 0$. So the first column of $B_1, \ldots, B_h$ are all equal to 0, and a similar proof is applied for every other columns of $B_1, \ldots, B_h$. So $B_1 = \ldots = B_h = 0$. $\qquad\square$

**Lemma A.2.** *Let $D$ be a positive integer, and $V$ is a non-empty open set of $\mathbb{R}^D$ with the usual topology. If $U$ is a linear subspace of $\mathbb{R}^D$ that contains $V$, then $U = \mathbb{R}^D$.*

*Proof.* Since $V$ is non-empty, let $x \in V$. Since $V$ is open, there exists $r > 0$ such that the closed ball

$$\bar{B}_r(x) = \{y \in \mathbb{R}^D \ : \ \|x - y\| \leqslant r\} \subset V. \tag{42}$$

Then for all $y \in \mathbb{R}^D$ that $y \neq 0$, we have

$$\left\| x - \left( x + r \cdot \frac{y}{\|y\|} \right) \right\| = r, \tag{43}$$

which means

$$x + r \cdot \frac{y}{\|y\|} \in \bar{B}_r(x) \subset V \subset U. \tag{44}$$

Since $x$ is also in $V \subset U$, and $U$ is a linear subspace, then $y \in U$. So, for $y \in \mathbb{R}^D$ that $y \neq 0$, we have $y \in U$, and clearly $0 \in U$, so $U = \mathbb{R}^D$. $\qquad\square$

*Remark* 8. Lemma A.2 still holds if we replace $\mathbb{R}^D$ by a normed vector space.

**Lemma A.3.** *Let $n, h$ be two positive integers, and $f_1, \ldots, f_h \in \mathbb{R}[x_1, \ldots, x_n]$ be $h$ pairwise distinct real polynomials in $n$ variables. Then there exists a non-empty open subset $U$ of $\mathbb{R}^n$ such that $f_1(x), \ldots, f_h(x)$ are pairwise distinct for all $x \in U$.*

*Proof.* Consider the polynomial

$$f = \prod_{1 \leqslant i < j \leqslant h} (f_i - f_j) \in \mathbb{R}[x_1, \ldots, x_n]. \tag{45}$$

Since $f_1, \ldots, f_h$ are pairwise distinct, then $f_i - f_j$ is non-zero for all $1 \leqslant i < j \leqslant h$. It is well-known that the real polynomial ring is an integral domain, so $f$ is non-zero. There exists $a \in \mathbb{R}^n$ such that $f(a) \neq 0$. Since $\mathbb{R}$ is Hausdorff, we can choose an open set $V$ of $\mathbb{R}$ that contains $a$ and does not contain 0. Let $U = f^{-1}(V)$, and since $f$ is continuous, $U$ is open. We have $f(x) \neq 0$ for all $x \in U$, which means $f_1(x), \ldots, f_n(x)$ are pairwise distinct for all $x \in U$ $\qquad\square$

**Lemma A.4.** *Let $h$ be a positive integer, and $h$ distinct real numbers $s_1, \ldots, s_h$. Then there exists $h$ real numbers $t_1, \ldots, t_h$ and a positive integer $L$ such that the matrix*

$$A = (a_{ij})_{1 \leqslant i \leqslant h, 1 \leqslant j \leqslant h} \in \mathbb{R}^{h \times h}, \quad \text{where } a_{ij} = \frac{1}{e^{t_i \cdot s_j} + L}, \tag{46}$$

*is full rank. In other words, there exists $h$ real numbers $t_1, \ldots, t_h$ and a positive integer $L$ such that $h$ vectors*

$$\left( \frac{1}{e^{t_i \cdot s_1} + L}, \frac{1}{e^{t_i \cdot s_2} + L}, \ldots, \frac{1}{e^{t_i \cdot s_h} + L} \right) \in \mathbb{R}^h \tag{47}$$

*for $i = 1, \ldots, h$, form a basis of $\mathbb{R}^h$.*

*Proof.* We first have an observations. Let $t_1, \ldots, t_h$ be fixed and $x$ be a variable. Consider the matrix

$$A(x) = (a_{ij})_{1 \leqslant i \leqslant h, 1 \leqslant j \leqslant h}, \quad \text{where } a_{ij} = \frac{1}{e^{t_i \cdot s_j} + x} \tag{48}$$

Then the determinant of $A(x)$, denoted by $\det(A(x))$ can be viewed as a real rational function, i.e. a function that can be written as the ratio of two real polynomials. So, in the case that this rational function is zero, $\det(A(x)) = 0$ for all $x \in \mathbb{R}$, and in the case that this rational function is non-zero, there are a finite number of $x \in \mathbb{R}$ such that $\det(A(x)) = 0$ or $\det(A(x))$ is not defined. In other words, $A(x)$ is not full rank for all $x$, or for only a finite number of $x$. So, in Lemma A.4, if there exists $h$ real numbers $t_1, \ldots, t_h$ and a real number $L$ that makes $A$ becomes full rank, then $L$ can be made to be a positive integer.

Back to the problem, we will prove by mathematical induction. We will show that for every $h$, it is possible to choose $t_1, \ldots, t_h$ and $L$ to make $\det(A)$ becomes non-zero. For $h = 1$, then the matrix $A$ is full rank since its single entry is always positive for $t = L = 1$. Assume that the result holds for a positive integer $h - 1$, we will show it holds for $h$. For $j = 1, \ldots, h$, let $B_j$ is the $(h-1) \times (h-1)$ matrix obtained by removing the first row and the $j^{\text{th}}$ column of matrix $A$. By computing the determinant of $A$ via the Laplace expansion along the first row, we have

$$\det(A) = \sum_{j=1}^{h} (-1)^{1+j} \cdot a_{1j} \cdot \det(B_j) = \sum_{j=1}^{h} (-1)^{1+j} \cdot \frac{1}{e^{t_1 \cdot s_j} + 2} \cdot \det(B_j) \tag{49}$$

Denote $c_j = (-1)^{1+j} \cdot \det(B_j)$, and note that $c_j$ depends on the choice of $t_2, \ldots, t_h$. Without loss of generality, assume $s_1 \neq 0$. Since $s_2, \ldots, s_h$ are $h - 1$ pairwise distinct real numbers, by the induction hypothesis, there exists $t_2, \ldots, t_h$ such that $c_1$ is non-zero for at least one $L \in \mathbb{R}$. So with this choice of $t_2, \ldots, t_h$, there exists $\alpha \in \mathbb{R}$ such that

$$c_1 \text{ is non-zero for all } L < \alpha. \tag{50}$$

Since $s_1, \ldots, s_h$ are pairwise distinct and $s_1 \neq 0$, we can choose $t = t_1 \in \mathbb{R}$ such that:

1. $e^{t_1 \cdot s_1} > 1 - \alpha$; and,

2. $|e^{t_1 \cdot s_1} - e^{t_i \cdot s_j}| > 3$; for all $(i, j) \neq (1, 1)$.

With this choice of $t_1$, let $\Delta = [-1 - e^{t_1 \cdot s_1}, 1 - e^{t_1 \cdot s_1}]$. Then for $L \in \Delta$

1. We have $e^{t_1 \cdot s_1} + L \in [-1, 1]$.

2. For $(i, j) \neq (1, 1)$, since $|e^{t_1 \cdot s_1} - e^{t_i \cdot s_j}| > 3$, then $e^{t_i \cdot s_j} + L \notin [-1, 1]$.

3. We have $L \leqslant 1 - e^{t_1 \cdot s_1} < 1 - (1 - \alpha) = \alpha$.

We show that with this choice of $t_1, t_2, \ldots, t_h$, there exists $L \in \mathbb{R}$ such that $\det(A)$ is non-zero. Assume the contrary that

$$\sum_{j=1}^{h} \frac{1}{e^{t_1 \cdot s_j} + L} \cdot c_j = 0 \tag{51}$$

for all $L \in \mathbb{R}$. This implies that

$$\frac{1}{e^{t_1 \cdot s_1} + L} \cdot c_1 = \sum_{j=2}^{h} \frac{1}{e^{t_1 \cdot s_j} + L} \cdot c_j, \tag{52}$$

so

$$\left| \frac{1}{e^{t_1 \cdot s_1} + L} \right| \cdot |c_1| = \left| \sum_{j=2}^{h} \frac{1}{e^{t_1 \cdot s_j} + L} \cdot c_j \right| \leqslant \sum_{j=2}^{h} \left| \frac{1}{e^{t_1 \cdot s_j} + L} \right| \cdot |c_j| \tag{53}$$

Considering $c_1, \ldots, c_h$ as functions in $L$, we have these functions are well-defined on the closed interval $\Delta$, since they are determinants of matrices where their entries are

$$\frac{1}{e^{t_i \cdot s_j} + L} \tag{54}$$

for $1 < i \leqslant h$ and $1 \leqslant j \leqslant h$, and these entries are defined on $\Delta$ by the choice of $t_1, \ldots, t_h$. Moreover, $c_1, \ldots, c_h$ are continuous on $\Delta$. Since a continuous function on a compact set is bounded, so there exists $\delta_1 > 0$ such that

$$|c_1|, \ldots, |c_h| < \delta_1, \quad \text{for all } L \in \Delta. \tag{55}$$

Moreover, since $L \in \Delta$ implies $L < \alpha$, then $|c_1| > 0$ for all $L < \alpha$, which means there exists $\delta_2 > 0$ such that

$$|c_1| > \delta_2, \quad \text{for all } L \in \Delta. \tag{56}$$

Similarly, for $1 < j \leqslant h$,

$$\frac{1}{e^{t_1 \cdot s_j} + L}, \tag{57}$$

considered as functions in $L$, is well-defined and continuous on $\Delta$, so there exist $\delta_3 > 0$ such that

$$\left| \frac{1}{e^{t_1 \cdot s_2} + L} \right|, \ldots, \left| \frac{1}{e^{t_1 \cdot s_h} + L} \right| < \delta_3 \text{ for all } L \in \Delta. \tag{58}$$

From Equation (53), since we have Equations (55), (56), (58), then for all $L \in \Delta \setminus \{-e^{t_1 \cdot s_1}\}$,

$$\delta_2 \cdot \left| \frac{1}{e^{t_1 \cdot s_1} + L} \right| \leqslant (h-1) \cdot \delta_1 \cdot \delta_3. \tag{59}$$

As $L \to -e^{t_1 \cdot s_1}$, the LHS of Equation (59) goes to $\infty$, but the RHS is a constant, which is a contradiction. So with the choice of $t_1, \ldots, t_h$, there exists $L \in \mathbb{R}$ such that $\det(A) \neq 0$. The result holds for $h$. By mathematical induction, it holds for every positive integers $h$. The proof is done. □

*Remark* 9. If we fix a positive integer $L$, there might not exist $t_1, \ldots, t_h$ satisfy the condition. For instance, if $h \geqslant 4$, and $s_1 + s_2 = s_3 + s_4 = 0$, then the matrix $A$ is not full rank for all $t_1, \ldots, t_h$.

We have two direct corollaries of Theorem A.1.

**Corollary A.5.** *Let $D$ be a positive integer. Assume that, for two positive integers $h, h'$, collections $\{A_i\}_{i=1}^h$, $\{A_i'\}_{i=1}^{h'}$, $\{B_i\}_{i=1}^h$, $\{B_i'\}_{i=1}^{h'}$ of matrices in $\mathbb{R}^{D \times D}$, we have*

$$F\left(X; \{A_i, B_i\}_{i=1}^h\right) = F\left(X; \{A_i', B_i'\}_{i=1}^{h'}\right) \tag{60}$$

*for all positive integers $L$ and $X \in \mathbb{R}^{L \times D}$. Then, for all $A \in \mathbb{R}^{D \times D}$, we have*

$$\sum_{i \,:\, A_i = A} B_i = \sum_{i \,:\, A_i' = A} B_i'. \tag{61}$$

*Proof.* From Equation (60), we have

$$\sum_{A \in \mathbb{R}^{D \times D}} f(XAX^\top) \cdot X \cdot \left( \sum_{i \,:\, A_i = A} B_i - \sum_{i \,:\, A_i' = A} B_i' \right) = 0, \tag{62}$$

then the result is directly from Corollary A.5. □

**Corollary A.6.** *Let $D, D_k, D_v$ and $h, h'$ be positive integers. For $1 \leqslant i \leqslant h$ and $1 \leqslant j \leqslant h'$, let*

$$\left( W^{(Q,i)}, W^{(K,i)}, W^{(V,i)}, W^{(O,i)} \right) \tag{63}$$

$$\left( \overline{W}^{(Q,j)}, \overline{W}^{(K,j)}, \overline{W}^{(V,j)}, \overline{W}^{(O,j)} \right) \tag{64}$$

*be elements of $\mathbb{R}^{D \times D_k} \times \mathbb{R}^{D \times D_k} \times \mathbb{R}^{D \times D_v} \times \mathbb{R}^{D_v \times D}$. Assume that, the two corresponding Multi-Head's are identical, i.e.*

$$\text{MultiHead}\left( X; \left\{ W^{(Q,i)}, W^{(K,i)}, W^{(V,i)}, W^{(O,i)} \right\}_{i=1}^h \right)$$

$$= \text{MultiHead}\left( X; \left\{ \left( \overline{W}^{(Q,j)}, \overline{W}^{(K,j)}, \overline{W}^{(V,j)}, \overline{W}^{(O,j)} \right) \right\}_{j=1}^{h'} \right). \tag{65}$$

*for all positive integers $L$ and $X \in \mathbb{R}^{L \times D}$. Then, for all $A \in \mathbb{R}^{D \times D}$, we have*

$$\sum_{i \,:\, W^{(Q,i)} \cdot \left( W^{(K,i)} \right)^\top = A} W^{(V,i)} \cdot W^{(O,i)} = \sum_{j \,:\, \overline{W}^{(Q,j)} \cdot \left( \overline{W}^{(K,j)} \right)^\top = A} \overline{W}^{(V,j)} \cdot \overline{W}^{(O,j)}. \tag{66}$$

We characterize the symmetries of the weights of MultiHead in the following theorem.

**Theorem A.7.** *Let $h, D, D_k, D_v$ be positive integers. Let $\left(W^{(Q,i)}, W^{(K,i)}, W^{(V,i)}, W^{(O,i)}\right)$ and $\left(\overline{W}^{(Q,i)}, \overline{W}^{(K,i)}, \overline{W}^{(V,i)}, \overline{W}^{(O,i)}\right)$ be arbitrary elements of $\mathbb{R}^{D \times D_k} \times \mathbb{R}^{D \times D_k} \times \mathbb{R}^{D \times D_v} \times \mathbb{R}^{D_v \times D}$ with $i = 1, \ldots, h$. Assume that*

*(a) $\max(D_k, D_v) \leqslant D$,*

*(b) the matrices $W^{(Q,i)} \cdot \left(W^{(K,i)}\right)^\top$, $\overline{W}^{(Q,i)} \cdot \left(\overline{W}^{(K,i)}\right)^\top$, $W^{(V,i)} \cdot W^{(O,i)}$, and $\overline{W}^{(V,i)} \cdot \overline{W}^{(O,i)}$ are of full rank,*

*(c) the matrices $W^{(Q,i)} \cdot \left(W^{(K,i)}\right)^\top$ with $i = 1, \ldots, h$ are pairwise distinct,*

*(d) the matrices $\overline{W}^{(Q,i)} \cdot \left(\overline{W}^{(K,i)}\right)^\top$ with $i = 1, \ldots, h$ are pairwise distinct.*

*Then the following are equivalent:*

*1. For every positive integer $L$ and every $X \in \mathbb{R}^{L \times D}$, we always have*

$$\text{MultiHead}\left(X; \left\{W^{(Q,i)}, W^{(K,i)}, W^{(V,i)}, W^{(O,i)}\right\}_{i=1}^h\right)$$

$$= \text{MultiHead}\left(X; \left\{\overline{W}^{(Q,i)}, \overline{W}^{(K,i)}, \overline{W}^{(V,i)}, \overline{W}^{(O,i)}\right\}_{i=1}^h\right).$$

*2. There exist matrices $M^{(i)} \in \text{GL}_{D_k}(\mathbb{R})$ and $N^{(i)} \in \text{GL}_{D_v}(\mathbb{R})$ for each $i = 1, \ldots, h$, as well as a permutation $\tau \in \mathcal{S}_h$, such that*

$$\left(\overline{W}^{(Q,\tau(i))}, \overline{W}^{(K,\tau(i))}, \overline{W}^{(V,i)}, \overline{W}^{(O,\tau(i))}\right)$$

$$= \left(W^{(Q,i)} \cdot (M^{(i)})^\top, W^{(K,i)} \cdot (M^{(i)})^{-1}, W^{(V,i)} \cdot N^{(i)}, (N^{(i)})^{-1} \cdot W^{(O,i)}\right).$$

*Proof.* The implication $(2.) \Rightarrow (1.)$ is clear. Let us consider the implication $(1.) \Rightarrow (2.)$. For each $s = 1, \ldots, h$, we set

$$A^{(s)} = W^{(Q,s)} \cdot \left(W^{(K,s)}\right)^\top, \qquad \overline{A}^{(s)} = \overline{W}^{(Q,s)} \cdot \left(\overline{W}^{(K,s)}\right)^\top,$$

$$B^{(s)} = W^{(V,s)} \cdot W^{(O,s)}, \qquad \overline{B}^{(s)} = \overline{W}^{(V,s)} \cdot \overline{W}^{(O,s)},$$

which are matrices in $\mathbb{R}^{D \times D}$. By applying Corollary A.6 for $A = A^{(s)}$, we see that

$$B^{(s)} = \sum_{j \,:\, \overline{A}^{(j)} = A^{(s)}} \overline{B}^{(j)}. \tag{67}$$

However, since the matrices $\overline{A}^{(j)}$ with $j = 1, \ldots, h$ are pairwise distinct, there exist a unique index $j_s \in \{1, \ldots, h\}$ such that $\overline{A}^{(j_s)} = A^{(s)}$. The correspondence $s \mapsto j_s$ is a permutation of $\{1, \ldots, h\}$. Therefore, we can write $j_s = \tau(s)$ for some $\tau \in \mathcal{S}_h$. Hence, $\overline{A}^{(\tau(s))} = A^{(s)}$, and thus equation 67 becomes

$$\overline{B}^{(\tau(s))} = B^{(s)}.$$

The theorem is then followed by the rank factorization of the matrices $A^{(s)}$ and $B^{(s)}$ (Piziak & Odell, 1999). □

## B  MATRIX GROUP PRESERVED BY LayerNorm

In our setting, layer normalization is a row-wise operator. In particular, for a row vector $x = (x_1, \ldots, x_D) \in \mathbb{R}^D$, the standard layer normalization of $x$, denoted by $\text{LayerNorm}(x)$, is determined as

$$\text{LayerNorm}(x) = \sqrt{D} \cdot \frac{x - \bar{x} \cdot \mathbf{1}_D}{||x - \bar{x} \cdot \mathbf{1}_D||_2}, \tag{68}$$

where $\bar{x} = \frac{1}{D}(x_1 + \ldots + x_D)$ is the mean of the coordinates of $x$ and $\mathbf{1}_D$ is a row vector in $\mathbb{R}^D$ whose coordinates are all equal to 1. Geometrically, we can view LayerNorm as the composition

$$\text{LayerNorm} = \tau \circ \rho, \tag{69}$$

of two transformations $\tau$ and $\rho$ on $\mathbb{R}^D$ defined belows.

- The perpendicular projection map $\rho\colon \mathbb{R}^D \to \langle \mathbf{1}_D \rangle^{\perp}$ which maps $x \mapsto x - \bar{x} \cdot \mathbf{1}_D$ for each row vector $x \in \mathbb{R}^D$. Here, the hyperplane $\langle \mathbf{1}_D \rangle^{\perp}$ is the orthogonal complement of $\langle \mathbf{1}_D \rangle$ in $\mathbb{R}^D$ with respect to the standard dot product. It is noted that $\rho$ is a linear map with the kernel $\text{Ker}(\rho) = \langle \mathbf{1}_D \rangle$, which is the one-dimensional vector subspace of $\mathbb{R}^D$ generated by the row vector $\mathbf{1}_D$.
- The scaling map $\tau\colon \mathbb{R}^D \setminus \{0\} \to \mathbb{R}^D$ which maps $x \mapsto \sqrt{D}\frac{x}{||x||_2}$. The image of $\tau$ is the sphere centered at the origin of radius $\sqrt{D}$.

It is noted that, the LayerNorm operator is not defined on the whole $\mathbb{R}^D$. Indeed, since $\tau$ is not defined at the origin, it is necesarily that $\rho(x) \neq 0$. This means that $x$ should not lie in the kernel of $\rho$. Therefore, the LayerNorm operator actually defines a map from the set $\mathbb{R}^D \setminus \langle \mathbf{1}_D \rangle$ to $\mathbb{R}^D$.

Under the view of the LayerNorm as a composition above, it is natural to ask which matrix (or linear map represented by this matrix) will commute with both the projection $\rho$ and the scaling $\tau$. The following theorem gives a complete answer to this question.

**Theorem B.1.** *Let $D$ be a positive integer. Let $\rho$ and $\tau$ be the projection and scaling map defined above. The following are equivalent for an arbitrary matrix $M \in \text{GL}_D(\mathbb{R})$:*

1. *$M$ commutes with $\rho$ and $\tau$, i.e. $\rho(xM) = \rho(x)M$ and $\tau(xM) = \tau(x)M$ for all row vector $x \in \mathbb{R}^D$.*

2. *$M$ is an orthogonal matrix such that its row sums and column sums are all equal.*

*Proof.* One can direct verify (2.) implies (1.). To show (1.) implies (2.), let $x = (x_1, \ldots, x_D) \in \mathbb{R}^D$ and $M = (m_{ij})_{1 \leqslant i,j \leqslant D} \in \mathbb{R}^{D \times D}$. From $\rho(xM) = \rho(x)M$, we have

$$xM - \overline{xM} \cdot \mathbf{1}_D = (x - \bar{x} \cdot \mathbf{1}_D)M, \tag{70}$$

which means

$$\overline{xM} \cdot \mathbf{1}_D = \bar{x} \cdot \mathbf{1}_D \cdot M. \tag{71}$$

So

$$\left( \sum_{i=1}^{D} \left( x_i \cdot \left( \sum_{j=1}^{D} m_{ij} \right) \right) \right) \cdot \mathbf{1}_D = \left( \sum_{i=1}^{D} x_i \right) \cdot \left( \sum_{i=1}^{D} m_{i1}, \sum_{i=1}^{D} m_{i2}, \ldots, \sum_{i=1}^{D} m_{iD} \right). \tag{72}$$

It implies that

$$\sum_{i=1}^{D} x_i \cdot \left( \sum_{j=1}^{D} m_{ij} \right) = \left( \sum_{i=1}^{D} x_i \right) \cdot \left( \sum_{i=1}^{D} m_{i1} \right) = \ldots = \left( \sum_{i=1}^{D} x_i \right) \cdot \left( \sum_{i=1}^{D} m_{iD} \right). \tag{73}$$

The above equation holds for all feasible $x$ (as $\rho$ and $\tau$ are not defined on the whole $\mathbb{R}^d$). But since the set of feasible $x$ is dense in $\mathbb{R}^d$, by continuity, the equation holds for all $x \in \mathbb{R}^d$. It implies row sums and column sums of $M$ are all equal.

In addition, since $\tau(xM) = \tau(x)M$, we have $\|xM\|_2 = \|x\|_2$. By the same argument, it holds for all $x \in \mathbb{R}^d$. Hence, $M$ is orthogonal. $\square$

*Remark* 10 (generalized doubly stochastic matrix). A matrix whose row sums and column sums are all equal to one is called a *generalized doubly stochastic matrix*. Smoktunowicz et al. (2019) characterized all orthogonal generalized doubly stochastic matrices. In particular, let $\text{O}(D)$ is the set of $D \times D$ orthogonal matrices, and $\mathcal{U}_D$ is the subset of $\text{O}(D)$ consists of all orthogonal matrices with the first column is $(1/\sqrt{D}) \cdot (\mathbf{1}_D)^{\top}$, then every orthogonal generalized doubly stochastic matrix can be writen in the form

$$\begin{pmatrix} 1 & 0^{\top} \\ 0 & X \end{pmatrix} \tag{74}$$

for some $(D-1) \times (D-1)$ orthogonal matrix $X$.

There are invertible matrices that do not permute $\phi$ and $\tau$, but still have nice behavior with LayerNorm, as we see in the following theorem.

**Theorem B.2.** *Let $D$ be a positive integer. Then for arbitrary permutation matrix $P \in \mathcal{P}_D$ and real number $\lambda \neq 0$, we have*

$$\mathrm{LayerNorm}(\lambda x P) = \mathrm{sign}(\lambda) \, \mathrm{LayerNorm}(x) P, \tag{75}$$

*for every row vector $x \in \mathbb{R}^D$.*

*Proof.* We have

$$\mathrm{LayerNorm}(\lambda x P) = \tau \circ \rho(\lambda x P) \tag{76}$$

$$= \tau(\lambda x P - \overline{\lambda x P} \cdot \mathbf{1}_D) \tag{77}$$

$$= \tau(\lambda x P - \lambda \overline{x} \cdot \mathbf{1}_D \cdot P) \qquad = \mathrm{sign}(\lambda) \tau(x - \overline{x} \cdot \mathbf{1}_D) P \tag{78}$$

$$= \mathrm{sign}(\lambda) \, \mathrm{LayerNorm}(x) P \tag{79}$$

We finish the proof. $\qquad\square$

## C   WEIGHT SPACE AND GROUP ACTION ON WEIGHT SPACE

In this section, we recall the weight space of a transformer block and the group action on it. Then, we prove Theorem C.5 (which corresponds to Theorem 4.3 in the main text). Group actions on models are extensively studied in Machine Learning Tran et al. (2025a;c;b; 2024c;b).

### C.1   WEIGHT SPACE

Recall that a standard transformer block, which is denoted by $\mathrm{Attn}$, is defined as follows: for each $X \in \mathbb{R}^{L \times D}$, we have

$$\mathrm{Attn}(X) = \mathrm{LayerNorm}\left(\mathrm{ReLU}(\hat{X} \cdot W^{(A)} + \mathbf{1}_L \cdot b^{(A)}) \cdot W^{(B)} + \mathbf{1}_L \cdot b^{(B)}\right), \tag{80}$$

where

$$\hat{X} = \mathrm{LayerNorm}\left(\mathrm{MultiHead}(X; \{W^{(Q,i)}, W^{(K,i)}, W^{(V,i)}, W^{(O,i)}\}_{i=1}^h)\right). \tag{81}$$

with $\mathbf{1}_L = [1, \ldots, 1]^\top \in \mathbb{R}^{L \times 1}$.

The weight space $\mathcal{U}$ of the above transformer block is the vector space:

$$\mathcal{U} = \left(\mathbb{R}^{D \times D_k} \times \mathbb{R}^{D \times D_k} \times \mathbb{R}^{D \times D_v} \times \mathbb{R}^{D_v \times D}\right)^h \times \left(\mathbb{R}^{D \times D_A} \times \mathbb{R}^{D_A \times D}\right) \times \left(\mathbb{R}^{1 \times D_A} \times \mathbb{R}^{1 \times D}\right). \tag{82}$$

An element $U \in \mathcal{U}$ is of the form

$$U = \left(\left([W]^{(Q,i)}, [W]^{(K,i)}, [W]^{(V,i)}, [W]^{(O,i)}\right)_{i=1,\ldots,h}, \left([W]^{(A)}, [W]^{(B)}\right), \left([b]^{(A)}, [b]^{(B)}\right)\right). \tag{83}$$

To emphasize the weights of $\mathrm{Attn}$, we will write $\mathrm{Attn}(X; U)$ instead of $\mathrm{Attn}(X)$.

### C.2   GROUP ACTION ON WEIGHT SPACE

Consider the weight space $\mathcal{U}$ defined in Equation (82) and Equation (83), set

$$\mathcal{G}_{\mathcal{U}} = \mathrm{S}_h \times \mathrm{GL}_{D_k}(\mathbb{R})^h \times \mathrm{GL}_{D_v}(\mathbb{R})^h \times \mathcal{P}_D \times \mathcal{P}_{D_A}. \tag{84}$$

Each element $g \in \mathcal{G}_{\mathcal{U}}$ has the form

$$g = \left(\tau, (M_i)_{i=1,\ldots,h}, (N_i)_{i=1,\ldots,h}, P_{\pi_O}, P_{\pi_A}\right), \tag{85}$$

where $\tau, \pi_O, \pi_A$ are permutations and $M_i, N_i$ are invertible matrices of appropriate sizes.

**Definition C.1** (Group action on $\mathcal{G}_{\mathcal{U}}$). The action of $\mathcal{G}_{\mathcal{U}}$ on $\mathcal{U}$ is defined to be a map $\mathcal{G}_{\mathcal{U}} \times \mathcal{U} \to \mathcal{U}$ which maps an element $(g, U) \in \mathcal{G}_{\mathcal{U}} \times \mathcal{U}$ with $U$ and $g$ given in Equation (83) and Equation (85) to the element

$$gU = \left(\left([gW]^{(Q,i)}, [gW]^{(K,i)}, [gW]^{(V,i)}, [gW]^{(O,i)}\right)_{i=1,\ldots,h}, \tag{86}\right.$$

$$\left.\left([gW]^{(A)}, [gW]^{(B)}\right), \left([gb]^{(A)}, [gb]^{(B)}\right)\right), \tag{87}$$

where

$$[gW]^{(Q,i)} = [W]^{(Q,\tau(i))} \cdot \left( M^{(\tau(i))} \right)^{\top},$$

$$[gW]^{(K,i)} = [W]^{(K,\tau(i))} \cdot \left( M^{(\tau(i))} \right)^{-1},$$

$$[gW]^{(V,i)} = [W]^{(V,\tau(i))} \cdot N^{(\tau(i))},$$

$$[gW]^{(O,i)} = \left( N^{(\tau(i))} \right)^{-1} \cdot [W]^{(O,\tau(i))} \cdot P_{\pi_O},$$

$$[gW]^{(A)} = P_{\pi_O}^{-1} \cdot [W]^{(A)} \cdot P_{\pi_A},$$

$$[gW]^{(B)} = P_{\pi_A}^{-1} \cdot [W]^{(B)},$$

$$[gb]^{(A)} = [b]^{(A)} \cdot P_{\pi_A},$$

$$[gb]^{(B)} = [b]^{(B)}.$$

In addition to the above definition, we will also need the following ones for the construction of the equivariant and invariant maps later.

**Definition C.2.** With notations as above, for each $i = 1, \ldots, h$, we denote:

$$[WW]^{(QK,i)} := [W]^{(Q,i)} \cdot \left( [W]^{(K,i)} \right)^{\top}, \quad \text{and} \quad [WW]^{(VO,i)} := [W]^{(V,i)} \cdot [W]^{(O,i)}. \tag{88}$$

*Remark* 11. The terms $[WW]^{(QK,i)}$ and $[WW]^{(VO,i)}$ are equivariant under the action of $g$, since

$$[gWgW]^{(QK,i)} := \left( [gW]^{(Q,i)} \cdot [gW]^{(K,i)} \right)^{\top}$$

$$= \left( [W]^{(Q,\tau(i))} \cdot (M^{\tau(i)})^{\top} \right) \cdot \left( [W]^{(K,\tau(i))} \cdot (M^{\tau(i)})^{-1} \right)^{\top}$$

$$= [W]^{(Q,\tau(i))} \cdot \left( [W]^{(K,\tau(i))} \right)^{\top}$$

$$= [WW]^{(QK,\tau(i))},$$

and

$$[gWgW]^{(VO,i)} := [gW]^{(V,i)} \cdot [gW]^{(O,i)}$$

$$= \left( [W]^{(V,\tau(i))} \cdot M^{(\tau(i))} \right) \cdot \left( (M^{(\tau(i))})^{-1} \cdot [W]^{(O,\tau(i))} \cdot P_{\pi_O} \right)$$

$$= [W]^{(V,\tau(i))} \cdot [W]^{(O,\tau(i))} \cdot P_{\pi_O}$$

$$= [WW]^{(VO,\tau(i))} \cdot P_{\pi_O}.$$

Therefore, we can intuitively say that these terms are compatible under the action of $\mathcal{G}_{\mathcal{U}}$.

**Proposition C.3.** *With notations as above, we have*

$$[gWgW]_{j,k}^{(QK,i)} = [WW]_{j,k}^{(QK,\tau(i))},$$

$$[gWgW]_{j,k}^{(VO,i)} = [WW]_{j,\pi_O(k)}^{(VO,\tau(i))},$$

$$[gW]_{j,k}^{(Q,i)} = \left[ [W]^{(Q,\tau(i))} \cdot \left( M^{(\tau(i))} \right)^{\top} \right]_{j,k},$$

$$[gW]_{j,k}^{(K,i)} = \left[ [W]^{(K,\tau(i))} \cdot \left( M^{(\tau(i))} \right)^{-1} \right]_{j,k},$$

$$[gW]_{j,k}^{(V,i)} = \left[ [W]^{(V,\tau(i))} \cdot N^{(\tau(i))} \right]_{j,k},$$

$$[gW]^{(O,i)} = \left[ \left( N^{(\tau(i))} \right)^{-1} \cdot [W]^{(O,\tau(i))} \right]_{j,\pi_O(k)},$$

$$[gW]_{j,k}^{(A)} = [W]_{\pi_O(j),\pi_A(k)}^{(A)},$$

$$[gW]_{j,k}^{(B)} = [W]_{\pi_A(j),k}^{(B)},$$

$$[gb]_k^{(A)} = [b]_{\pi_A(k)}^{(A)},$$
$$[gb]_k^{(B)} = [b]_k^{(B)}.$$

*for all appropriate indices $j$ and $k$.*

*Proof.* This proposition follows immediately from the definition of the group action on $\mathcal{U}$. $\qquad\square$

**Lemma C.4.** *Let $n$ be a positive integer. The following are equivalent for a matrix $M \in \mathrm{GL}_n(\mathbb{R})$:*

1. $\mathrm{ReLU}(x \cdot M) = \mathrm{ReLU}(x) \cdot M$ *for all row vector $x \in \mathbb{R}^n$.*

2. $M$ *is a monomial matrix whose nonzero entries are positive numbers.*

*Proof.* See (Godfrey et al., 2022). $\qquad\square$

**Theorem C.5** (Invariance of $\mathrm{Attn}$ under the action of $\mathcal{G}_{\mathcal{U}}$)**.** *With notations as above, we have*
$$\mathrm{Attn}(X; gU) = \mathrm{Attn}(X; U) \tag{89}$$
*for all $U \in \mathcal{U}$, $g \in \mathcal{G}$, and $X \in \mathbb{R}^{L \times D}$.*

*Proof.* According to Equation (80) and Equation (81), we have

$$\mathrm{Attn}(X; gU) \tag{90}$$

$$= \mathrm{LayerNorm}\left(\mathrm{ReLU}(\hat{X} \cdot [gW]^{(A)} + \mathbf{1}_L \cdot [gb]^{(A)}) \cdot [gW]^{(B)} + \mathbf{1}_L \cdot [gb]^{(B)}\right) \tag{91}$$

$$= \mathrm{LayerNorm}\left(\mathrm{ReLU}(\hat{X} \cdot P_{\pi_O}^{-1} \cdot [W]^{(A)} P_{\pi_A} + \mathbf{1}_L \cdot [b]^{(A)} P_{\pi_A}) \cdot P_{\pi_A}^{-1} \cdot [W]^{(B)} + \mathbf{1}_L \cdot [b]^{(B)}\right) \tag{92}$$

$$= \mathrm{LayerNorm}\left(\mathrm{ReLU}(\hat{X} P_{\pi_O}^{-1} [W]^{(A)} + \mathbf{1}_L [b]^{(A)}) P_{\pi_A} P_{\pi_A}^{-1} [W]^{(B)} + \mathbf{1}_L [b]^{(B)}\right) \tag{93}$$

$$= \mathrm{LayerNorm}\left(\mathrm{ReLU}(\hat{X} P_{\pi_O}^{-1} [W]^{(A)} + \mathbf{1}_L [b]^{(A)}) [W]^{(B)} + \mathbf{1}_L [b]^{(B)}\right). \tag{94}$$

In the above equalities, Equation (92) following from the definition of the group action of $\mathcal{G}_{\mathcal{U}}$ on $\mathcal{U}$, while Equation (93) follows from Lemma C.4. In addition, we proceed the term $\hat{X} P_{\pi_O}^{-1}$ inside Equation (94) as:

$$\hat{X} P_{\pi_O}^{-1} \tag{95}$$

$$= \mathrm{LayerNorm}\left(\mathrm{MultiHead}(X; \{[gW]^{(Q,i)}, [gW]^{(K,i)}, [gW]^{(V,i)}, [gW]^{(O,i)}\}_{i=1}^h)\right) \cdot P_{\pi_O}^{-1} \tag{96}$$

$$= \mathrm{LayerNorm}\left(\sum_{i=1}^h \mathrm{Head}(X; \{[gW]^{(Q,i)}, [gW]^{(K,i)}, [gW]^{(V,i)}\}_{i=1}^h)[gW]^{(O,i)}\right) \cdot P_{\pi_O}^{-1} \tag{97}$$

$$= \mathrm{LayerNorm}\left(\sum_{i=1}^h \mathrm{Head}\left(X; \{\overline{[W]}^{(Q,\tau(i))}, \overline{[W]}^{(K,\tau(i))}, \overline{[W]}^{(V,\tau(i))}\}_{i=1}^h\right) \overline{[W]}^{(O,\tau(i))}\right) \tag{98}$$

$$= \mathrm{LayerNorm}\left(\sum_{i=1}^h \mathrm{Head}\left(X; \{[W]^{(Q,i)}, [W]^{(K,i)}, [W]^{(V,i)}\}_{i=1}^h\right) [W]^{(O,i)}\right) \tag{99}$$

$$= \hat{X}. \tag{100}$$

In the above equalities, we used the notations

$$\overline{[W]}^{(Q,\tau(i))} = [W]^{(Q,\tau(i))} \cdot (M^{(\tau(i))})^\top,$$
$$\overline{[W]}^{(K,\tau(i))} = [W]^{(K,\tau(i))} \cdot (M^{(Q,\tau(i))})^{-1},$$
$$\overline{[W]}^{(V,\tau(i))} = [W]^{(V,\tau(i))} \cdot M^{(V,\tau(i))},$$
$$\overline{[W]}^{(O,\tau(i))} = (M^{(V,\tau(i))})^{-1} \cdot [W]^{(O,\tau(i))}.$$

In the above equalities, Equation (98) follows from Theorem A.1 and Corollary B.2. In addition, Equation (99) obtained by permuting terms in the sum by $\tau$. From Equation (94) and Equation (100), we see that
$$\mathrm{Attn}(X; gU) = \mathrm{Attn}(X; U),$$
for all $X, g$ and $U$. The theorem is then proved. $\qquad\square$

# D  AN EQUIVARIANT POLYNOMIAL LAYER

We now proceed to construct a $\mathcal{G}_\mathcal{U}$-equivariant polynomial layer, denoted as $E$. These layers serve as the fundamental building blocks for our Transformer-NFNs. Our method is based on parameter sharing mechanism.

## D.1  A GENERAL FORM WITH UNKNOWN COEFFICIENTS

We want to build a map $E \colon \mathcal{U} \to \mathcal{U}$ such that $E$ is $\mathcal{G}_\mathcal{U}$-equivariant. We select $E(U)$ to be a linear combination of entries of the matrices $[W]^{(Q,s)}$, $[W]^{(K,s)}$, $[W]^{(V,s)}$, $[W]^{(O,s)}$, $[W]^{(A)}$, $[W]^{(B)}$, $[b]^{(A)}$, $[b]^{(B)}$ as well as the matrices $[WW]^{(QK,s)}$ and $[WW]^{(VO,s)}$, i.e

$$E(U) = \left( \left( \left[ [E(W)]^{(Q,i)}, [E(W)]^{(K,i)}, [E(W)]^{(V,i)}, [E(W)]^{(O,i)} \right)_{i=1,\dots,h}, \right.\right.$$
$$\left.\left. \left( [E(W)]^{(A)}, [E(W)]^{(B)} \right), \left( [E(b)]^{(A)}, [E(b)]^{(B)} \right) \right), \quad (101)$$

where

$$[E(W)]_{j,k}^{(Q,i)} = \sum_{s=1}^{h}\sum_{p=1}^{D}\sum_{q=1}^{D} \Phi_{(QK,s):p,q}^{(Q,i):j,k}[WW]_{p,q}^{(QK,s)} + \sum_{s=1}^{h}\sum_{p=1}^{D}\sum_{q=1}^{D} \Phi_{(VO,s):p,q}^{(Q,i):j,k}[WW]_{p,q}^{(VO,s)}$$
$$+ \sum_{s=1}^{h}\sum_{p=1}^{D}\sum_{q=1}^{D_k} \Phi_{(Q,s):p,q}^{(Q,i):j,k}[W]_{p,q}^{(Q,s)} + \sum_{s=1}^{h}\sum_{p=1}^{D}\sum_{q=1}^{D_k} \Phi_{(K,s):p,q}^{(Q,i):j,k}[W]_{p,q}^{(K,s)}$$
$$+ \sum_{s=1}^{h}\sum_{p=1}^{D}\sum_{q=1}^{D_v} \Phi_{(V,s):p,q}^{(Q,i):j,k}[W]_{p,q}^{(V,s)} + \sum_{s=1}^{h}\sum_{p=1}^{D_v}\sum_{q=1}^{D} \Phi_{(O,s):p,q}^{(Q,i):j,k}[W]_{p,q}^{(O,s)}$$
$$+ \sum_{p=1}^{D}\sum_{q=1}^{D_A} \Phi_{(A):p,q}^{(Q,i):j,k}[W]_{p,q}^{(A)} + \sum_{p=1}^{D_A}\sum_{q=1}^{D} \Phi_{(B):p,q}^{(Q,i):j,k}[W]_{p,q}^{(B)}$$
$$+ \sum_{q=1}^{D_A} \Phi_{(A):q}^{(Q,i):j,k}[b]_{q}^{(A)} + \sum_{q=1}^{D} \Phi_{(B):q}^{(Q,i):j,k}[b]_{q}^{(B)} + \Phi^{(Q,i):j,k}, \quad (102)$$

$$[E(W)]_{j,k}^{(K,i)} = \sum_{s=1}^{h}\sum_{p=1}^{D}\sum_{q=1}^{D} \Phi_{(QK,s):p,q}^{(K,i):j,k}[WW]_{p,q}^{(QK,s)} + \sum_{s=1}^{h}\sum_{p=1}^{D}\sum_{q=1}^{D} \Phi_{(VO,s):p,q}^{(K,i):j,k}[WW]_{p,q}^{(VO,s)}$$
$$+ \sum_{s=1}^{h}\sum_{p=1}^{D}\sum_{q=1}^{D_k} \Phi_{(Q,s):p,q}^{(K,i):j,k}[W]_{p,q}^{(Q,s)} + \sum_{s=1}^{h}\sum_{p=1}^{D}\sum_{q=1}^{D_k} \Phi_{(K,s):p,q}^{(K,i):j,k}[W]_{p,q}^{(K,s)}$$
$$+ \sum_{s=1}^{h}\sum_{p=1}^{D}\sum_{q=1}^{D_v} \Phi_{(V,s):p,q}^{(K,i):j,k}[W]_{p,q}^{(V,s)} + \sum_{s=1}^{h}\sum_{p=1}^{D_v}\sum_{q=1}^{D} \Phi_{(O,s):p,q}^{(K,i):j,k}[W]_{p,q}^{(O,s)}$$
$$+ \sum_{p=1}^{D}\sum_{q=1}^{D_A} \phi_{(A):p,q}^{(K,i):j,k}[W]_{p,q}^{(A)} + \sum_{p=1}^{D_A}\sum_{q=1}^{D} \Phi_{(B):p,q}^{(K,i):j,k}[W]_{p,q}^{(B)}$$
$$+ \sum_{q=1}^{D_A} \Phi_{(A):q}^{(K,i):j,k}[b]_{q}^{(A)} + \sum_{q=1}^{D} \Phi_{(B):q}^{(K,i):j,k}[b]_{q}^{(B)} + \Phi^{(K,i):j,k}, \quad (103)$$

$$[E(W)]_{j,k}^{(V,i)} = \sum_{s=1}^{h}\sum_{p=1}^{D}\sum_{q=1}^{D} \Phi_{(QK,s):p,q}^{(V,i):j,k}[WW]_{p,q}^{(QK,s)} + \sum_{s=1}^{h}\sum_{p=1}^{D}\sum_{q=1}^{D} \Phi_{(VO,s):p,q}^{(V,i):j,k}[WW]_{p,q}^{(VO,s)}$$
$$+ \sum_{s=1}^{h}\sum_{p=1}^{D}\sum_{q=1}^{D_k} \Phi_{(Q,s):p,q}^{(V,i):j,k}[W]_{p,q}^{(Q,s)} + \sum_{s=1}^{h}\sum_{p=1}^{D}\sum_{q=1}^{D_k} \Phi_{(K,s):p,q}^{(V,i):j,k}[W]_{p,q}^{(K,s)}$$
$$+ \sum_{s=1}^{h}\sum_{p=1}^{D}\sum_{q=1}^{D_v} \Phi_{(V,s):p,q}^{(V,i):j,k}[W]_{p,q}^{(V,s)} + \sum_{s=1}^{h}\sum_{p=1}^{D_v}\sum_{q=1}^{D} \Phi_{(O,s):p,q}^{(V,i):j,k}[W]_{p,q}^{(O,s)}$$

$$+ \sum_{p=1}^{D} \sum_{q=1}^{D_A} \Phi_{(A):p,q}^{(V,i):j,k} [W]_{p,q}^{(A)} + \sum_{p=1}^{D_A} \sum_{q=1}^{D} \Phi_{(B):p,q}^{(V,i):j,k} [W]_{p,q}^{(B)}$$

$$+ \sum_{q=1}^{D_A} \Phi_{(A):q}^{(V,i):j,k} [b]_q^{(A)} + \sum_{q=1}^{D} \Phi_{(B):q}^{(V,i):j,k} [b]_q^{(B)} + \Phi^{(V,i):j,k}, \tag{104}$$

$$[E(W)]_{j,k}^{(O,i)} = \sum_{s=1}^{h} \sum_{p=1}^{D} \sum_{q=1}^{D} \Phi_{(QK,s):p,q}^{(O,i):j,k} [WW]_{p,q}^{(QK,s)} + \sum_{s=1}^{h} \sum_{p=1}^{D} \sum_{q=1}^{D} \Phi_{(VO,s):p,q}^{(O,i):j,k} [WW]_{p,q}^{(VO,s)}$$

$$+ \sum_{s=1}^{h} \sum_{p=1}^{D} \sum_{q=1}^{D_k} \Phi_{(Q,s):p,q}^{(O,i):j,k} [W]_{p,q}^{(Q,s)} + \sum_{s=1}^{h} \sum_{p=1}^{D} \sum_{q=1}^{D_k} \Phi_{(K,s):p,q}^{(O,i):j,k} [W]_{p,q}^{(K,s)}$$

$$+ \sum_{s=1}^{h} \sum_{p=1}^{D} \sum_{q=1}^{D_v} \Phi_{(V,s):p,q}^{(O,i):j,k} [W]_{p,q}^{(V,s)} + \sum_{s=1}^{h} \sum_{p=1}^{D_v} \sum_{q=1}^{D} \Phi_{(O,s):p,q}^{(O,i):j,k} [W]_{p,q}^{(O,s)}$$

$$+ \sum_{p=1}^{D} \sum_{q=1}^{D_A} \Phi_{(A):p,q}^{(O,i):j,k} [W]_{p,q}^{(A)} + \sum_{p=1}^{D_A} \sum_{q=1}^{D} \Phi_{(B):p,q}^{(O,i):j,k} [W]_{p,q}^{(B)}$$

$$+ \sum_{q=1}^{D_A} \Phi_{(A):q}^{(O,i):j,k} [b]_q^{(A)} + \sum_{q=1}^{D} \Phi_{(B):q}^{(O,i):j,k} [b]_q^{(B)} + \Phi^{(O,i):j,k}, \tag{105}$$

$$[E(W)]_{j,k}^{(A)} = \sum_{s=1}^{h} \sum_{p=1}^{D} \sum_{q=1}^{D} \Phi_{(QK,s):p,q}^{(A):j,k} [WW]_{p,q}^{(QK,s)} + \sum_{s=1}^{h} \sum_{p=1}^{D} \sum_{q=1}^{D} \Phi_{(VO,s):p,q}^{(A):j,k} [WW]_{p,q}^{(VO,s)}$$

$$+ \sum_{s=1}^{h} \sum_{p=1}^{D} \sum_{q=1}^{D_k} \Phi_{(Q,s):p,q}^{(A):j,k} [W]_{p,q}^{(Q,s)} + \sum_{s=1}^{h} \sum_{p=1}^{D} \sum_{q=1}^{D_k} \Phi_{(K,s):p,q}^{(A):j,k} [W]_{p,q}^{(K,s)}$$

$$+ \sum_{s=1}^{h} \sum_{p=1}^{D} \sum_{q=1}^{D_v} \Phi_{(V,s):p,q}^{(A):j,k} [W]_{p,q}^{(V,s)} + \sum_{s=1}^{h} \sum_{p=1}^{D_v} \sum_{q=1}^{D} \Phi_{(O,s):p,q}^{(A):j,k} [W]_{p,q}^{(O,s)}$$

$$+ \sum_{p=1}^{D} \sum_{q=1}^{D_A} \Phi_{(A):p,q}^{(A):j,k} [W]_{p,q}^{(A)} + \sum_{p=1}^{D_A} \sum_{q=1}^{D} \Phi_{(B):p,q}^{(A):j,k} [W]_{p,q}^{(B)}$$

$$+ \sum_{q=1}^{D_A} \Phi_{(A):q}^{(A):j,k} [b]_q^{(A)} + \sum_{q=1}^{D} \Phi_{(B):q}^{(A):j,k} [b]_q^{(B)} + \Phi^{(A):j,k}, \tag{106}$$

$$[E(W)]_{j,k}^{(B)} = \sum_{s=1}^{h} \sum_{p=1}^{D} \sum_{q=1}^{D} \Phi_{(QK,s):p,q}^{(B):j,k} [WW]_{p,q}^{(QK,s)} + \sum_{s=1}^{h} \sum_{p=1}^{D} \sum_{q=1}^{D} \Phi_{(VO,s):p,q}^{(B):j,k} [WW]_{p,q}^{(VO,s)}$$

$$+ \sum_{s=1}^{h} \sum_{p=1}^{D} \sum_{q=1}^{D_k} \Phi_{(Q,s):p,q}^{(B):j,k} [W]_{p,q}^{(Q,s)} + \sum_{s=1}^{h} \sum_{p=1}^{D} \sum_{q=1}^{D_k} \Phi_{(K,s):p,q}^{(B):j,k} [W]_{p,q}^{(K,s)}$$

$$+ \sum_{s=1}^{h} \sum_{p=1}^{D} \sum_{q=1}^{D_v} \Phi_{(V,s):p,q}^{(B):j,k} [W]_{p,q}^{(V,s)} + \sum_{s=1}^{h} \sum_{p=1}^{D_v} \sum_{q=1}^{D} \Phi_{(O,s):p,q}^{(B):j,k} [W]_{p,q}^{(O,s)}$$

$$+ \sum_{p=1}^{D} \sum_{q=1}^{D_A} \Phi_{(A):p,q}^{(B):j,k} [W]_{p,q}^{(A)} + \sum_{p=1}^{D_A} \sum_{q=1}^{D} \Phi_{(B):p,q}^{(B):j,k} [W]_{p,q}^{(B)}$$

$$+ \sum_{q=1}^{D_A} \Phi_{(A):q}^{(B):j,k} [b]_q^{(A)} + \sum_{q=1}^{D} \Phi_{(B):q}^{(B):j,k} [b]_q^{(B)} + \Phi^{(B):j,k}, \tag{107}$$

$$[E(b)]_k^{(A)} = \sum_{s=1}^{h} \sum_{p=1}^{D} \sum_{q=1}^{D} \Phi_{(QK,s):p,q}^{(A):k} [WW]_{p,q}^{(QK,s)} + \sum_{s=1}^{h} \sum_{p=1}^{D} \sum_{q=1}^{D} \Phi_{(VO,s):p,q}^{(A):k} [WW]_{p,q}^{(VO,s)}$$

$$+ \sum_{s=1}^{h} \sum_{p=1}^{D} \sum_{q=1}^{D_k} \Phi_{(Q,s):p,q}^{(A):k} [W]_{p,q}^{(Q,s)} + \sum_{s=1}^{h} \sum_{p=1}^{D} \sum_{q=1}^{D_k} \Phi_{(K,s):p,q}^{(A):k} [W]_{p,q}^{(K,s)}$$

$$+ \sum_{s=1}^{h} \sum_{p=1}^{D} \sum_{q=1}^{D_v} \Phi_{(V,s):p,q}^{(A):k} [W]_{p,q}^{(V,s)} + \sum_{s=1}^{h} \sum_{p=1}^{D_v} \sum_{q=1}^{D} \Phi_{(O,s):p,q}^{(A):k} [W]_{p,q}^{(O,s)}$$

$$+ \sum_{p=1}^{D} \sum_{q=1}^{D_A} \phi_{(A):p,q}^{(A):k} [W]_{p,q}^{(A)} + \sum_{p=1}^{D_A} \sum_{q=1}^{D} \Phi_{(B):p,q}^{(A):k} [W]_{p,q}^{(B)}$$

$$+ \sum_{q=1}^{D_A} \Phi_{(A):q}^{(A):k} [b]_q^{(A)} + \sum_{q=1}^{D} \Phi_{(B):q}^{(A):k} [b]_q^{(B)} + \Phi^{(A):k}, \tag{108}$$

$$[E(b)]_k^{(B)} = \sum_{s=1}^{h} \sum_{p=1}^{D} \sum_{q=1}^{D} \Phi_{(QK,s):p,q}^{(B):k} [WW]_{p,q}^{(QK,s)} + \sum_{s=1}^{h} \sum_{p=1}^{D} \sum_{q=1}^{D} \Phi_{(VO,s):p,q}^{(B):k} [WW]_{p,q}^{(VO,s)}$$

$$+ \sum_{s=1}^{h} \sum_{p=1}^{D} \sum_{q=1}^{D_k} \Phi_{(Q,s):p,q}^{(B):k} [W]_{p,q}^{(Q,s)} + \sum_{s=1}^{h} \sum_{p=1}^{D} \sum_{q=1}^{D_k} \Phi_{(K,s):p,q}^{(B):k} [W]_{p,q}^{(K,s)}$$

$$+ \sum_{s=1}^{h} \sum_{p=1}^{D} \sum_{q=1}^{D_v} \Phi_{(V,s):p,q}^{(B):k} [W]_{p,q}^{(V,s)} + \sum_{s=1}^{h} \sum_{p=1}^{D_v} \sum_{q=1}^{D} \Phi_{(O,s):p,q}^{(B):k} [W]_{p,q}^{(O,s)}$$

$$+ \sum_{p=1}^{D} \sum_{q=1}^{D_A} \Phi_{(A):p,q}^{(B):k} [W]_{p,q}^{(A)} + \sum_{p=1}^{D_A} \sum_{q=1}^{D} \Phi_{(B):p,q}^{(B):k} [W]_{p,q}^{(B)}$$

$$+ \sum_{q=1}^{D_A} \Phi_{(A):q}^{(B):k} [b]_q^{(A)} + \sum_{q=1}^{D} \Phi_{(B):q}^{(B):k} [b]_q^{(B)} + \Phi^{(B):k}, \tag{109}$$

and the constants $\Phi_-^-$s are parameters.

In the following subsections, we will determine the constraints on the coefficients $\Phi_-^-$ such that $E(gU) = gE(U)$ for all $U \in \mathcal{U}$ and $g \in \mathcal{G}_{\mathcal{U}}$.

### D.2   AUXILIARY LEMMAS

We will need the following auxiliary lemmas in order to determine the constraints of the coefficients of $E$ later.

**Lemma D.1.** *Assume that $E \colon \mathcal{U} \to \mathcal{U}$ is a function defined as in Equation (101) for some coefficients $\Phi_-^-$. If $E(U) = 0$ for all $U \in \mathcal{U}$, then all coefficients are equal to zero.*

*Proof.* We view the entries of $U$ as indeterminates over $\mathbb{R}$. Under this view, since $E(U) = 0$ for all $U \in \mathcal{U}$, we can view $E(U)$ as a zero polynomial. Moreover, the set including all entries of the matrices $[W]^{(Q,s)}$, $[W]^{(K,s)}$, $[W]^{(V,s)}$, $[W]^{(O,s)}$, $[W]^{(A)}$, $[W]^{(B)}$, $[b]^{(A)}$, $[b]^{(B)}$ as well as the matrices $[WW]^{(QK,s)}$ and $[WW]^{(VO,s)}$ is a linear independent set over $\mathbb{R}$. Therefore, the coefficients of $E$, which are $\Phi_-^-$, must be equal to zero. $\square$

**Lemma D.2.** *Let $h$ and $D$ be positive integers. Let $f_s^{(1)}$, $f_s^{(2)} \colon \mathbb{R}^{D \times D} \to \mathbb{R}$ be $\mathbb{R}$-linear functions for each $s = 1, \ldots, h$. Assume that there exists a constant $\lambda \in \mathbb{R}$ such that*

$$\sum_{s=1}^{h} f_s^{(1)} \left( M^{(s)} \right) + f_s^{(2)} \left( \left( M^{(s)} \right)^{-1} \right) = \lambda, \tag{110}$$

*for all $\left( M^{(1)}, \ldots, M^{(h)} \right) \in \mathrm{GL}_D(\mathbb{R})^h$. Then*

$$f_s^{(1)} (M) = f_s^{(2)} (M) = \lambda = 0$$

*for all $s = 1, \ldots, h$ and $M \in \mathrm{GL}_D(\mathbb{R})$.*

*Proof.* Fix an arbitrary element $\left(M^{(1)}, \ldots, M^{(h)}\right)$ of $\mathrm{GL}_D(\mathbb{R})^h$. Then for arbitrary nonzero numbers $\lambda_1, \ldots, \lambda_h$, the tuple $\left(\lambda_1 M^{(1)}, \ldots, \lambda_h M^{(h)}\right)$ is also an element of $\mathrm{GL}_D(\mathbb{R})^h$. Since $f_s^{(1)}, f_s^{(2)}$ are linear, it follows from Equation (110) that

$$\sum_{s=1}^{h} \lambda_s f_s^{(1)}\left(M^{(s)}\right) + \frac{1}{\lambda_s} \cdot f_s^{(2)}\left(\left(M^{(s)}\right)^{-1}\right) = \lambda. \tag{111}$$

Consider the function

$$P(t_1, \ldots, t_h) := \left(\prod_{i=1}^{h} t_i\right) \cdot \left(\sum_{s=1}^{h} t_s f_s^{(1)}\left(M^{(s)}\right) + \frac{1}{t_s} \cdot f_s^{(2)}\left(\left(M^{(s)}\right)^{-1}\right) - \lambda\right) \tag{112}$$

as a polynomial in the variables $t_1, \ldots, t_h$ with coefficients in $\mathbb{R}$. Then, according to Equation (111), we have $P(\lambda_1, \ldots, \lambda_h) = 0$ for every nonzero numbers $\lambda_1, \ldots, \lambda_h$. This happens only when $P$ is a zero polynomial. Thus all coefficients of $P$ must be equal to zero. In particular, we have $\lambda = 0$ and $f_s^{(1)}\left(M^{(s)}\right) = f_s^{(2)}\left(\left(M^{(s)}\right)^{-1}\right) = 0$ for all $s = 1, \ldots, h$ and all $\left(M^{(1)}, \ldots, M^{(h)}\right) \in \mathrm{GL}_D(\mathbb{R})^h$. The lemma is then proved. $\square$

**Corollary D.3.** *Let $h$ and $D$ be a positive integers. Let $f_s^{(1)}, f_s^{(2)}, g_s^{(1)}, g_s^{(2)} : \mathbb{R}^{D \times D} \to \mathbb{R}$ be linear functions for each $s = 1, \ldots, h$. Assume that there are constants $\lambda_1, \lambda_2 \in \mathbb{R}$ such that:*

$$\sum_{s=1}^{h} f_s^{(1)}\left(M^{(s)}\right) + f_s^{(2)}\left(\left(M^{(s)}\right)^{-1}\right) + \lambda_1 = \sum_{s=1}^{h} g_s^{(1)}\left(M^{(s)}\right) + g_s^{(2)}\left(\left(M^{(s)}\right)^{-1}\right) + \lambda_2,$$
$$\tag{113}$$

*for all $\left(M^{(1)}, \ldots, M^{(h)}\right) \in \mathrm{GL}_D(\mathbb{R})^h$. Then $f_s^{(1)}(M) = g_s^{(1)}(M)$ and $f_s^{(2)}(M) = g_s^{(2)}(M)$ and $\lambda_1 = \lambda_2$ for all $s = 1, \ldots, h$ and $M \in \mathrm{GL}_D(\mathbb{R})$.*

*Proof.* Apply Lemma D.2 with $f_s^{(1)}$ is replace by $f_s^{(1)} - g_s^{(1)}$, $f_s^{(2)}$ is replace by $f_s^{(2)} - g_s^{(2)}$, and $\lambda$ is replaced by $\lambda_2 - \lambda_1$. $\square$

### D.3 FINDING THE CONSTRAINTS FOR THE UNKNOWN COEFFICIENTS

In the following, we will find necessary and sufficient conditions for the coefficients $\Phi^-$ such that $E(gU) = gE(U)$ for all $U \in \mathcal{U}$ and $g \in \mathcal{G}_{\mathcal{U}}$. We follow the parameter-sharing strategy. In particular, we first determine the entries of $E(gU)$ and $gE(U)$. Then, we compare the corresponding entries to determine the exact constraints on the coefficients.

#### D.3.1 COMPUTING $E(gU)$

Following Equation (86), we have

$$E(gU) = \left(\left([E(gW)]^{(Q,i)}, [E(gW)]^{(K,i)}, [E(gW)]^{(V,i)}, [E(gW)]^{(O,i)}\right)_{i=1,\ldots,h},\right.$$

$$\left.\left([E(gW)]^{(A)}, [E(gW)]^{(B)}\right), \left([E(gb)]^{(A)}, [E(gb)]^{(B)}\right)\right), \tag{114}$$

where

$$[E(gW)]_{j,k}^{(Q,i)} = \sum_{s=1}^{h} \sum_{p=1}^{D} \sum_{q=1}^{D} \Phi_{(QK,s):p,q}^{(Q,i):j,k} [gW gW]_{p,q}^{(QK,s)}$$

$$+ \sum_{s=1}^{h} \sum_{p=1}^{D} \sum_{q=1}^{D} \Phi_{(VO,s):p,q}^{(Q,i):j,k} [gW gW]_{p,q}^{(VO,s)}$$

$$+ \sum_{s=1}^{h} \sum_{p=1}^{D} \sum_{q=1}^{D_k} \Phi_{(Q,s):p,q}^{(Q,i):j,k} [gW]_{p,q}^{(Q,s)} + \sum_{s=1}^{h} \sum_{p=1}^{D} \sum_{q=1}^{D_k} \Phi_{(K,s):p,q}^{(Q,i):j,k} [gW]_{p,q}^{(K,s)}$$

$$+ \sum_{s=1}^{h} \sum_{p=1}^{D} \sum_{q=1}^{D_v} \Phi_{(V,s):p,q}^{(Q,i):j,k} [gW]_{p,q}^{(V,s)} + \sum_{s=1}^{h} \sum_{p=1}^{D_v} \sum_{q=1}^{D} \Phi_{(O,s):p,q}^{(Q,i):j,k} [gW]_{p,q}^{(O,s)}$$

$$+ \sum_{p=1}^{D} \sum_{q=1}^{D_A} \Phi_{(A):p,q}^{(Q,i):j,k} [gW]_{p,q}^{(A)} + \sum_{p=1}^{D_A} \sum_{q=1}^{D} \Phi_{(B):p,q}^{(Q,i):j,k} [gW]_{p,q}^{(B)}$$

$$+ \sum_{q=1}^{D_A} \Phi_{(A):q}^{(Q,i):j,k} [gb]_q^{(A)} + \sum_{q=1}^{D} \Phi_{(B):q}^{(Q,i):j,k} [gb]_q^{(B)} + \Phi^{(Q,i):j,k}, \quad (115)$$

$$[E(gW)]_{j,k}^{(K,i)} = \sum_{s=1}^{h} \sum_{p=1}^{D} \sum_{q=1}^{D} \Phi_{(QK,s):p,q}^{(K,i):j,k} [gWgW]_{p,q}^{(QK,s)}$$

$$+ \sum_{s=1}^{h} \sum_{p=1}^{D} \sum_{q=1}^{D} \Phi_{(VO,s):p,q}^{(K,i):j,k} [gWgW]_{p,q}^{(VO,s)}$$

$$+ \sum_{s=1}^{h} \sum_{p=1}^{D} \sum_{q=1}^{D_k} \Phi_{(Q,s):p,q}^{(K,i):j,k} [gW]_{p,q}^{(Q,s)} + \sum_{s=1}^{h} \sum_{p=1}^{D} \sum_{q=1}^{D_k} \Phi_{(K,s):p,q}^{(K,i):j,k} [gW]_{p,q}^{(K,s)}$$

$$+ \sum_{s=1}^{h} \sum_{p=1}^{D} \sum_{q=1}^{D_v} \Phi_{(V,s):p,q}^{(K,i):j,k} [gW]_{p,q}^{(V,s)} + \sum_{s=1}^{h} \sum_{p=1}^{D_v} \sum_{q=1}^{D} \Phi_{(O,s):p,q}^{(K,i):j,k} [gW]_{p,q}^{(O,s)}$$

$$+ \sum_{p=1}^{D} \sum_{q=1}^{D_A} \phi_{(A):p,q}^{(K,i):j,k} [gW]_{p,q}^{(A)} + \sum_{p=1}^{D_A} \sum_{q=1}^{D} \Phi_{(B):p,q}^{(K,i):j,k} [gW]_{p,q}^{(B)}$$

$$+ \sum_{q=1}^{D_A} \Phi_{(A):q}^{(K,i):j,k} [gb]_q^{(A)} + \sum_{q=1}^{D} \Phi_{(B):q}^{(K,i):j,k} [gb]_q^{(B)} + \Phi^{(K,i):j,k}, \quad (116)$$

$$[E(gW)]_{j,k}^{(V,i)} = \sum_{s=1}^{h} \sum_{p=1}^{D} \sum_{q=1}^{D} \Phi_{(QK,s):p,q}^{(V,i):j,k} [gWgW]_{p,q}^{(QK,s)}$$

$$+ \sum_{s=1}^{h} \sum_{p=1}^{D} \sum_{q=1}^{D} \Phi_{(VO,s):p,q}^{(V,i):j,k} [gWgW]_{p,q}^{(VO,s)}$$

$$+ \sum_{s=1}^{h} \sum_{p=1}^{D} \sum_{q=1}^{D_k} \Phi_{(Q,s):p,q}^{(V,i):j,k} [gW]_{p,q}^{(Q,s)} + \sum_{s=1}^{h} \sum_{p=1}^{D} \sum_{q=1}^{D_k} \Phi_{(K,s):p,q}^{(V,i):j,k} [gW]_{p,q}^{(K,s)}$$

$$+ \sum_{s=1}^{h} \sum_{p=1}^{D} \sum_{q=1}^{D_v} \Phi_{(V,s):p,q}^{(V,i):j,k} [gW]_{p,q}^{(V,s)} + \sum_{s=1}^{h} \sum_{p=1}^{D_v} \sum_{q=1}^{D} \Phi_{(O,s):p,q}^{(V,i):j,k} [gW]_{p,q}^{(O,s)}$$

$$+ \sum_{p=1}^{D} \sum_{q=1}^{D_A} \Phi_{(A):p,q}^{(V,i):j,k} [gW]_{p,q}^{(A)} + \sum_{p=1}^{D_A} \sum_{q=1}^{D} \Phi_{(B):p,q}^{(V,i):j,k} [gW]_{p,q}^{(B)}$$

$$+ \sum_{q=1}^{D_A} \Phi_{(A):q}^{(V,i):j,k} [gb]_q^{(A)} + \sum_{q=1}^{D} \Phi_{(B):q}^{(V,i):j,k} [gb]_q^{(B)} + \Phi^{(V,i):j,k}, \quad (117)$$

$$[E(gW)]_{j,k}^{(O,i)} = \sum_{s=1}^{h} \sum_{p=1}^{D} \sum_{q=1}^{D} \Phi_{(QK,s):p,q}^{(O,i):j,k} [gWgW]_{p,q}^{(QK,s)}$$

$$+ \sum_{s=1}^{h} \sum_{p=1}^{D} \sum_{q=1}^{D} \Phi_{(VO,s):p,q}^{(O,i):j,k} [gWgW]_{p,q}^{(VO,s)}$$

$$+ \sum_{s=1}^{h} \sum_{p=1}^{D} \sum_{q=1}^{D_k} \Phi_{(Q,s):p,q}^{(O,i):j,k} [gW]_{p,q}^{(Q,s)} + \sum_{s=1}^{h} \sum_{p=1}^{D} \sum_{q=1}^{D_k} \Phi_{(K,s):p,q}^{(O,i):j,k} [gW]_{p,q}^{(K,s)}$$

$$+ \sum_{s=1}^{h} \sum_{p=1}^{D} \sum_{q=1}^{D_v} \Phi_{(V,s):p,q}^{(O,i):j,k} [gW]_{p,q}^{(V,s)} + \sum_{s=1}^{h} \sum_{p=1}^{D} \sum_{q=1}^{D} \Phi_{(O,s):p,q}^{(O,i):j,k} [gW]_{p,q}^{(O,s)}$$

$$+ \sum_{p=1}^{D}\sum_{q=1}^{D_A} \Phi_{(A):p,q}^{(O,i):j,k}[gW]_{p,q}^{(A)} + \sum_{p=1}^{D_A}\sum_{q=1}^{D} \Phi_{(B):p,q}^{(O,i):j,k}[gW]_{p,q}^{(B)}$$

$$+ \sum_{q=1}^{D_A} \Phi_{(A):q}^{(O,i):j,k}[gb]_{q}^{(A)} + \sum_{q=1}^{D} \Phi_{(B):q}^{(O,i):j,k}[gb]_{q}^{(B)} + \Phi^{(O,i):j,k}, \qquad (118)$$

$$[E(gW)]_{j,k}^{(A)} = \sum_{s=1}^{h}\sum_{p=1}^{D}\sum_{q=1}^{D} \Phi_{(QK,s):p,q}^{(A):j,k}[gWgW]_{p,q}^{(QK,s)}$$

$$+ \sum_{s=1}^{h}\sum_{p=1}^{D}\sum_{q=1}^{D} \Phi_{(VO,s):p,q}^{(A):j,k}[gWgW]_{p,q}^{(VO,s)}$$

$$+ \sum_{s=1}^{h}\sum_{p=1}^{D}\sum_{q=1}^{D_k} \Phi_{(Q,s):p,q}^{(A):j,k}[gW]_{p,q}^{(Q,s)} + \sum_{s=1}^{h}\sum_{p=1}^{D}\sum_{q=1}^{D_k} \Phi_{(K,s):p,q}^{(A):j,k}[gW]_{p,q}^{(K,s)}$$

$$+ \sum_{s=1}^{h}\sum_{p=1}^{D}\sum_{q=1}^{D_v} \Phi_{(V,s):p,q}^{(A):j,k}[gW]_{p,q}^{(V,s)} + \sum_{s=1}^{h}\sum_{p=1}^{D_v}\sum_{q=1}^{D} \Phi_{(O,s):p,q}^{(A):j,k}[gW]_{p,q}^{(O,s)}$$

$$+ \sum_{p=1}^{D}\sum_{q=1}^{D_A} \Phi_{(A):p,q}^{(A):j,k}[gW]_{p,q}^{(A)} + \sum_{p=1}^{D_A}\sum_{q=1}^{D} \Phi_{(B):p,q}^{(A):j,k}[gW]_{p,q}^{(B)}$$

$$+ \sum_{q=1}^{D_A} \Phi_{(A):q}^{(A):j,k}[gb]_{q}^{(A)} + \sum_{q=1}^{D} \Phi_{(B):q}^{(A):j,k}[gb]_{q}^{(B)} + \Phi^{(A):j,k}, \qquad (119)$$

$$[E(gW)]_{j,k}^{(B)} = \sum_{s=1}^{h}\sum_{p=1}^{D}\sum_{q=1}^{D} \Phi_{(QK,s):p,q}^{(B):j,k}[gWgW]_{p,q}^{(QK,s)}$$

$$+ \sum_{s=1}^{h}\sum_{p=1}^{D}\sum_{q=1}^{D} \Phi_{(VO,s):p,q}^{(B):j,k}[gWgW]_{p,q}^{(VO,s)}$$

$$+ \sum_{s=1}^{h}\sum_{p=1}^{D}\sum_{q=1}^{D_k} \Phi_{(Q,s):p,q}^{(B):j,k}[gW]_{p,q}^{(Q,s)} + \sum_{s=1}^{h}\sum_{p=1}^{D}\sum_{q=1}^{D_k} \Phi_{(K,s):p,q}^{(B):j,k}[gW]_{p,q}^{(K,s)}$$

$$+ \sum_{s=1}^{h}\sum_{p=1}^{D}\sum_{q=1}^{D_v} \Phi_{(V,s):p,q}^{(B):j,k}[gW]_{p,q}^{(V,s)} + \sum_{s=1}^{h}\sum_{p=1}^{D_v}\sum_{q=1}^{D} \Phi_{(O,s):p,q}^{(B):j,k}[gW]_{p,q}^{(O,s)}$$

$$+ \sum_{p=1}^{D}\sum_{q=1}^{D_A} \Phi_{(A):p,q}^{(B):j,k}[gW]_{p,q}^{(A)} + \sum_{p=1}^{D_A}\sum_{q=1}^{D} \Phi_{(B):p,q}^{(B):j,k}[gW]_{p,q}^{(B)}$$

$$+ \sum_{q=1}^{D_A} \Phi_{(A):q}^{(B):j,k}[gb]_{q}^{(A)} + \sum_{q=1}^{D} \Phi_{(B):q}^{(B):j,k}[gb]_{q}^{(B)} + \Phi^{(B):j,k}, \qquad (120)$$

$$[E(gb)]_{k}^{(A)} = \sum_{s=1}^{h}\sum_{p=1}^{D}\sum_{q=1}^{D} \Phi_{(QK,s):p,q}^{(A):k}[gWgW]_{p,q}^{(QK,s)}$$

$$+ \sum_{s=1}^{h}\sum_{p=1}^{D}\sum_{q=1}^{D} \Phi_{(VO,s):p,q}^{(A):k}[gWgW]_{p,q}^{(VO,s)}$$

$$+ \sum_{s=1}^{h}\sum_{p=1}^{D}\sum_{q=1}^{D_k} \Phi_{(Q,s):p,q}^{(A):k}[gW]_{p,q}^{(Q,s)} + \sum_{s=1}^{h}\sum_{p=1}^{D}\sum_{q=1}^{D_k} \Phi_{(K,s):p,q}^{(A):k}[gW]_{p,q}^{(K,s)}$$

$$+ \sum_{s=1}^{h}\sum_{p=1}^{D}\sum_{q=1}^{D_v} \Phi_{(V,s):p,q}^{(A):k}[gW]_{p,q}^{(V,s)} + \sum_{s=1}^{h}\sum_{p=1}^{D_v}\sum_{q=1}^{D} \Phi_{(O,s):p,q}^{(A):k}[gW]_{p,q}^{(O,s)}$$

$$+ \sum_{p=1}^{D} \sum_{q=1}^{D_A} \phi_{(A):p,q}^{(A):k} [gW]_{p,q}^{(A)} + \sum_{p=1}^{D_A} \sum_{q=1}^{D} \Phi_{(B):p,q}^{(A):k} [gW]_{p,q}^{(B)}$$

$$+ \sum_{q=1}^{D_A} \Phi_{(A):q}^{(A):k} [gb]_q^{(A)} + \sum_{q=1}^{D} \Phi_{(B):q}^{(A):k} [gb]_q^{(B)} + \Phi^{(A):k}, \tag{121}$$

$$[E(gb)]_k^{(B)} = \sum_{s=1}^{h} \sum_{p=1}^{D} \sum_{q=1}^{D} \Phi_{(QK,s):p,q}^{(B):k} [gWgW]_{p,q}^{(QK,s)}$$

$$+ \sum_{s=1}^{h} \sum_{p=1}^{D} \sum_{q=1}^{D} \Phi_{(VO,s):p,q}^{(B):k} [gWgW]_{p,q}^{(VO,s)}$$

$$+ \sum_{s=1}^{h} \sum_{p=1}^{D} \sum_{q=1}^{D_k} \Phi_{(Q,s):p,q}^{(B):k} [gW]_{p,q}^{(Q,s)} + \sum_{s=1}^{h} \sum_{p=1}^{D} \sum_{q=1}^{D_k} \Phi_{(K,s):p,q}^{(B):k} [gW]_{p,q}^{(K,s)}$$

$$+ \sum_{s=1}^{h} \sum_{p=1}^{D} \sum_{q=1}^{D_v} \Phi_{(V,s):p,q}^{(B):k} [gW]_{p,q}^{(V,s)} + \sum_{s=1}^{h} \sum_{p=1}^{D_v} \sum_{q=1}^{D} \Phi_{(O,s):p,q}^{(B):k} [gW]_{p,q}^{(O,s)}$$

$$+ \sum_{p=1}^{D} \sum_{q=1}^{D_A} \Phi_{(A):p,q}^{(B):k} [gW]_{p,q}^{(A)} + \sum_{p=1}^{D_A} \sum_{q=1}^{D} \Phi_{(B):p,q}^{(B):k} [gW]_{p,q}^{(B)}$$

$$+ \sum_{q=1}^{D_A} \Phi_{(A):q}^{(B):k} [gb]_q^{(A)} + \sum_{q=1}^{D} \Phi_{(B):q}^{(B):k} [gb]_q^{(B)} + \Phi^{(B):k}. \tag{122}$$

Using Proposition C.3, we can proceed the right hand side of the above equation further as:

$$[E(gW)]_{j,k}^{(Q,i)} = \sum_{s=1}^{h} \sum_{p=1}^{D} \sum_{q=1}^{D} \Phi_{(QK,s):p,q}^{(Q,i):j,k} [WW]_{p,q}^{(QK,\tau(s))}$$

$$+ \sum_{s=1}^{h} \sum_{p=1}^{D} \sum_{q=1}^{D} \Phi_{(VO,s):p,q}^{(Q,i):j,k} [WW]_{p,\pi_O(q)}^{(VO,\tau(s))}$$

$$+ \sum_{s=1}^{h} \sum_{p=1}^{D} \sum_{q=1}^{D_k} \Phi_{(Q,s):p,q}^{(Q,i):j,k} \left[ [W]^{(Q,\tau(s))} \cdot \left( M^{(\tau(s))} \right)^{\top} \right]_{p,q}$$

$$+ \sum_{s=1}^{h} \sum_{p=1}^{D} \sum_{q=1}^{D_k} \Phi_{(K,s):p,q}^{(Q,i):j,k} \left[ [W]^{(K,\tau(s))} \cdot \left( M^{(\tau(s))} \right)^{-1} \right]_{p,q}$$

$$+ \sum_{s=1}^{h} \sum_{p=1}^{D} \sum_{q=1}^{D_v} \Phi_{(V,s):p,q}^{(Q,i):j,k} \left[ [W]^{(V,\tau(s))} \cdot N^{(\tau(s))} \right]_{p,q}$$

$$+ \sum_{s=1}^{h} \sum_{p=1}^{D_v} \sum_{q=1}^{D} \Phi_{(O,s):p,q}^{(Q,i):j,k} \left[ \left( N^{(\tau(s))} \right)^{-1} \cdot [W]^{(O,\tau(s))} \right]_{p,\pi_O(q)}$$

$$+ \sum_{p=1}^{D} \sum_{q=1}^{D_A} \Phi_{(A):p,q}^{(Q,i):j,k} [W]_{\pi_O(p),\pi_A(q)}^{(A)} + \sum_{p=1}^{D_A} \sum_{q=1}^{D} \Phi_{(B):p,q}^{(Q,i):j,k} [W]_{\pi_A(p),q}^{(B)}$$

$$+ \sum_{q=1}^{D_A} \Phi_{(A):q}^{(Q,i):j,k} [b]_{\pi_A(q)}^{(A)} + \sum_{q=1}^{D} \Phi_{(B):q}^{(Q,i):j,k} [b]_q^{(B)} + \Phi^{(Q,i):j,k}, \tag{123}$$

$$[E(gW)]_{j,k}^{(K,i)} = \sum_{s=1}^{h} \sum_{p=1}^{D} \sum_{q=1}^{D} \Phi_{(QK,s):p,q}^{(K,i):j,k} [WW]_{p,q}^{(QK,\tau(s))}$$

$$+ \sum_{s=1}^{h} \sum_{p=1}^{D} \sum_{q=1}^{D} \Phi_{(VO,s):p,q}^{(K,i):j,k} [WW]_{p,\pi_O(q)}^{(VO,\tau(s))}$$

$$+ \sum_{s=1}^{h} \sum_{p=1}^{D} \sum_{q=1}^{D_k} \Phi_{(Q,s):p,q}^{(K,i):j,k} \left[ [W]^{(Q,\tau(s))} \cdot \left( M^{(\tau(s))} \right)^{\top} \right]_{p,q}$$

$$+ \sum_{s=1}^{h} \sum_{p=1}^{D} \sum_{q=1}^{D_k} \Phi_{(K,s):p,q}^{(K,i):j,k} \left[ [W]^{(K,\tau(s))} \cdot \left( M^{(\tau(s))} \right)^{-1} \right]_{p,q}$$

$$+ \sum_{s=1}^{h} \sum_{p=1}^{D} \sum_{q=1}^{D_v} \Phi_{(V,s):p,q}^{(K,i):j,k} \left[ [W]^{(V,\tau(s))} \cdot N^{(\tau(s))} \right]_{p,q}$$

$$+ \sum_{s=1}^{h} \sum_{p=1}^{D_v} \sum_{q=1}^{D} \Phi_{(O,s):p,q}^{(K,i):j,k} \left[ \left( N^{(\tau(s))} \right)^{-1} \cdot [W]^{(O,\tau(s))} \right]_{p,\pi_O(q)}$$

$$+ \sum_{p=1}^{D} \sum_{q=1}^{D_A} \Phi_{(A):p,q}^{(K,i):j,k} [W]_{\pi_O(p),\pi_A(q)}^{(A)} + \sum_{p=1}^{D_A} \sum_{q=1}^{D} \Phi_{(B):p,q}^{(K,i):j,k} [W]_{\pi_A(p),q}^{(B)}$$

$$+ \sum_{q=1}^{D_A} \Phi_{(A):q}^{(K,i):j,k} [b]_{\pi_A(q)}^{(A)} + \sum_{q=1}^{D} \Phi_{(B):q}^{(K,i):j,k} [b]_q^{(B)} + \Phi^{(K,i):j,k}, \qquad (124)$$

$$[E(gW)]_{j,k}^{(V,i)} = \sum_{s=1}^{h} \sum_{p=1}^{D} \sum_{q=1}^{D} \Phi_{(QK,s):p,q}^{(V,i):j,k} [WW]_{p,q}^{(QK,\tau(s))}$$

$$+ \sum_{s=1}^{h} \sum_{p=1}^{D} \sum_{q=1}^{D} \Phi_{(VO,s):p,q}^{(V,i):j,k} [WW]_{p,\pi_O(q)}^{(VO,\tau(s))}$$

$$+ \sum_{s=1}^{h} \sum_{p=1}^{D} \sum_{q=1}^{D_k} \Phi_{(Q,s):p,q}^{(V,i):j,k} \left[ [W]^{(Q,\tau(s))} \cdot \left( M^{(\tau(s))} \right)^{\top} \right]_{p,q}$$

$$+ \sum_{s=1}^{h} \sum_{p=1}^{D} \sum_{q=1}^{D_k} \Phi_{(K,s):p,q}^{(V,i):j,k} \left[ [W]^{(K,\tau(s))} \cdot \left( M^{(\tau(s))} \right)^{-1} \right]_{p,q}$$

$$+ \sum_{s=1}^{h} \sum_{p=1}^{D} \sum_{q=1}^{D_v} \Phi_{(V,s):p,q}^{(V,i):j,k} \left[ [W]^{(V,\tau(s))} \cdot N^{(\tau(s))} \right]_{p,q}$$

$$+ \sum_{s=1}^{h} \sum_{p=1}^{D_v} \sum_{q=1}^{D} \Phi_{(O,s):p,q}^{(V,i):j,k} \left[ \left( N^{(\tau(s))} \right)^{-1} \cdot [W]^{(O,\tau(s))} \right]_{p,\pi_O(q)}$$

$$+ \sum_{p=1}^{D} \sum_{q=1}^{D_A} \Phi_{(A):p,q}^{(V,i):j,k} [W]_{\pi_O(p),\pi_A(q)}^{(A)} + \sum_{p=1}^{D_A} \sum_{q=1}^{D} \Phi_{(B):p,q}^{(V,i):j,k} [W]_{\pi_A(p),q}^{(B)}$$

$$+ \sum_{q=1}^{D_A} \Phi_{(A):q}^{(V,i):j,k} [b]_{\pi_A(q)}^{(A)} + \sum_{q=1}^{D} \Phi_{(B):q}^{(V,i):j,k} [b]_q^{(B)} + \Phi^{(V,i):j,k}, \qquad (125)$$

$$[E(gW)]_{j,k}^{(O,i)} = \sum_{s=1}^{h} \sum_{p=1}^{D} \sum_{q=1}^{D} \Phi_{(QK,s):p,q}^{(O,i):j,k} [WW]_{p,q}^{(QK,\tau(s))}$$

$$+ \sum_{s=1}^{h} \sum_{p=1}^{D} \sum_{q=1}^{D} \Phi_{(VO,s):p,q}^{(O,i):j,k} [WW]_{p,\pi_O(q)}^{(VO,\tau(s))}$$

$$+ \sum_{s=1}^{h} \sum_{p=1}^{D} \sum_{q=1}^{D_k} \Phi_{(Q,s):p,q}^{(O,i):j,k} \left[ [W]^{(Q,\tau(s))} \cdot \left( M^{(\tau(s))} \right)^{\top} \right]_{p,q}$$

$$+ \sum_{s=1}^{h} \sum_{p=1}^{D} \sum_{q=1}^{D_k} \Phi_{(K,s):p,q}^{(O,i):j,k} \left[ [W]^{(K,\tau(s))} \cdot \left( M^{(\tau(s))} \right)^{-1} \right]_{p,q}$$

$$+ \sum_{s=1}^{h} \sum_{p=1}^{D} \sum_{q=1}^{D_v} \Phi_{(V,s):p,q}^{(O,i):j,k} \left[ [W]^{(V,\tau(s))} \cdot N^{(\tau(s))} \right]_{p,q}$$

$$+ \sum_{s=1}^{h} \sum_{p=1}^{D_v} \sum_{q=1}^{D} \Phi_{(O,s):p,q}^{(O,i):j,k} \left[ \left( N^{(\tau(s))} \right)^{-1} \cdot [W]^{(O,\tau(s))} \right]_{p,\pi_O(q)}$$

$$+ \sum_{p=1}^{D} \sum_{q=1}^{D_A} \Phi_{(A):p,q}^{(O,i):j,k} [W]_{\pi_O(p),\pi_A(q)}^{(A)} + \sum_{p=1}^{D_A} \sum_{q=1}^{D} \Phi_{(B):p,q}^{(O,i):j,k} [W]_{\pi_A(p),q}^{(B)}$$

$$+ \sum_{q=1}^{D_A} \Phi_{(A):q}^{(O,i):j,k} [b]_{\pi_A(q)}^{(A)} + \sum_{q=1}^{D} \Phi_{(B):q}^{(O,i):j,k} [b]_q^{(B)} + \Phi^{(O,i):j,k}, \qquad (126)$$

$$[E(gW)]_{j,k}^{(A)} = \sum_{s=1}^{h} \sum_{p=1}^{D} \sum_{q=1}^{D} \Phi_{(QK,s):p,q}^{(A):j,k} [WW]_{p,q}^{(QK,\tau(s))}$$

$$+ \sum_{s=1}^{h} \sum_{p=1}^{D} \sum_{q=1}^{D} \Phi_{(VO,s):p,q}^{(A):j,k} [WW]_{p,\pi_O(q)}^{(VO,\tau(s))}$$

$$+ \sum_{s=1}^{h} \sum_{p=1}^{D} \sum_{q=1}^{D_k} \Phi_{(Q,s):p,q}^{(A):j,k} \left[ [W]^{(Q,\tau(s))} \cdot \left( M^{(\tau(s))} \right)^{\top} \right]_{p,q}$$

$$+ \sum_{s=1}^{h} \sum_{p=1}^{D} \sum_{q=1}^{D_k} \Phi_{(K,s):p,q}^{(A):j,k} \left[ [W]^{(K,\tau(s))} \cdot \left( M^{(\tau(s))} \right)^{-1} \right]_{p,q}$$

$$+ \sum_{s=1}^{h} \sum_{p=1}^{D} \sum_{q=1}^{D_v} \Phi_{(V,s):p,q}^{(A):j,k} \left[ [W]^{(V,\tau(s))} \cdot N^{(\tau(s))} \right]_{p,q}$$

$$+ \sum_{s=1}^{h} \sum_{p=1}^{D_v} \sum_{q=1}^{D} \Phi_{(O,s):p,q}^{(A):j,k} \left[ \left( N^{(\tau(s))} \right)^{-1} \cdot [W]^{(O,\tau(s))} \right]_{p,\pi_O(q)}$$

$$+ \sum_{p=1}^{D} \sum_{q=1}^{D_A} \Phi_{(A):p,q}^{(A):j,k} [W]_{\pi_O(p),\pi_A(q)}^{(A)} + \sum_{p=1}^{D_A} \sum_{q=1}^{D} \Phi_{(B):p,q}^{(A):j,k} [W]_{\pi_A(p),q}^{(B)}$$

$$+ \sum_{q=1}^{D_A} \Phi_{(A):q}^{(A):j,k} [b]_{\pi_A(q)}^{(A)} + \sum_{q=1}^{D} \Phi_{(B):q}^{(A):j,k} [b]_q^{(B)} + \Phi^{(A):j,k}, \qquad (127)$$

$$[E(gW)]_{j,k}^{(B)} = \sum_{s=1}^{h} \sum_{p=1}^{D} \sum_{q=1}^{D} \Phi_{(QK,s):p,q}^{(B):j,k} [WW]_{p,q}^{(QK,\tau(s))}$$

$$+ \sum_{s=1}^{h} \sum_{p=1}^{D} \sum_{q=1}^{D} \Phi_{(VO,s):p,q}^{(B):j,k} [WW]_{p,\pi_O(q)}^{(VO,\tau(s))}$$

$$+ \sum_{s=1}^{h} \sum_{p=1}^{D} \sum_{q=1}^{D_k} \Phi_{(Q,s):p,q}^{(B):j,k} \left[ [W]^{(Q,\tau(s))} \cdot \left( M^{(\tau(s))} \right)^{\top} \right]_{p,q} \qquad (128)$$

$$+ \sum_{s=1}^{h} \sum_{p=1}^{D} \sum_{q=1}^{D_k} \Phi_{(K,s):p,q}^{(B):j,k} \left[ [W]^{(K,\tau(s))} \cdot \left( M^{(\tau(s))} \right)^{-1} \right]_{p,q}$$

$$+ \sum_{s=1}^{h} \sum_{p=1}^{D} \sum_{q=1}^{D_v} \Phi_{(V,s):p,q}^{(B):j,k} \left[ [W]^{(V,\tau(s))} \cdot N^{(\tau(s))} \right]_{p,q}$$

$$+ \sum_{s=1}^{h} \sum_{p=1}^{D_v} \sum_{q=1}^{D} \Phi_{(O,s):p,q}^{(B):j,k} \left[ \left( N^{(\tau(s))} \right)^{-1} \cdot [W]^{(O,\tau(s))} \right]_{p,\pi_O(q)}$$

$$+ \sum_{p=1}^{D} \sum_{q=1}^{D_A} \Phi_{(A):p,q}^{(B):j,k} [W]_{\pi_O(p),\pi_A(q)}^{(A)} + \sum_{p=1}^{D_A} \sum_{q=1}^{D} \Phi_{(B):p,q}^{(B):j,k} [W]_{\pi_A(p),q}^{(B)}$$

$$+ \sum_{q=1}^{D_A} \Phi_{(A):q}^{(B):j,k} [b]_{\pi_A(q)}^{(A)} + \sum_{q=1}^{D} \Phi_{(B):q}^{(B):j,k} [b]_q^{(B)} + \Phi^{(B):j,k}, \tag{129}$$

$$[E(gb)]_k^{(A)} = \sum_{s=1}^{h} \sum_{p=1}^{D} \sum_{q=1}^{D} \Phi_{(QK,s):p,q}^{(A):k} [WW]_{p,q}^{(QK,\tau(s))}$$

$$+ \sum_{s=1}^{h} \sum_{p=1}^{D} \sum_{q=1}^{D} \Phi_{(VO,s):p,q}^{(A):k} [WW]_{p,\pi_O(q)}^{(VO,\tau(s))}$$

$$+ \sum_{s=1}^{h} \sum_{p=1}^{D} \sum_{q=1}^{D_k} \Phi_{(Q,s):p,q}^{(A):k} \left[ [W]^{(Q,\tau(s))} \cdot \left( M^{(\tau(s))} \right)^\top \right]_{p,q}$$

$$+ \sum_{s=1}^{h} \sum_{p=1}^{D} \sum_{q=1}^{D_k} \Phi_{(K,s):p,q}^{(A):k} \left[ [W]^{(K,\tau(s))} \cdot \left( M^{(\tau(s))} \right)^{-1} \right]_{p,q}$$

$$+ \sum_{s=1}^{h} \sum_{p=1}^{D} \sum_{q=1}^{D_v} \Phi_{(V,s):p,q}^{(A):k} \left[ [W]^{(V,\tau(s))} \cdot N^{(\tau(s))} \right]_{p,q}$$

$$+ \sum_{s=1}^{h} \sum_{p=1}^{D_v} \sum_{q=1}^{D} \Phi_{(O,s):p,q}^{(A):k} \left[ \left( N^{(\tau(s))} \right)^{-1} \cdot [W]^{(O,\tau(s))} \right]_{p,\pi_O(q)}$$

$$+ \sum_{p=1}^{D} \sum_{q=1}^{D_A} \Phi_{(A):p,q}^{(A):k} [W]_{\pi_O(p),\pi_A(q)}^{(A)} + \sum_{p=1}^{D_A} \sum_{q=1}^{D} \Phi_{(B):p,q}^{(A):k} [W]_{\pi_A(p),q}^{(B)}$$

$$+ \sum_{q=1}^{D_A} \Phi_{(A):q}^{(A):k} [b]_{\pi_A(q)}^{(A)} + \sum_{q=1}^{D} \Phi_{(B):q}^{(A):k} [b]_q^{(B)} + \Phi^{(A):k}, \tag{130}$$

$$[E(gb)]_k^{(B)} = \sum_{s=1}^{h} \sum_{p=1}^{D} \sum_{q=1}^{D} \Phi_{(QK,s):p,q}^{(B):k} [WW]_{p,q}^{(QK,\tau(s))}$$

$$+ \sum_{s=1}^{h} \sum_{p=1}^{D} \sum_{q=1}^{D} \Phi_{(VO,s):p,q}^{(B):k} [WW]_{p,\pi_O(q)}^{(VO,\tau(s))}$$

$$+ \sum_{s=1}^{h} \sum_{p=1}^{D} \sum_{q=1}^{D_k} \Phi_{(Q,s):p,q}^{(B):k} \left[ [W]^{(Q,\tau(s))} \cdot \left( M^{(\tau(s))} \right)^\top \right]_{p,q}$$

$$+ \sum_{s=1}^{h} \sum_{p=1}^{D} \sum_{q=1}^{D_k} \Phi_{(K,s):p,q}^{(B):k} \left[ [W]^{(K,\tau(s))} \cdot \left( M^{(\tau(s))} \right)^{-1} \right]_{p,q}$$

$$+ \sum_{s=1}^{h} \sum_{p=1}^{D} \sum_{q=1}^{D_v} \Phi_{(V,s):p,q}^{(B):k} \left[ [W]^{(V,\tau(s))} \cdot N^{(\tau(s))} \right]_{p,q}$$

$$+ \sum_{s=1}^{h} \sum_{p=1}^{D_v} \sum_{q=1}^{D} \Phi_{(O,s):p,q}^{(B):k} \left[ \left( N^{(\tau(s))} \right)^{-1} \cdot [W]^{(O,\tau(s))} \right]_{p,\pi_O(q)}$$

$$+ \sum_{p=1}^{D} \sum_{q=1}^{D_A} \Phi_{(A):p,q}^{(B):k} [W]_{\pi_O(p),\pi_A(q)}^{(A)} + \sum_{p=1}^{D_A} \sum_{q=1}^{D} \Phi_{(B):p,q}^{(B):k} [W]_{\pi_A(p),q}^{(B)}$$

$$+ \sum_{q=1}^{D_A} \Phi_{(A):q}^{(B):k} [b]_{\pi_A(q)}^{(A)} + \sum_{q=1}^{D} \Phi_{(B):q}^{(B):k} [b]_q^{(B)} + \Phi^{(B):k}. \tag{131}$$

Using the symmetries of the indices, we can rearrange the right hand sides of the above equations as:

$$
[E(gW)]_{j,k}^{(Q,i)} = \sum_{s=1}^{h} \sum_{p=1}^{D} \sum_{q=1}^{D} \Phi_{(QK,\tau^{-1}(s)):p,q}^{(Q,i):j,k} [WW]_{p,q}^{(QK,s)}
$$

$$
+ \sum_{s=1}^{h} \sum_{p=1}^{D} \sum_{q=1}^{D} \Phi_{(VO,\tau^{-1}(s)):p,\pi_O^{-1}(q)}^{(Q,i):j,k} [WW]_{p,q}^{(VO,s)}
$$

$$
+ \sum_{s=1}^{h} \sum_{p=1}^{D} \sum_{q=1}^{D_k} \Phi_{(Q,\tau^{-1}(s)):p,q}^{(Q,i):j,k} \left[ [W]^{(Q,s)} \cdot \left( M^{(s)} \right)^{\top} \right]_{p,q}
$$

$$
+ \sum_{s=1}^{h} \sum_{p=1}^{D} \sum_{q=1}^{D_k} \Phi_{(K,\tau^{-1}(s)):p,q}^{(Q,i):j,k} \left[ [W]^{(K,s)} \cdot \left( M^{(s)} \right)^{-1} \right]_{p,q}
$$

$$
+ \sum_{s=1}^{h} \sum_{p=1}^{D} \sum_{q=1}^{D_v} \Phi_{(V,\tau^{-1}(s)):p,q}^{(Q,i):j,k} \left[ [W]^{(V,s)} \cdot N^{(s)} \right]_{p,q}
$$

$$
+ \sum_{s=1}^{h} \sum_{p=1}^{D_v} \sum_{q=1}^{D} \Phi_{(O,\tau^{-1}(s)):p,\pi_O^{-1}(q)}^{(Q,i):j,k} \left[ \left( N^{(s)} \right)^{-1} \cdot [W]^{(O,s)} \right]_{p,q}
$$

$$
+ \sum_{p=1}^{D} \sum_{q=1}^{D_A} \Phi_{(A):\pi_O^{-1}(p),\pi_A^{-1}(q)}^{(Q,i):j,k} [W]_{p,q}^{(A)} + \sum_{p=1}^{D_A} \sum_{q=1}^{D} \Phi_{(B):\pi_A^{-1}(p),q}^{(Q,i):j,k} [W]_{p,q}^{(B)}
$$

$$
+ \sum_{q=1}^{D_A} \Phi_{(A):\pi_A^{-1}(q)}^{(Q,i):j,k} [b]_q^{(A)} + \sum_{q=1}^{D} \Phi_{(B):q}^{(Q,i):j,k} [b]_q^{(B)} + \Phi^{(Q,i):j,k}, \tag{132}
$$

$$
[E(gW)]_{j,k}^{(K,i)} = \sum_{s=1}^{h} \sum_{p=1}^{D} \sum_{q=1}^{D} \Phi_{(QK,\tau^{-1}(s)):p,q}^{(K,i):j,k} [WW]_{p,q}^{(QK,s)}
$$

$$
+ \sum_{s=1}^{h} \sum_{p=1}^{D} \sum_{q=1}^{D} \Phi_{(VO,\tau^{-1}(s)):p,\pi_O^{-1}(q)}^{(K,i):j,k} [WW]_{p,q}^{(VO,s)}
$$

$$
+ \sum_{s=1}^{h} \sum_{p=1}^{D} \sum_{q=1}^{D_k} \Phi_{(Q,\tau^{-1}(s)):p,q}^{(K,i):j,k} \left[ [W]^{(Q,s)} \cdot \left( M^{(s)} \right)^{\top} \right]_{p,q}
$$

$$
+ \sum_{s=1}^{h} \sum_{p=1}^{D} \sum_{q=1}^{D_k} \Phi_{(K,\tau^{-1}(s)):p,q}^{(K,i):j,k} \left[ [W]^{(K,s)} \cdot \left( M^{(s)} \right)^{-1} \right]_{p,q}
$$

$$
+ \sum_{s=1}^{h} \sum_{p=1}^{D} \sum_{q=1}^{D_v} \Phi_{(V,\tau^{-1}(s)):p,q}^{(K,i):j,k} \left[ [W]^{(V,s)} \cdot N^{(s)} \right]_{p,q}
$$

$$
+ \sum_{s=1}^{h} \sum_{p=1}^{D_v} \sum_{q=1}^{D} \Phi_{(O,\tau^{-1}(s)):p,\pi_O^{-1}(q)}^{(K,i):j,k} \left[ \left( N^{(s)} \right)^{-1} \cdot [W]^{(O,s)} \right]_{p,q}
$$

$$
+ \sum_{p=1}^{D} \sum_{q=1}^{D_A} \Phi_{(A):\pi_O^{-1}(p),\pi_A^{-1}(q)}^{(K,i):j,k} [W]_{p,q}^{(A)} + \sum_{p=1}^{D_A} \sum_{q=1}^{D} \Phi_{(B):\pi_A^{-1}(p),q}^{(K,i):j,k} [W]_{p,q}^{(B)}
$$

$$
+ \sum_{q=1}^{D_A} \Phi_{(A):\pi_A^{-1}(q)}^{(K,i):j,k} [b]_q^{(A)} + \sum_{q=1}^{D} \Phi_{(B):q}^{(K,i):j,k} [b]_q^{(B)} + \Phi^{(K,i):j,k}, \tag{133}
$$

$$[E(gW)]_{j,k}^{(V,i)} = \sum_{s=1}^{h}\sum_{p=1}^{D}\sum_{q=1}^{D}\Phi_{(QK,\tau^{-1}(s)):p,q}^{(V,i):j,k}[WW]_{p,q}^{(QK,s)}$$

$$+ \sum_{s=1}^{h}\sum_{p=1}^{D}\sum_{q=1}^{D}\Phi_{(VO,\tau^{-1}(s)):p,\pi_O^{-1}(q)}^{(V,i):j,k}[WW]_{p,q}^{(VO,s)}$$

$$+ \sum_{s=1}^{h}\sum_{p=1}^{D}\sum_{q=1}^{D_k}\Phi_{(Q,\tau^{-1}(s)):p,q}^{(V,i):j,k}\left[[W]^{(Q,s)}\cdot\left(M^{(s)}\right)^{\top}\right]_{p,q}$$

$$+ \sum_{s=1}^{h}\sum_{p=1}^{D}\sum_{q=1}^{D_k}\Phi_{(K,\tau^{-1}(s)):p,q}^{(V,i):j,k}\left[[W]^{(K,s)}\cdot\left(M^{(s)}\right)^{-1}\right]_{p,q}$$

$$+ \sum_{s=1}^{h}\sum_{p=1}^{D}\sum_{q=1}^{D_v}\Phi_{(V,\tau^{-1}(s)):p,q}^{(V,i):j,k}\left[[W]^{(V,s)}\cdot N^{(s)}\right]_{p,q}$$

$$+ \sum_{s=1}^{h}\sum_{p=1}^{D_v}\sum_{q=1}^{D}\Phi_{(O,\tau^{-1}(s)):p,\pi_O^{-1}(q)}^{(V,i):j,k}\left[\left(N^{(s)}\right)^{-1}\cdot[W]^{(O,s)}\right]_{p,q}$$

$$+ \sum_{p=1}^{D}\sum_{q=1}^{D_A}\Phi_{(A):\pi_O^{-1}(p),\pi_A^{-1}(q)}^{(V,i):j,k}[W]_{p,q}^{(A)} + \sum_{p=1}^{D_A}\sum_{q=1}^{D}\Phi_{(B):\pi_A^{-1}(p),q}^{(V,i):j,k}[W]_{p,q}^{(B)}$$

$$+ \sum_{q=1}^{D_A}\Phi_{(A):\pi_A^{-1}(q)}^{(V,i):j,k}[b]_q^{(A)} + \sum_{q=1}^{D}\Phi_{(B):q}^{(V,i):j,k}[b]_q^{(B)} + \Phi^{(V,i):j,k}, \qquad (134)$$

$$[E(gW)]_{j,k}^{(O,i)} = \sum_{s=1}^{h}\sum_{p=1}^{D}\sum_{q=1}^{D}\Phi_{(QK,\tau^{-1}(s)):p,q}^{(O,i):j,k}[WW]_{p,q}^{(QK,s)}$$

$$+ \sum_{s=1}^{h}\sum_{p=1}^{D}\sum_{q=1}^{D}\Phi_{(VO,\tau^{-1}(s)):p,\pi_O^{-1}(q)}^{(O,i):j,k}[WW]_{p,q}^{(VO,s)}$$

$$+ \sum_{s=1}^{h}\sum_{p=1}^{D}\sum_{q=1}^{D_k}\Phi_{(Q,\tau^{-1}(s)):p,q}^{(O,i):j,k}\left[[W]^{(Q,s)}\cdot\left(M^{(s)}\right)^{\top}\right]_{p,q}$$

$$+ \sum_{s=1}^{h}\sum_{p=1}^{D}\sum_{q=1}^{D_k}\Phi_{(K,\tau^{-1}(s)):p,q}^{(O,i):j,k}\left[[W]^{(K,s)}\cdot\left(M^{(s)}\right)^{-1}\right]_{p,q}$$

$$+ \sum_{s=1}^{h}\sum_{p=1}^{D}\sum_{q=1}^{D_v}\Phi_{(V,\tau^{-1}(s)):p,q}^{(O,i):j,k}\left[[W]^{(V,s)}\cdot N^{(s)}\right]_{p,q}$$

$$+ \sum_{s=1}^{h}\sum_{p=1}^{D_v}\sum_{q=1}^{D}\Phi_{(O,\tau^{-1}(s)):p,\pi_O^{-1}(q)}^{(O,i):j,k}\left[\left(N^{(s)}\right)^{-1}\cdot[W]^{(O,s)}\right]_{p,q}$$

$$+ \sum_{p=1}^{D}\sum_{q=1}^{D_A}\Phi_{(A):\pi_O^{-1}(p),\pi_A^{-1}(q)}^{(O,i):j,k}[W]_{p,q}^{(A)} + \sum_{p=1}^{D_A}\sum_{q=1}^{D}\Phi_{(B):\pi_A^{-1}(p),q}^{(O,i):j,k}[W]_{p,q}^{(B)}$$

$$+ \sum_{q=1}^{D_A}\Phi_{(A):\pi_A^{-1}(q)}^{(O,i):j,k}[b]_q^{(A)} + \sum_{q=1}^{D}\Phi_{(B):q}^{(O,i):j,k}[b]_q^{(B)} + \Phi^{(O,i):j,k}, \qquad (135)$$

$$[E(gW)]_{j,k}^{(A)} = \sum_{s=1}^{h}\sum_{p=1}^{D}\sum_{q=1}^{D}\Phi_{(QK,\tau^{-1}(s)):p,q}^{(A):j,k}[WW]_{p,q}^{(QK,s)}$$

$$+ \sum_{s=1}^{h}\sum_{p=1}^{D}\sum_{q=1}^{D}\Phi_{(VO,\tau^{-1}(s)):p,\pi_O^{-1}(q)}^{(A):j,k}[WW]_{p,q}^{(VO,s)}$$

$$
+ \sum_{s=1}^{h} \sum_{p=1}^{D} \sum_{q=1}^{D_k} \Phi_{(Q,\tau^{-1}(s)):p,q}^{(A):j,k} \left[ [W]^{(Q,s)} \cdot \left( M^{(s)} \right)^{\top} \right]_{p,q}
$$

$$
+ \sum_{s=1}^{h} \sum_{p=1}^{D} \sum_{q=1}^{D_k} \Phi_{(K,\tau^{-1}(s)):p,q}^{(A):j,k} \left[ [W]^{(K,s)} \cdot \left( M^{(s)} \right)^{-1} \right]_{p,q}
$$

$$
+ \sum_{s=1}^{h} \sum_{p=1}^{D} \sum_{q=1}^{D_v} \Phi_{(V,\tau^{-1}(s)):p,q}^{(A):j,k} \left[ [W]^{(V,s)} \cdot N^{(s)} \right]_{p,q}
$$

$$
+ \sum_{s=1}^{h} \sum_{p=1}^{D_v} \sum_{q=1}^{D} \Phi_{(O,\tau^{-1}(s)):p,\pi_O^{-1}(q)}^{(A):j,k} \left[ \left( N^{(s)} \right)^{-1} \cdot [W]^{(O,s)} \right]_{p,q}
$$

$$
+ \sum_{p=1}^{D} \sum_{q=1}^{D_A} \Phi_{(A):\pi_O^{-1}(p),\pi_A^{-1}(q)}^{(A):j,k} [W]_{p,q}^{(A)} + \sum_{p=1}^{D_A} \sum_{q=1}^{D} \Phi_{(B):\pi_A^{-1}(p),q}^{(A):j,k} [W]_{p,q}^{(B)}
$$

$$
+ \sum_{q=1}^{D_A} \Phi_{(A):\pi_A^{-1}(q)}^{(A):j,k} [b]_q^{(A)} + \sum_{q=1}^{D} \Phi_{(B):q}^{(A):j,k} [b]_q^{(B)} + \Phi^{(A):j,k}, \qquad (136)
$$

$$
[E(gW)]_{j,k}^{(B)} = \sum_{s=1}^{h} \sum_{p=1}^{D} \sum_{q=1}^{D} \Phi_{(QK,\tau^{-1}(s)):p,q}^{(B):j,k} [WW]_{p,q}^{(QK,s)}
$$

$$
+ \sum_{s=1}^{h} \sum_{p=1}^{D} \sum_{q=1}^{D} \Phi_{(VO,\tau^{-1}(s)):p,\pi_O^{-1}(q)}^{(B):j,k} [WW]_{p,q}^{(VO,s)}
$$

$$
+ \sum_{s=1}^{h} \sum_{p=1}^{D} \sum_{q=1}^{D_k} \Phi_{(Q,\tau^{-1}(s)):p,q}^{(B):j,k} \left[ [W]^{(Q,s)} \cdot \left( M^{(s)} \right)^{\top} \right]_{p,q} \qquad (137)
$$

$$
+ \sum_{s=1}^{h} \sum_{p=1}^{D} \sum_{q=1}^{D_k} \Phi_{(K,\tau^{-1}(s)):p,q}^{(B):j,k} \left[ [W]^{(K,s)} \cdot \left( M^{(s)} \right)^{-1} \right]_{p,q}
$$

$$
+ \sum_{s=1}^{h} \sum_{p=1}^{D} \sum_{q=1}^{D_v} \Phi_{(V,\tau^{-1}(s)):p,q}^{(B):j,k} \left[ [W]^{(V,s)} \cdot N^{(s)} \right]_{p,q}
$$

$$
+ \sum_{s=1}^{h} \sum_{p=1}^{D_v} \sum_{q=1}^{D} \Phi_{(O,\tau^{-1}(s)):p,\pi_O^{-1}(q)}^{(B):j,k} \left[ \left( N^{(s)} \right)^{-1} \cdot [W]^{(O,s)} \right]_{p,q}
$$

$$
+ \sum_{p=1}^{D} \sum_{q=1}^{D_A} \Phi_{(A):\pi_O^{-1}(p),\pi_A^{-1}(q)}^{(B):j,k} [W]_{p,q}^{(A)} + \sum_{p=1}^{D_A} \sum_{q=1}^{D} \Phi_{(B):\pi_A^{-1}(p),q}^{(B):j,k} [W]_{p,q}^{(B)}
$$

$$
+ \sum_{q=1}^{D_A} \Phi_{(A):\pi_A^{-1}(q)}^{(B):j,k} [b]_q^{(A)} + \sum_{q=1}^{D} \Phi_{(B):q}^{(B):j,k} [b]_q^{(B)} + \Phi^{(B):j,k}, \qquad (138)
$$

$$
[E(gb)]_k^{(A)} = \sum_{s=1}^{h} \sum_{p=1}^{D} \sum_{q=1}^{D} \Phi_{(QK,\tau^{-1}(s)):p,q}^{(A):k} [WW]_{p,q}^{(QK,s)}
$$

$$
+ \sum_{s=1}^{h} \sum_{p=1}^{D} \sum_{q=1}^{D} \Phi_{(VO,\tau^{-1}(s)):p,\pi_O^{-1}(q)}^{(A):k} [WW]_{p,q}^{(VO,s)}
$$

$$
+ \sum_{s=1}^{h} \sum_{p=1}^{D} \sum_{q=1}^{D_k} \Phi_{(Q,\tau^{-1}(s)):p,q}^{(A):k} \left[ [W]^{(Q,s)} \cdot \left( M^{(s)} \right)^{\top} \right]_{p,q}
$$

$$
+ \sum_{s=1}^{h} \sum_{p=1}^{D} \sum_{q=1}^{D_k} \Phi_{(K,\tau^{-1}(s)):p,q}^{(A):k} \left[ [W]^{(K,s)} \cdot \left( M^{(s)} \right)^{-1} \right]_{p,q}
$$

$$+ \sum_{s=1}^{h} \sum_{p=1}^{D} \sum_{q=1}^{D_v} \Phi^{(A):k}_{(V,\tau^{-1}(s)):p,q} \left[ [W]^{(V,s)} \cdot N^{(s)} \right]_{p,q}$$

$$+ \sum_{s=1}^{h} \sum_{p=1}^{D_v} \sum_{q=1}^{D} \Phi^{(A):k}_{(O,\tau^{-1}(s)):p,\pi_O^{-1}(q)} \left[ \left( N^{(s)} \right)^{-1} \cdot [W]^{(O,s)} \right]_{p,q}$$

$$+ \sum_{p=1}^{D} \sum_{q=1}^{D_A} \Phi^{(A):k}_{(A):\pi_O^{-1}(p),\pi_A^{-1}(q)} [W]^{(A)}_{p,q} + \sum_{p=1}^{D_A} \sum_{q=1}^{D} \Phi^{(A):k}_{(B):\pi_A^{-1}(p),q} [W]^{(B)}_{p,q}$$

$$+ \sum_{q=1}^{D_A} \Phi^{(A):k}_{(A):\pi_A^{-1}(q)} [b]^{(A)}_q + \sum_{q=1}^{D} \Phi^{(A):k}_{(B):q} [b]^{(B)}_q + \Phi^{(A):k}, \tag{139}$$

$$[E(gb)]^{(B)}_k = \sum_{s=1}^{h} \sum_{p=1}^{D} \sum_{q=1}^{D} \Phi^{(B):k}_{(QK,\tau^{-1}(s)):p,q} [WW]^{(QK,s)}_{p,q}$$

$$+ \sum_{s=1}^{h} \sum_{p=1}^{D} \sum_{q=1}^{D} \Phi^{(B):k}_{(VO,\tau^{-1}(s)):p,\pi_O^{-1}(q)} [WW]^{(VO,s)}_{p,q}$$

$$+ \sum_{s=1}^{h} \sum_{p=1}^{D} \sum_{q=1}^{D_k} \Phi^{(B):k}_{(Q,\tau^{-1}(s)):p,q} \left[ [W]^{(Q,s)} \cdot \left( M^{(s)} \right)^{\top} \right]_{p,q}$$

$$+ \sum_{s=1}^{h} \sum_{p=1}^{D} \sum_{q=1}^{D_k} \Phi^{(B):k}_{(K,\tau^{-1}(s)):p,q} \left[ [W]^{(K,s)} \cdot \left( M^{(s)} \right)^{-1} \right]_{p,q}$$

$$+ \sum_{s=1}^{h} \sum_{p=1}^{D} \sum_{q=1}^{D_v} \Phi^{(B):k}_{(V,\tau^{-1}(s)):p,q} \left[ [W]^{(V,s)} \cdot N^{(s)} \right]_{p,q}$$

$$+ \sum_{s=1}^{h} \sum_{p=1}^{D_v} \sum_{q=1}^{D} \Phi^{(B):k}_{(O,\tau^{-1}(s)):p,\pi_O^{-1}(q)} \left[ \left( N^{(s)} \right)^{-1} \cdot [W]^{(O,s)} \right]_{p,q}$$

$$+ \sum_{p=1}^{D} \sum_{q=1}^{D_A} \Phi^{(B):k}_{(A):\pi_O^{-1}(p),\pi_A^{-1}(q)} [W]^{(A)}_{p,q} + \sum_{p=1}^{D_A} \sum_{q=1}^{D} \Phi^{(B):k}_{(B):\pi_A^{-1}(p),q} [W]^{(B)}_{p,q}$$

$$+ \sum_{q=1}^{D_A} \Phi^{(B):k}_{(A):\pi_A^{-1}(q)} [b]^{(A)}_q + \sum_{q=1}^{D} \Phi^{(B):k}_{(B):q} [b]^{(B)}_q + \Phi^{(B):k}. \tag{140}$$

### D.3.2 COMPUTING $gE(U)$.

According to Equation (86) and Equation (184), we have:

$$gE(U) = \left( \left( [gE(W)]^{(Q,i)}, [gE(W)]^{(K,i)}, [gE(W)]^{(V,i)}, [gE(W)]^{(O,i)} \right)_{i=1,\dots,h}, \right.$$
$$\left. \left( [gE(W)]^{(A)}, [gE(W)]^{(B)} \right), \left( [gE(b)]^{(A)}, [gE(b)]^{(B)} \right) \right),$$

where

$$[gE(W)]^{(Q,i)}_{j,k} = \left[ [E(W)]^{(Q,\tau(i))} \cdot \left( M^{(\tau(i))} \right)^{\top} \right]_{j,k},$$

$$[gE(W)]^{(K,i)}_{j,k} = \left[ [E(W)]^{(K,\tau(i))} \cdot \left( M^{(\tau(i))} \right)^{-1} \right]_{j,k},$$

$$[gE(W)]^{(V,i)}_{j,k} = \left[ [E(W)]^{(V,\tau(i))} \cdot N^{(\tau(i))} \right]_{j,k},$$

$$[gE(W)]^{(O,i)}_{j,k} = \left[ \left( N^{(\tau(i))} \right)^{-1} \cdot [E(W)]^{(O,\tau(i))} \right]_{j,\pi_O(k)},$$

$$[gE(W)]_{j,k}^{(A)} = [E(W)]_{\pi_O(j),\pi_A(k)}^{(A)},$$
$$[gE(W)]_{j,k}^{(B)} = [E(W)]_{\pi_A(j),k}^{(B)},$$
$$[gE(b)]_{k}^{(A)} = [E(b)]_{\pi_A(k)}^{(A)},$$
$$[gE(b)]_{k}^{(B)} = [E(b)]_{k}^{(B)}.$$

We calculate these entries in detail below.

$$[gE(W)]_{j,k}^{(Q,i)} = \left[ [E(W)]^{(Q,\tau(i))} \cdot \left( M^{(\tau(i))} \right)^{\top} \right]_{j,k}$$

$$= \sum_{l=1}^{D_k} [E(W)]_{j,l}^{(Q,\tau(i))} \cdot \left( M^{(\tau(i))} \right)_{l,k}^{\top}$$

$$= \sum_{l=1}^{D_k} M_{k,l}^{(\tau(i))} \cdot \left\{ \begin{array}{l} \sum_{s=1}^{h}\sum_{p=1}^{D}\sum_{q=1}^{D} \Phi_{(QK,s):p,q}^{(Q,\tau(i)):j,l}[WW]_{p,q}^{(QK,s)} + \sum_{s=1}^{h}\sum_{p=1}^{D}\sum_{q=1}^{D} \Phi_{(VO,s):p,q}^{(Q,\tau(i)):j,l}[WW]_{p,q}^{(VO,s)} \\[2mm] + \sum_{s=1}^{h}\sum_{p=1}^{D}\sum_{q=1}^{D_k} \Phi_{(Q,s):p,q}^{(Q,\tau(i)):j,l}[W]_{p,q}^{(Q,s)} + \sum_{s=1}^{h}\sum_{p=1}^{D}\sum_{q=1}^{D_k} \Phi_{(K,s):p,q}^{(Q,\tau(i)):j,l}[W]_{p,q}^{(K,s)} \\[2mm] + \sum_{s=1}^{h}\sum_{p=1}^{D}\sum_{q=1}^{D_v} \Phi_{(V,s):p,q}^{(Q,\tau(i)):j,l}[W]_{p,q}^{(V,s)} + \sum_{s=1}^{h}\sum_{p=1}^{D_v}\sum_{q=1}^{D} \Phi_{(O,s):p,q}^{(Q,\tau(i)):j,l}[W]_{p,q}^{(O,s)} \\[2mm] + \sum_{p=1}^{D}\sum_{q=1}^{D_A} \phi_{(A):p,q}^{(Q,\tau(i)):j,l}[W]_{p,q}^{(A)} + \sum_{p=1}^{D_A}\sum_{q=1}^{D} \Phi_{(B):p,q}^{(Q,\tau(i)):j,l}[W]_{p,q}^{(B)} \\[2mm] + \sum_{q=1}^{D_A} \Phi_{(A):q}^{(Q,\tau(i)):j,l}[b]_{q}^{(A)} + \sum_{q=1}^{D} \Phi_{(B):q}^{(Q,\tau(i)):j,l}[b]_{q}^{(B)} + \Phi^{(Q,\tau(i)):j,l} \end{array} \right\}.$$

$$[gE(W)]_{j,k}^{(K,i)} = \left[ [E(W)]^{(K,\tau(i))} \cdot \left( M^{(\tau(i))} \right)^{-1} \right]_{j,k}$$

$$= \sum_{l=1}^{D_k} [E(W)]_{j,l}^{(K,\tau(i))} \cdot \left( \left( M^{(\tau(i))} \right)^{-1} \right)_{l,k}$$

$$= \sum_{l=1}^{D_k} [E(W)]_{j,l}^{(K,\tau(i))} \cdot \left( \left( M^{(\tau(i))} \right)^{-1} \right)_{l,k} \tag{141}$$

$$= \sum_{l=1}^{D_k} \left( \left( M^{(\tau(i))} \right)^{-1} \right)_{l,k} \cdot \left\{ \begin{array}{l} \sum_{s=1}^{h}\sum_{p=1}^{D}\sum_{q=1}^{D} \Phi_{(QK,s):p,q}^{(K,\tau(i)):j,l}[WW]_{p,q}^{(QK,s)} + \sum_{s=1}^{h}\sum_{p=1}^{D}\sum_{q=1}^{D} \Phi_{(VO,s):p,q}^{(K,\tau(i)):j,l}[WW]_{p,q}^{(VO,s)} \\[2mm] + \sum_{s=1}^{h}\sum_{p=1}^{D}\sum_{q=1}^{D_k} \Phi_{(Q,s):p,q}^{(K,\tau(i)):j,l}[W]_{p,q}^{(Q,s)} + \sum_{s=1}^{h}\sum_{p=1}^{D}\sum_{q=1}^{D_k} \Phi_{(K,s):p,q}^{(K,\tau(i)):j,l}[W]_{p,q}^{(K,s)} \\[2mm] + \sum_{s=1}^{h}\sum_{p=1}^{D}\sum_{q=1}^{D_v} \Phi_{(V,s):p,q}^{(K,\tau(i)):j,l}[W]_{p,q}^{(V,s)} + \sum_{s=1}^{h}\sum_{p=1}^{D_v}\sum_{q=1}^{D} \Phi_{(O,s):p,q}^{(K,\tau(i)):j,l}[W]_{p,q}^{(O,s)} \\[2mm] + \sum_{p=1}^{D}\sum_{q=1}^{D_A} \phi_{(A):p,q}^{(K,\tau(i)):j,l}[W]_{p,q}^{(A)} + \sum_{p=1}^{D_A}\sum_{q=1}^{D} \Phi_{(B):p,q}^{(K,\tau(i)):j,l}[W]_{p,q}^{(B)} \\[2mm] + \sum_{q=1}^{D_A} \Phi_{(A):q}^{(K,\tau(i)):j,l}[b]_{q}^{(A)} + \sum_{q=1}^{D} \Phi_{(B):q}^{(K,\tau(i)):j,l}[b]_{q}^{(B)} + \Phi^{(K,\tau(i)):j,l} \end{array} \right\}.$$

$$[gE(W)]_{j,k}^{(V,i)} = \left[ [E(W)]^{(V,\tau(i))} \cdot N^{(\tau(i))} \right]_{j,k}, \tag{142}$$

$$= \sum_{l=1}^{D_v} [E(W)]_{j,l}^{(V,\tau(i))} \cdot N_{l,k}^{(\tau(i))} \tag{143}$$

$$= \sum_{l=1}^{D_v} N_{l,k}^{(\tau(i))} \cdot \left\{ \begin{array}{l} \sum_{s=1}^{h}\sum_{p=1}^{D}\sum_{q=1}^{D} \Phi_{(QK,s):p,q}^{(V,\tau(i)):j,l}[WW]_{p,q}^{(QK,s)} + \sum_{s=1}^{h}\sum_{p=1}^{D}\sum_{q=1}^{D} \Phi_{(VO,s):p,q}^{(V,\tau(i)):j,l}[WW]_{p,q}^{(VO,s)} \\[6pt] + \sum_{s=1}^{h}\sum_{p=1}^{D}\sum_{q=1}^{D_k} \Phi_{(Q,s):p,q}^{(V,\tau(i)):j,l}[W]_{p,q}^{(Q,s)} + \sum_{s=1}^{h}\sum_{p=1}^{D}\sum_{q=1}^{D_k} \Phi_{(K,s):p,q}^{(V,\tau(i)):j,l}[W]_{p,q}^{(K,s)} \\[6pt] + \sum_{s=1}^{h}\sum_{p=1}^{D}\sum_{q=1}^{D_v} \Phi_{(V,s):p,q}^{(V,\tau(i)):j,l}[W]_{p,q}^{(V,s)} + \sum_{s=1}^{h}\sum_{p=1}^{D_v}\sum_{q=1}^{D} \Phi_{(O,s):p,q}^{(V,\tau(i)):j,l}[W]_{p,q}^{(O,s)} \\[6pt] + \sum_{p=1}^{D}\sum_{q=1}^{D_A} \Phi_{(A):p,q}^{(V,\tau(i)):j,l}[W]_{p,q}^{(A)} + \sum_{p=1}^{D_A}\sum_{q=1}^{D} \Phi_{(B):p,q}^{(V,\tau(i)):j,l}[W]_{p,q}^{(B)} \\[6pt] + \sum_{q=1}^{D_A} \Phi_{(A):q}^{(V,\tau(i)):j,l}[b]_{q}^{(A)} + \sum_{q=1}^{D} \Phi_{(B):q}^{(V,\tau(i)):j,l}[b]_{q}^{(B)} + \Phi^{(V,\tau(i)):j,l} \end{array} \right\} .$$

$$[gE(W)]_{j,k}^{(O,i)} = \left[ \left( N^{(\tau(i))} \right)^{-1} \cdot [E(W)]^{(O,\tau(i))} \right]_{j,\pi_O(k)}, \tag{144}$$

$$= \sum_{l=1}^{D_v} \left( \left( N^{(\tau(i))} \right)^{-1} \right)_{j,l} \cdot [E(W)]_{l,\pi_O(k)}^{(O,\tau(i))} \tag{145}$$

$$= \sum_{l=1}^{D_v} \left( \left( N^{(\tau(i))} \right)^{-1} \right)_{j,l} \cdot \left\{ \begin{array}{l} \sum_{s=1}^{h}\sum_{p=1}^{D}\sum_{q=1}^{D} \Phi_{(QK,s):p,q}^{(O,\tau(i)):l,\pi_O(k)}[WW]_{p,q}^{(QK,s)} + \sum_{s=1}^{h}\sum_{p=1}^{D}\sum_{q=1}^{D} \Phi_{(VO,s):p,q}^{(O,\tau(i)):l,\pi_O(k)}[WW]_{p,q}^{(VO,s)} \\[6pt] + \sum_{s=1}^{h}\sum_{p=1}^{D}\sum_{q=1}^{D_k} \Phi_{(Q,s):p,q}^{(O,\tau(i)):l,\pi_O(k)}[W]_{p,q}^{(Q,s)} + \sum_{s=1}^{h}\sum_{p=1}^{D}\sum_{q=1}^{D_k} \Phi_{(K,s):p,q}^{(O,\tau(i)):l,\pi_O(k)}[W]_{p,q}^{(K,s)} \\[6pt] + \sum_{s=1}^{h}\sum_{p=1}^{D}\sum_{q=1}^{D_v} \Phi_{(V,s):p,q}^{(O,\tau(i)):l,\pi_O(k)}[W]_{p,q}^{(V,s)} + \sum_{s=1}^{h}\sum_{p=1}^{D_v}\sum_{q=1}^{D} \Phi_{(O,s):p,q}^{(O,\tau(i)):l,\pi_O(k)}[W]_{p,q}^{(O,s)} \\[6pt] + \sum_{p=1}^{D}\sum_{q=1}^{D_A} \Phi_{(A):p,q}^{(O,\tau(i)):l,\pi_O(k)}[W]_{p,q}^{(A)} + \sum_{p=1}^{D_A}\sum_{q=1}^{D} \Phi_{(B):p,q}^{(O,\tau(i)):l,\pi_O(k)}[W]_{p,q}^{(B)} \\[6pt] + \sum_{q=1}^{D_A} \Phi_{(A):q}^{(O,\tau(i)):l,\pi_O(k)}[b]_{q}^{(A)} + \sum_{q=1}^{D} \Phi_{(B):q}^{(O,\tau(i)):l,\pi_O(k)}[b]_{q}^{(B)} + \Phi^{(O,\tau(i)):l,\pi_O(k)} \end{array} \right\} .$$

$$[gE(W)]_{j,k}^{(A)} = [E(W)]_{\pi_O(j),\pi_A(k)}^{(A)}, \tag{146}$$

$$= \sum_{s=1}^{h}\sum_{p=1}^{D}\sum_{q=1}^{D} \Phi_{(QK,s):p,q}^{(A):\pi_O(j),\pi_A(k)}[WW]_{p,q}^{(QK,s)}$$

$$+ \sum_{s=1}^{h}\sum_{p=1}^{D}\sum_{q=1}^{D} \Phi_{(VO,s):p,q}^{(A):\pi_O(j),\pi_A(k)}[WW]_{p,q}^{(VO,s)}$$

$$+ \sum_{s=1}^{h}\sum_{p=1}^{D}\sum_{q=1}^{D_k} \Phi_{(Q,s):p,q}^{(A):\pi_O(j),\pi_A(k)}[W]_{p,q}^{(Q,s)} + \sum_{s=1}^{h}\sum_{p=1}^{D}\sum_{q=1}^{D_k} \Phi_{(K,s):p,q}^{(A):\pi_O(j),\pi_A(k)}[W]_{p,q}^{(K,s)}$$

$$+ \sum_{s=1}^{h}\sum_{p=1}^{D}\sum_{q=1}^{D_v} \Phi_{(V,s):p,q}^{(A):\pi_O(j),\pi_A(k)}[W]_{p,q}^{(V,s)} + \sum_{s=1}^{h}\sum_{p=1}^{D_v}\sum_{q=1}^{D} \Phi_{(O,s):p,q}^{(A):\pi_O(j),\pi_A(k)}[W]_{p,q}^{(O,s)}$$

$$+ \sum_{p=1}^{D}\sum_{q=1}^{D_A} \Phi_{(A):p,q}^{(A):\pi_O(j),\pi_A(k)}[W]_{p,q}^{(A)} + \sum_{p=1}^{D_A}\sum_{q=1}^{D} \Phi_{(B):p,q}^{(A):\pi_O(j),\pi_A(k)}[W]_{p,q}^{(B)}$$

$$+ \sum_{q=1}^{D_A} \Phi_{(A):q}^{(A):\pi_O(j),\pi_A(k)}[b]_{q}^{(A)} + \sum_{q=1}^{D} \Phi_{(B):q}^{(A):\pi_O(j),\pi_A(k)}[b]_{q}^{(B)}$$

$$+ \Phi^{(A):\pi_O(j),\pi_A(k)}. \tag{147}$$

$$[gE(W)]_{j,k}^{(B)} = [E(W)]_{\pi_A(j),k}^{(B)}, \tag{148}$$

$$= \sum_{s=1}^{h} \sum_{p=1}^{D} \sum_{q=1}^{D} \Phi_{(QK,s):p,q}^{(B):\pi_A(j),k} [WW]_{p,q}^{(QK,s)}$$

$$+ \sum_{s=1}^{h} \sum_{p=1}^{D} \sum_{q=1}^{D} \Phi_{(VO,s):p,q}^{(B):\pi_A(j),k} [WW]_{p,q}^{(VO,s)}$$

$$+ \sum_{s=1}^{h} \sum_{p=1}^{D} \sum_{q=1}^{D_k} \Phi_{(Q,s):p,q}^{(B):\pi_A(j),k} [W]_{p,q}^{(Q,s)} + \sum_{s=1}^{h} \sum_{p=1}^{D} \sum_{q=1}^{D_k} \Phi_{(K,s):p,q}^{(B):\pi_A(j),k} [W]_{p,q}^{(K,s)}$$

$$+ \sum_{s=1}^{h} \sum_{p=1}^{D} \sum_{q=1}^{D_v} \Phi_{(V,s):p,q}^{(B):\pi_A(j),k} [W]_{p,q}^{(V,s)} + \sum_{s=1}^{h} \sum_{p=1}^{D_v} \sum_{q=1}^{D} \Phi_{(O,s):p,q}^{(B):\pi_A(j),k} [W]_{p,q}^{(O,s)}$$

$$+ \sum_{p=1}^{D} \sum_{q=1}^{D_A} \Phi_{(A):p,q}^{(B):\pi_A(j),k} [W]_{p,q}^{(A)} + \sum_{p=1}^{D_A} \sum_{q=1}^{D} \Phi_{(B):p,q}^{(B):\pi_A(j),k} [W]_{p,q}^{(B)}$$

$$+ \sum_{q=1}^{D_A} \Phi_{(A):q}^{(B):\pi_A(j),k} [b]_q^{(A)} + \sum_{q=1}^{D} \Phi_{(B):q}^{(B):\pi_A(j),k} [b]_q^{(B)}$$

$$+ \Phi^{(B):\pi_A(j),k} \tag{149}$$

$$[gE(b)]_k^{(A)} = [E(b)]_{\pi_A(k)}^{(A)}, \tag{150}$$

$$= \sum_{s=1}^{h} \sum_{p=1}^{D} \sum_{q=1}^{D} \Phi_{(QK,s):p,q}^{(A):\pi_A(k)} [WW]_{p,q}^{(QK,s)}$$

$$+ \sum_{s=1}^{h} \sum_{p=1}^{D} \sum_{q=1}^{D} \Phi_{(VO,s):p,q}^{(A):\pi_A(k)} [WW]_{p,q}^{(VO,s)}$$

$$+ \sum_{s=1}^{h} \sum_{p=1}^{D} \sum_{q=1}^{D_k} \Phi_{(Q,s):p,q}^{(A):\pi_A(k)} [W]_{p,q}^{(Q,s)} + \sum_{s=1}^{h} \sum_{p=1}^{D} \sum_{q=1}^{D_k} \Phi_{(K,s):p,q}^{(A):\pi_A(k)} [W]_{p,q}^{(K,s)}$$

$$+ \sum_{s=1}^{h} \sum_{p=1}^{D} \sum_{q=1}^{D_v} \Phi_{(V,s):p,q}^{(A):\pi_A(k)} [W]_{p,q}^{(V,s)} + \sum_{s=1}^{h} \sum_{p=1}^{D_v} \sum_{q=1}^{D} \Phi_{(O,s):p,q}^{(A):\pi_A(k)} [W]_{p,q}^{(O,s)}$$

$$+ \sum_{p=1}^{D} \sum_{q=1}^{D_A} \phi_{(A):p,q}^{(A):\pi_A(k)} [W]_{p,q}^{(A)} + \sum_{p=1}^{D_A} \sum_{q=1}^{D} \Phi_{(B):p,q}^{(A):\pi_A(k)} [W]_{p,q}^{(B)}$$

$$+ \sum_{q=1}^{D_A} \Phi_{(A):q}^{(A):\pi_A(k)} [b]_q^{(A)} + \sum_{q=1}^{D} \Phi_{(B):q}^{(A):\pi_A(k)} [b]_q^{(B)} + \Phi^{(A):\pi_A(k)}. \tag{151}$$

$$[gE(b)]_k^{(B)} = [E(b)]_k^{(B)} \tag{152}$$

$$= \sum_{s=1}^{h} \sum_{p=1}^{D} \sum_{q=1}^{D} \Phi_{(QK,s):p,q}^{(B):k} [WW]_{p,q}^{(QK,s)}$$

$$+ \sum_{s=1}^{h} \sum_{p=1}^{D} \sum_{q=1}^{D} \Phi_{(VO,s):p,q}^{(B):k} [WW]_{p,q}^{(VO,s)}$$

$$+ \sum_{s=1}^{h} \sum_{p=1}^{D} \sum_{q=1}^{D_k} \Phi_{(Q,s):p,q}^{(B):k} [W]_{p,q}^{(Q,s)} + \sum_{s=1}^{h} \sum_{p=1}^{D} \sum_{q=1}^{D_k} \Phi_{(K,s):p,q}^{(B):k} [W]_{p,q}^{(K,s)}$$

$$+ \sum_{s=1}^{h}\sum_{p=1}^{D}\sum_{q=1}^{D_v} \Phi_{(V,s):p,q}^{(B):k} [W]_{p,q}^{(V,s)} + \sum_{s=1}^{h}\sum_{p=1}^{D_v}\sum_{q=1}^{D} \Phi_{(O,s):p,q}^{(B):k} [W]_{p,q}^{(O,s)}$$

$$+ \sum_{p=1}^{D}\sum_{q=1}^{D_A} \Phi_{(A):p,q}^{(B):k} [W]_{p,q}^{(A)} + \sum_{p=1}^{D_A}\sum_{q=1}^{D} \Phi_{(B):p,q}^{(B):k} [W]_{p,q}^{(B)}$$

$$+ \sum_{q=1}^{D_A} \Phi_{(A):q}^{(B):k} [b]_{q}^{(A)} + \sum_{q=1}^{D} \Phi_{(B):q}^{(B):k} [b]_{q}^{(B)} + \Phi^{(B):k}. \tag{153}$$

### D.3.3 COEFFICIENTS COMPARISON IN THE EQUATION $E(gU) = gE(U)$

Since $E(gU) = gE(U)$, we must have:

$$[E(gW)]^{(Q,i)} = [gE(W)]^{(Q,i)},$$
$$[E(gW)]^{(K,i)} = [gE(W)]^{(K,i)},$$
$$[E(gW)]^{(V,i)} = [gE(W)]^{(V,i)},$$
$$[E(gW)]^{(O,i)} = [gE(W)]^{(O,i)},$$
$$[E(gW)]^{(A)} = [gE(W)]^{(A)},$$
$$[E(gW)]^{(B)} = [gE(W)]^{(B)},$$
$$[E(gb)]^{(A)} = [gE(b)]^{(A)},$$
$$[E(gb)]^{(B)} = [gE(b)]^{(B)},$$

and these equalities hold for all $i = 1, \ldots, h$, $U \in \mathcal{U}$ and $g \in \mathcal{G}_{\mathcal{U}}$.

In the following, we will solve these equalities, one by one, to find the constraints of the parameters $\Phi^-$s of $E(U)$.

**Case 1. Solve the equation** $[E(gW)]^{(Q,i)} = [gE(W)]^{(Q,i)}$. For every $j, k$, we have

$$[E(gW)]_{j,k}^{(Q,i)} = [gE(W)]_{j,k}^{(Q,i)},$$

or equivalently,

$$\sum_{s=1}^{h}\sum_{p=1}^{D}\sum_{q=1}^{D} \Phi_{(QK,\tau^{-1}(s)):p,q}^{(Q,i):j,k} [WW]_{p,q}^{(QK,s)}$$

$$+ \sum_{s=1}^{h}\sum_{p=1}^{D}\sum_{q=1}^{D} \Phi_{(VO,\tau^{-1}(s)):p,\pi_O^{-1}(q)}^{(Q,i):j,k} [WW]_{p,q}^{(VO,s)}$$

$$+ \sum_{s=1}^{h}\sum_{p=1}^{D}\sum_{q=1}^{D_k} \Phi_{(Q,\tau^{-1}(s)):p,q}^{(Q,i):j,k} \left[ [W]^{(Q,s)} \cdot \left( M^{(s)} \right)^{\top} \right]_{p,q}$$

$$+ \sum_{s=1}^{h}\sum_{p=1}^{D}\sum_{q=1}^{D_k} \Phi_{(K,\tau^{-1}(s)):p,q}^{(Q,i):j,k} \left[ [W]^{(K,s)} \cdot \left( M^{(s)} \right)^{-1} \right]_{p,q}$$

$$+ \sum_{s=1}^{h}\sum_{p=1}^{D}\sum_{q=1}^{D_v} \Phi_{(V,\tau^{-1}(s)):p,q}^{(Q,i):j,k} \left[ [W]^{(V,s)} \cdot N^{(s)} \right]_{p,q}$$

$$+ \sum_{s=1}^{h}\sum_{p=1}^{D_v}\sum_{q=1}^{D} \Phi_{(O,\tau^{-1}(s)):p,\pi_O^{-1}(q)}^{(Q,i):j,k} \left[ \left( N^{(s)} \right)^{-1} \cdot [W]^{(O,s)} \right]_{p,q}$$

$$+ \sum_{p=1}^{D}\sum_{q=1}^{D_A} \Phi_{(A):\pi_O^{-1}(p),\pi_A^{-1}(q)}^{(Q,i):j,k} [W]_{p,q}^{(A)} + \sum_{p=1}^{D_A}\sum_{q=1}^{D} \Phi_{(B):\pi_A^{-1}(p),q}^{(Q,i):j,k} [W]_{p,q}^{(B)}$$

$$+ \sum_{q=1}^{D_A} \Phi_{(A):\pi_A^{-1}(q)}^{(Q,i):j,k} [b]_{q}^{(A)} + \sum_{q=1}^{D} \Phi_{(B):q}^{(Q,i):j,k} [b]_{q}^{(B)} + \Phi^{(Q,i):j,k}$$

$$= \sum_{l=1}^{D_k} [E(W)]_{j,l}^{(Q,\tau(i))} \cdot \left( M^{(\tau(i))} \right)_{l,k}^{\top}. \tag{154}$$

Observe that the right hand side of the above equation is just a linear function on $M^{(\tau(i))}$. By applying Corollary D.3, we see that all terms of the left hand side are equal to zero, except those containing $M^{(\tau(i))}$ will be identically equal to the right hand side, i.e.

$$\sum_{s=1}^{h}\sum_{p=1}^{D}\sum_{q=1}^{D} \Phi_{(QK,\tau^{-1}(s)):p,q}^{(Q,i):j,k} [WW]_{p,q}^{(QK,s)}$$

$$+ \sum_{s=1}^{h}\sum_{p=1}^{D}\sum_{q=1}^{D} \Phi_{(VO,\tau^{-1}(s)):p,\pi_O^{-1}(q)}^{(Q,i):j,k} [WW]_{p,q}^{(VO,s)}$$

$$+ \sum_{\substack{s=1 \\ s\neq\tau(i)}}^{h}\sum_{p=1}^{D}\sum_{q=1}^{D_k} \Phi_{(Q,\tau^{-1}(s)):p,q}^{(Q,i):j,k} \left[ [W]^{(Q,s)} \cdot \left( M^{(s)} \right)^{\top} \right]_{p,q}$$

$$+ \sum_{s=1}^{h}\sum_{p=1}^{D}\sum_{q=1}^{D_k} \Phi_{(K,\tau^{-1}(s)):p,q}^{(Q,i):j,k} \left[ [W]^{(K,s)} \cdot \left( M^{(s)} \right)^{-1} \right]_{p,q}$$

$$+ \sum_{s=1}^{h}\sum_{p=1}^{D}\sum_{q=1}^{D_v} \Phi_{(V,\tau^{-1}(s)):p,q}^{(Q,i):j,k} \left[ [W]^{(V,s)} \cdot N^{(s)} \right]_{p,q}$$

$$+ \sum_{s=1}^{h}\sum_{p=1}^{D_v}\sum_{q=1}^{D} \Phi_{(O,\tau^{-1}(s)):p,\pi_O^{-1}(q)}^{(Q,i):j,k} \left[ \left( N^{(s)} \right)^{-1} \cdot [W]^{(O,s)} \right]_{p,q}$$

$$+ \sum_{p=1}^{D}\sum_{q=1}^{D_A} \Phi_{(A):\pi_O^{-1}(p),\pi_A^{-1}(q)}^{(Q,i):j,k} [W]_{p,q}^{(A)} + \sum_{p=1}^{D_A}\sum_{q=1}^{D} \Phi_{(B):\pi_A^{-1}(p),q}^{(Q,i):j,k} [W]_{p,q}^{(B)}$$

$$+ \sum_{q=1}^{D_A} \Phi_{(A):\pi_A^{-1}(q)}^{(Q,i):j,k} [b]_q^{(A)} + \sum_{q=1}^{D} \Phi_{(B):q}^{(Q,i):j,k} [b]_q^{(B)} + \Phi^{(Q,i):j,k}$$

$$= 0, \tag{155}$$

and

$$\sum_{p=1}^{D}\sum_{q=1}^{D_k} \Phi_{(Q,i):p,q}^{(Q,i):j,k} \left[ [W]^{(Q,\tau(i))} \cdot \left( M^{(\tau(i))} \right)^{\top} \right]_{p,q} = \sum_{l=1}^{D_k} [E(W)]_{j,l}^{(Q,\tau(i))} \cdot \left( M^{(\tau(i))} \right)_{l,k}^{\top}. \tag{156}$$

We choose $g$ to be the identity of $\mathcal{G}_{\mathcal{U}}$ (i.e. $\tau$, $\pi_O$, $\pi_A$, $M^{(s)}$ and $N^{(s)}$ are all identities) in Equation (155), then substituting the obtained result into Equation (102), we obtain a more compact formula for $[E(W)]_{j,k}^{(Q,i)}$ as follows:

$$[E(W)]_{j,k}^{(Q,i)} = \sum_{p=1}^{D}\sum_{q=1}^{D_k} \Phi_{(Q,i):p,q}^{(Q,i):j,k} [W]_{p,q}^{(Q,i)}. \tag{157}$$

Then by substituting this compact formula of $[E(W)]_{j,k}^{(Q,i)}$ to Equation (156), we obtain the first constraints:

$$\sum_{p=1}^{D}\sum_{q=1}^{D_k} \Phi_{(Q,i):p,q}^{(Q,i):j,k} \left[ [W]^{(Q,\tau(i))} \cdot \left( M^{(\tau(i))} \right)^{\top} \right]_{p,q}$$

$$= \sum_{l=1}^{D_k} \left( \sum_{p=1}^{D}\sum_{q=1}^{D_k} \Phi_{(Q,\tau(i)):p,q}^{(Q,\tau(i)):j,l} [W]_{p,q}^{(Q,\tau(i))} \right) \cdot M_{k,l}^{(\tau(i))},$$

or equivalently,

$$\sum_{p=1}^{D}\sum_{q=1}^{D_k} \Phi_{(Q,\tau(i)):p,q}^{(Q,\tau(i)):j,k} \left[ [W]^{(Q,i)} \cdot \left( M^{(i)} \right)^{\top} \right]_{p,q}$$

$$= \sum_{l=1}^{D_k} \left( \sum_{p=1}^{D} \sum_{q=1}^{D_k} \Phi_{(Q,i):p,q}^{(Q,i):j,l} [W]_{p,q}^{(Q,i)} \right) \cdot M_{k,l}^{(i)}, \tag{158}$$

We view the left hand side of Equation (158) as a sum of linear functions on the $q$-th rows of $M^{(i)}$ with $q = 1, \ldots, D_k$, while the right hand side is just a linear function on the $k$-th row of $M^{(i)}$. The by applying Corollary D.3, we see that all of the coefficients $\Phi_{(Q,\tau(i)):p,q}^{(Q,\tau(i)):j,k}$ of the left side are equal to zero, except those with the indices $k = q$. Therefore, we can reduce the formula for $[E(W)]^{(Q,i)}$ and the constraint further as follows:

$$[E(W)]_{j,k}^{(Q,i)} = \sum_{p=1}^{D} \Phi_{(Q,i):p,k}^{(Q,i):j,k} [W]_{p,k}^{(Q,i)}. \tag{159}$$

To find the constraints for the coefficients, we observe that, the left hand side of Equation (158) depends on $\tau$, while the right hand side is not. Therefore, by varying $\tau$, the left hand side does not change. As a consequence, we must have

$$\Phi_{(Q,\tau(i)):p,k}^{(Q,\tau(i)):j,k} = \Phi_{(Q,i):p,k}^{(Q,i):j,k}, \tag{160}$$

for all $i, j, p, k$ and $\tau$.

Next, by substituting Equation (159) and Equation (160) to Equation (156), we obtain:

$$\sum_{p=1}^{D} \Phi_{(Q,i):p,k}^{(Q,i):j,k} \left( \sum_{l=1}^{D_k} [W]_{p,l}^{(Q,i)} \cdot M_{k,l}^{(i)} \right) = \sum_{l=1}^{D_k} \left( \sum_{p=1}^{D} \Phi_{(Q,i):p,l}^{(Q,i):j,l} [W]_{p,l}^{(Q,i)} \right) \cdot M_{k,l}^{(i)}. \tag{161}$$

By comparing the corresponding coefficients both sides, we finally obtain the constraints

$$\Phi_{(Q,\tau(i)):p,k}^{(Q,\tau(i)):j,k} = \Phi_{(Q,i):p,l}^{(Q,i):j,l}, \tag{162}$$

for all $i, j, p, k, l$ and $\tau$.

One can verify directly that $[E(W)]_{j,k}^{(K,i)}$ defined in the form

$$[E(W)]_{j,k}^{(Q,i)} = \sum_{p=1}^{D} \Phi_{(Q,i):p,k}^{(Q,i):j,k} [W]_{p,k}^{(Q,i)}, \tag{163}$$

with the constraints

$$\Phi_{(Q,i):p,k}^{(Q,i):j,k} = \Phi_{(Q,\tau(i)):p,k'}^{(Q,\tau(i)):j,k'}, \tag{164}$$

for all $i, j, p, k, k', \tau$, satisfies the constraint in Equation (156).

**Case 2. Solve the equation $[E(gW)]^{(K,i)} = [gE(W)]^{(K,i)}$.** By using the same argument, we also obtain $[E(W)]_{j,k}^{(K,i)}$ in the form

$$[E(W)]_{j,k}^{(K,i)} = \sum_{p=1}^{D} \Phi_{(K,i):p,k}^{(K,i):j,k} [W]_{p,k}^{(K,i)}, \tag{165}$$

with the constraints

$$\Phi_{(K,i):p,k}^{(K,i):j,k} = \Phi_{(K,\tau(i)):p,k'}^{(K,\tau(i)):j,k'}, \tag{166}$$

for all $i, j, p, k, k', \tau$.

**Case 3. Solve the equation $[E(gW)]^{(V,i)} = [gE(W)]^{(V,i)}$.** By using the same argument, we also obtain $[E(W)]_{j,k}^{(V,i)}$ in the form

$$[E(W)]_{j,k}^{(V,i)} = \sum_{p=1}^{D} \Phi_{(V,i):p,k}^{(V,i):j,k} [W]_{p,k}^{(V,i)}, \tag{167}$$

with the constraints

$$\Phi_{(V,i):p,k}^{(V,i):j,k} = \Phi_{(V,\tau(i)):p,k'}^{(V,\tau(i)):j,k'}, \tag{168}$$

for all $i, j, p, k, k', \tau$.

**Case 4. Solve the equation** $[E(gW)]^{(O,i)} = [gE(W)]^{(O,i)}$. Since
$$[E(gW)]^{(O,i)}_{j,k} = [gE(W)]^{(O,i)}_{j,k},$$
it follows from Equation (105) that

$$\sum_{s=1}^{h}\sum_{p=1}^{D}\sum_{q=1}^{D}\Phi^{(O,i):j,k}_{(QK,\tau^{-1}(s)):p,q}[WW]^{(QK,s)}_{p,q}$$

$$+\sum_{s=1}^{h}\sum_{p=1}^{D}\sum_{q=1}^{D}\Phi^{(O,i):j,k}_{(VO,\tau^{-1}(s)):p,\pi_O^{-1}(q)}[WW]^{(VO,s)}_{p,q}$$

$$+\sum_{s=1}^{h}\sum_{p=1}^{D}\sum_{q=1}^{D_k}\Phi^{(O,i):j,k}_{(Q,\tau^{-1}(s)):p,q}\left[[W]^{(Q,s)}\cdot\left(M^{(s)}\right)^{\top}\right]_{p,q}$$

$$+\sum_{s=1}^{h}\sum_{p=1}^{D}\sum_{q=1}^{D_k}\Phi^{(O,i):j,k}_{(K,\tau^{-1}(s)):p,q}\left[[W]^{(K,s)}\cdot\left(M^{(s)}\right)^{-1}\right]_{p,q}$$

$$+\sum_{s=1}^{h}\sum_{p=1}^{D}\sum_{q=1}^{D_v}\Phi^{(O,i):j,k}_{(V,\tau^{-1}(s)):p,q}\left[[W]^{(V,s)}\cdot N^{(s)}\right]_{p,q}$$

$$+\sum_{s=1}^{h}\sum_{p=1}^{D_v}\sum_{q=1}^{D}\Phi^{(O,i):j,k}_{(O,\tau^{-1}(s)):p,\pi_O^{-1}(q)}\left[\left(N^{(s)}\right)^{-1}\cdot[W]^{(O,s)}\right]_{p,q}$$

$$+\sum_{p=1}^{D}\sum_{q=1}^{D_A}\Phi^{(O,i):j,k}_{(A):\pi_O^{-1}(p),\pi_A^{-1}(q)}[W]^{(A)}_{p,q}+\sum_{p=1}^{D_A}\sum_{q=1}^{D}\Phi^{(O,i):j,k}_{(B):\pi_A^{-1}(p),q}[W]^{(B)}_{p,q}$$

$$+\sum_{q=1}^{D_A}\Phi^{(O,i):j,k}_{(A):\pi_A^{-1}(q)}[b]^{(A)}_q+\sum_{q=1}^{D}\Phi^{(O,i):j,k}_{(B):q}[b]^{(B)}_q+\Phi^{(O,i):j,k}$$

$$=\sum_{l=1}^{D_v}\left(\left(N^{(\tau(i))}\right)^{-1}\right)_{j,l}\cdot[E(W)]^{(O,\tau(i))}_{l,\pi_O(k)}.\tag{169}$$

The right hand side of the above equation is just a linear function on $\left(N^{(\tau(i))}\right)^{-1}$. Therefore, it follows from Corollary D.3 that all terms of the left hand side are equal to zero, except those containing $\left(N^{(\tau(i))}\right)^{-1}$ will be identically equal to the right hand side, i.e.

$$\sum_{s=1}^{h}\sum_{p=1}^{D}\sum_{q=1}^{D}\Phi^{(O,i):j,k}_{(QK,\tau^{-1}(s)):p,q}[WW]^{(QK,s)}_{p,q}$$

$$+\sum_{s=1}^{h}\sum_{p=1}^{D}\sum_{q=1}^{D}\Phi^{(O,i):j,k}_{(VO,\tau^{-1}(s)):p,\pi_O^{-1}(q)}[WW]^{(VO,s)}_{p,q}$$

$$+\sum_{s=1}^{h}\sum_{p=1}^{D}\sum_{q=1}^{D_k}\Phi^{(O,i):j,k}_{(Q,\tau^{-1}(s)):p,q}\left[[W]^{(Q,s)}\cdot\left(M^{(s)}\right)^{\top}\right]_{p,q}$$

$$+\sum_{s=1}^{h}\sum_{p=1}^{D}\sum_{q=1}^{D_k}\Phi^{(O,i):j,k}_{(K,\tau^{-1}(s)):p,q}\left[[W]^{(K,s)}\cdot\left(M^{(s)}\right)^{-1}\right]_{p,q}$$

$$+\sum_{s=1}^{h}\sum_{p=1}^{D}\sum_{q=1}^{D_v}\Phi^{(O,i):j,k}_{(V,\tau^{-1}(s)):p,q}\left[[W]^{(V,s)}\cdot N^{(s)}\right]_{p,q}$$

$$+\sum_{\substack{s=1\\s\neq\tau(i)}}^{h}\sum_{p=1}^{D_v}\sum_{q=1}^{D}\Phi^{(O,i):j,k}_{(O,\tau^{-1}(s)):p,\pi_O^{-1}(q)}\left[\left(N^{(s)}\right)^{-1}\cdot[W]^{(O,s)}\right]_{p,q}$$

$$+ \sum_{p=1}^{D} \sum_{q=1}^{D_A} \Phi^{(O,i):j,k}_{(A):\pi_O^{-1}(p),\pi_A^{-1}(q)} [W]^{(A)}_{p,q} + \sum_{p=1}^{D_A} \sum_{q=1}^{D} \Phi^{(O,i):j,k}_{(B):\pi_A^{-1}(p),q} [W]^{(B)}_{p,q}$$

$$+ \sum_{q=1}^{D_A} \Phi^{(O,i):j,k}_{(A):\pi_A^{-1}(q)} [b]^{(A)}_q + \sum_{q=1}^{D} \Phi^{(O,i):j,k}_{(B):q} [b]^{(B)}_q + \Phi^{(O,i):j,k}$$

$$= 0. \tag{170}$$

By choosing $g$ to be the identity element of $\mathcal{G}_{\mathcal{U}}$ in Equation (170), and substituting the obtained result to Equation (184), we obtain a shorter formula for $[E(W)]^{(O,i)}_{j,k}$ as follow:

$$[E(W)]^{(O,i)}_{j,k} = \sum_{p=1}^{D_v} \sum_{q=1}^{D} \Phi^{(O,i):j,k}_{(O,i):p,q} [W]^{(O,i)}_{p,q}. \tag{171}$$

Thus, the constraint becomes

$$\sum_{p=1}^{D_v} \sum_{q=1}^{D} \Phi^{(O,i):j,k}_{(O,i):p,\pi_O^{-1}(q)} \left[ \left( N^{(\tau(i))} \right)^{-1} \cdot [W]^{(O,\tau(i))} \right]_{p,q}$$

$$= \sum_{l=1}^{D_v} \left( \left( N^{(\tau(i))} \right)^{-1} \right)_{j,l} \cdot [E(W)]^{(O,\tau(i))}_{l,\pi_O(k)},$$

or equivalently,

$$\sum_{p=1}^{D_v} \sum_{q=1}^{D} \Phi^{(O,\tau(i)):j,\pi_O(k)}_{(O,\tau(i)):p,\pi_O(q)} \left[ \left( N^{(i)} \right)^{-1} \cdot [W]^{(O,i)} \right]_{p,q}$$

$$= \sum_{l=1}^{D_v} \left( \left( N^{(i)} \right)^{-1} \right)_{j,l} \cdot [E(W)]^{(O,i)}_{l,k}. \tag{172}$$

Observe that, the left hand side of Equation (172) is just a linear function on the $j$-th row of $\left( N^{(i)} \right)^{-1}$, while the left hand side is a sum of linear combination of $p$-th rows of $\left( N^{(i)} \right)^{-1}$ with $p = 1, \ldots, D_v$. Therefore, only coefficients of $[E(W)]^{(O,i)}_{j,k}$ with $p = j$ are nonzero, i.e

$$[E(W)]^{(O,i)}_{j,k} = \sum_{q=1}^{D} \Phi^{(O,i):j,k}_{(O,i):j,q} [W]^{(O,i)}_{j,q}. \tag{173}$$

In addition, since the right hand side of Equation (172) is independent of $\tau$ and $\pi_O$, the left hand side must remain unchange if we vary $\tau$ and $\pi_O$. As a consequence, we have

$$\Phi^{(O,\tau(i)):j,\pi_O(k)}_{(O,\tau(i)):j,\pi_O(q)} = \Phi^{(O,i):j,k}_{(O,i):j,q} \tag{174}$$

for all $i, j, k, q, \tau$ and $\pi_O$. Next, by substituting Equation (173) and Equation (174) to Equation (172), we obtain

$$\sum_{q=1}^{D} \Phi^{(O,\tau(i)):j,\pi_O(k)}_{(O,\tau(i)):j,\pi_O(q)} \left[ \left( N^{(i)} \right)^{-1} \cdot [W]^{(O,i)} \right]_{j,q}$$

$$= \sum_{l=1}^{D_v} \left( \left( N^{(i)} \right)^{-1} \right)_{j,l} \cdot \left( \sum_{q=1}^{D} \Phi^{(O,i):l,k}_{(O,i):l,q} [W]^{(O,i)}_{l,q} \right),$$

or equivalently,

$$\sum_{q=1}^{D} \sum_{l=1}^{D_v} \Phi^{(O,\tau(i)):j,\pi_O(k)}_{(O,\tau(i)):j,\pi_O(q)} \left( \left( N^{(i)} \right)^{-1} \right)_{j,l} \cdot [W]^{(O,i)}_{l,q}$$

$$= \sum_{q=1}^{D} \sum_{l=1}^{D_v} \Phi^{(O,i):l,k}_{(O,i):l,q} \left( \left( N^{(i)} \right)^{-1} \right)_{j,l} \cdot [W]^{(O,i)}_{l,q}. \tag{175}$$

By comparing corresponding coefficients both sides, we have

$$\Phi^{(O,\tau(i)):j,\pi_O(k)}_{(O,\tau(i)):j,\pi_O(q)} = \Phi^{(O,i):l,k}_{(O,i):l,q}. \tag{176}$$

Finally, one can verify directly that $[E(W)]_{j,k}^{(O,i)}$ defined as $[E(W)]_{j,k}^{(O,i)}$ with $p = j$ are nonzero, i.e

$$[E(W)]_{j,k}^{(O,i)} = \sum_{q=1}^{D} \Phi_{(O,i):j,q}^{(O,i):j,k}[W]_{j,q}^{(O,i)}, \tag{177}$$

with the constraints

$$\Phi_{(O,i):j,q}^{(O,i):j,k} = \Phi_{(O,\tau(i)):j',\pi_O(q)}^{(O,\tau(i)):j',\pi_O(k)} \tag{178}$$

satisfy the constraint in Equation (172) for all $i, j, j', k, q$ and $\tau, \pi_O$.

**Case 5. Solve the equation** $[E(gW)]^{(A)} = [gE(W)]^{(A)}$**.** Since $[E(gW)]_{j,k}^{(A)} = [gE(W)]_{j,k}^{(A)}$, we have

$$\sum_{s=1}^{h}\sum_{p=1}^{D}\sum_{q=1}^{D} \Phi_{(QK,\tau^{-1}(s)):p,q}^{(A):j,k}[WW]_{p,q}^{(QK,s)}$$

$$+\sum_{s=1}^{h}\sum_{p=1}^{D}\sum_{q=1}^{D} \Phi_{(VO,\tau^{-1}(s)):p,\pi_O^{-1}(q)}^{(A):j,k}[WW]_{p,q}^{(VO,s)}$$

$$+\sum_{s=1}^{h}\sum_{p=1}^{D}\sum_{q=1}^{D_k} \Phi_{(Q,\tau^{-1}(s)):p,q}^{(A):j,k}\left[[W]^{(Q,s)}\cdot\left(M^{(s)}\right)^{\top}\right]_{p,q}$$

$$+\sum_{s=1}^{h}\sum_{p=1}^{D}\sum_{q=1}^{D_k} \Phi_{(K,\tau^{-1}(s)):p,q}^{(A):j,k}\left[[W]^{(K,s)}\cdot\left(M^{(s)}\right)^{-1}\right]_{p,q}$$

$$+\sum_{s=1}^{h}\sum_{p=1}^{D}\sum_{q=1}^{D_v} \Phi_{(V,\tau^{-1}(s)):p,q}^{(A):j,k}\left[[W]^{(V,s)}\cdot N^{(s)}\right]_{p,q}$$

$$+\sum_{s=1}^{h}\sum_{p=1}^{D_v}\sum_{q=1}^{D} \Phi_{(O,\tau^{-1}(s)):p,\pi_O^{-1}(q)}^{(A):j,k}\left[\left(N^{(s)}\right)^{-1}\cdot[W]^{(O,s)}\right]_{p,q}$$

$$+\sum_{p=1}^{D}\sum_{q=1}^{D_A} \Phi_{(A):\pi_O^{-1}(p),\pi_A^{-1}(q)}^{(A):j,k}[W]_{p,q}^{(A)} + \sum_{p=1}^{D_A}\sum_{q=1}^{D} \Phi_{(B):\pi_A^{-1}(p),q}^{(A):j,k}[W]_{p,q}^{(B)}$$

$$+\sum_{q=1}^{D_A} \Phi_{(A):\pi_A^{-1}(q)}^{(A):j,k}[b]_q^{(A)} + \sum_{q=1}^{D} \Phi_{(B):q}^{(A):j,k}[b]_q^{(B)} + \Phi^{(A):j,k}$$

$$=\sum_{s=1}^{h}\sum_{p=1}^{D}\sum_{q=1}^{D} \Phi_{(QK,s):p,q}^{(A):\pi_O(j),\pi_A(k)}[WW]_{p,q}^{(QK,s)}$$

$$+\sum_{s=1}^{h}\sum_{p=1}^{D}\sum_{q=1}^{D} \Phi_{(VO,s):p,q}^{(A):\pi_O(j),\pi_A(k)}[WW]_{p,q}^{(VO,s)}$$

$$+\sum_{s=1}^{h}\sum_{p=1}^{D}\sum_{q=1}^{D_k} \Phi_{(Q,s):p,q}^{(A):\pi_O(j),\pi_A(k)}[W]_{p,q}^{(Q,s)} + \sum_{s=1}^{h}\sum_{p=1}^{D}\sum_{q=1}^{D_k} \Phi_{(K,s):p,q}^{(A):\pi_O(j),\pi_A(k)}[W]_{p,q}^{(K,s)}$$

$$+\sum_{s=1}^{h}\sum_{p=1}^{D}\sum_{q=1}^{D_v} \Phi_{(V,s):p,q}^{(A):\pi_O(j),\pi_A(k)}[W]_{p,q}^{(V,s)} + \sum_{s=1}^{h}\sum_{p=1}^{D_v}\sum_{q=1}^{D} \Phi_{(O,s):p,q}^{(A):\pi_O(j),\pi_A(k)}[W]_{p,q}^{(O,s)}$$

$$+\sum_{p=1}^{D}\sum_{q=1}^{D_A} \Phi_{(A):p,q}^{(A):\pi_O(j),\pi_A(k)}[W]_{p,q}^{(A)} + \sum_{p=1}^{D_A}\sum_{q=1}^{D} \Phi_{(B):p,q}^{(A):\pi_O(j),\pi_A(k)}[W]_{p,q}^{(B)}$$

$$+\sum_{q=1}^{D_A} \Phi_{(A):q}^{(A):\pi_O(j),\pi_A(k)}[b]_q^{(A)} + \sum_{q=1}^{D} \Phi_{(B):q}^{(A):\pi_O(j),\pi_A(k)}[b]_q^{(B)} + \Phi^{(A):\pi_O(j),\pi_A(k)}. \tag{179}$$

We observe that the left hand side is a sum of linear functions on the matrices $M^{(s)}$, $\left(M^{(s)}\right)^{-1}$, $N^{(s)}$, and $\left(N^{(s)}\right)^{-1}$, while the right hand side is independent of these matrices. Therefore, according to Corollary D.3, the terms containing these matrices in the left hand side must be equal to zero. For the remaining terms, we can use Lemma D.1 to compare the corresponding ones with those in the right hand side and obtain the constraints:

$$\Phi^{(A):j,k}_{(QK,\tau^{-1}(s)):p,q} = \Phi^{(A):\pi_O(j),\pi_A(k)}_{(QK,s):p,q},$$

$$\Phi^{(A):j,k}_{(VO,\tau^{-1}(s)):p,\pi_O^{-1}(q)} = \Phi^{(A):\pi_O(j),\pi_A(k)}_{(VO,s):p,q},$$

$$\Phi^{(A):\pi_O(j),\pi_A(k)}_{(Q,s):p,q} = 0,$$

$$\Phi^{(A):\pi_O(j),\pi_A(k)}_{(K,s):p,q} = 0,$$

$$\Phi^{(A):\pi_O(j),\pi_A(k)}_{(V,s):p,q} = 0,$$

$$\Phi^{(A):\pi_O(j),\pi_A(k)}_{(O,s):p,q} = 0,$$

$$\Phi^{(A):j,k}_{(A):\pi_O^{-1}(p),\pi_A^{-1}(q)} = \Phi^{(A):\pi_O(j),\pi_A(k)}_{(A):p,q},$$

$$\Phi^{(A):j,k}_{(B):\pi_A^{-1}(p),q} = \Phi^{(A):\pi_O(j),\pi_A(k)}_{(B):p,q},$$

$$\Phi^{(A):j,k}_{(A):\pi_A^{-1}(q)} = \Phi^{(A):\pi_O(j),\pi_A(k)}_{(A):q},$$

$$\Phi^{(A):j,k}_{(B):q} = \Phi^{(A):\pi_O(j),\pi_A(k)}_{(B):q},$$

$$\Phi^{(A):j,k} = \Phi^{(A):\pi_O(j),\pi_A(k)}.$$

Therefore,

$$\Phi^{(A):j,k}_{(QK,s):p,q} = \Phi^{(A):\pi_O(j),\pi_A(k)}_{(QK,\tau(s)):p,q},$$

$$\Phi^{(A):j,k}_{(VO,s):p,q} = \Phi^{(A):\pi_O(j),\pi_A(k)}_{(VO,\tau(s)):p,\pi_O(q)},$$

$$\Phi^{(A):j,k}_{(Q,s):p,q} = 0,$$

$$\Phi^{(A):j,k}_{(K,s):p,q} = 0,$$

$$\Phi^{(A):j,k}_{(V,s):p,q} = 0,$$

$$\Phi^{(A):j,k}_{(O,s):p,q} = 0,$$

$$\Phi^{(A):j,k}_{(A):p,q} = \Phi^{(A):\pi_O(j),\pi_A(k)}_{(A):\pi_O(p),\pi_A(q)},$$

$$\Phi^{(A):j,k}_{(B):p,q} = \Phi^{(A):\pi_O(j),\pi_A(k)}_{(B):\pi_A(p),q},$$

$$\Phi^{(A):j,k}_{(A):q} = \Phi^{(A):\pi_O(j),\pi_A(k)}_{(A):\pi_A(q)},$$

$$\Phi^{(A):j,k}_{(B):q} = \Phi^{(A):\pi_O(j),\pi_A(k)}_{(B):q},$$

$$\Phi^{(A):j,k} = \Phi^{(A):\pi_O(j),\pi_A(k)}.$$

One can verify directly that $[E(W)]^{(A)}_{j,k}$ defined by

$$[E(W)]^{(A)}_{j,k} = \sum_{s=1}^{h}\sum_{p=1}^{D}\sum_{q=1}^{D} \Phi^{(A):j,k}_{(QK,s):p,q}[WW]^{(QK,s)}_{p,q}$$

$$+ \sum_{s=1}^{h}\sum_{p=1}^{D}\sum_{q=1}^{D} \Phi^{(A):j,k}_{(VO,s):p,q}[WW]^{(VO,s)}_{p,q}$$

$$+ \sum_{p=1}^{D}\sum_{q=1}^{D_A} \Phi^{(A):j,k}_{(A):p,q}[W]^{(A)}_{p,q} + \sum_{p=1}^{D_A}\sum_{q=1}^{D} \Phi^{(A):j,k}_{(B):p,q}[W]^{(B)}_{p,q}$$

$$+ \sum_{q=1}^{D_A} \Phi^{(A):j,k}_{(A):q}[b]^{(A)}_q + \sum_{q=1}^{D} \Phi^{(A):j,k}_{(B):q}[b]^{(B)}_q + \Phi^{(A):j,k}, \qquad (180)$$

with the constraints

$$\Phi^{(A):j,k}_{(QK,s):p,q} = \Phi^{(A):\pi_O(j),\pi_A(k)}_{(QK,\tau(s)):p,q},$$

$$\Phi^{(A):j,k}_{(VO,s):p,q} = \Phi^{(A):\pi_O(j),\pi_A(k)}_{(VO,\tau(s)):p,\pi_O(q)},$$

$$\Phi^{(A):j,k}_{(A):p,q} = \Phi^{(A):\pi_O(j),\pi_A(k)}_{(A):\pi_O(p),\pi_A(q)},$$

$$\Phi^{(A):j,k}_{(B):p,q} = \Phi^{(A):\pi_O(j),\pi_A(k)}_{(B):\pi_A(p),q},$$

$$\Phi^{(A):j,k}_{(A):q} = \Phi^{(A):\pi_O(j),\pi_A(k)}_{(A):\pi_A(q)},$$

$$\Phi^{(A):j,k}_{(B):q} = \Phi^{(A):\pi_O(j),\pi_A(k)}_{(B):q},$$

$$\Phi^{(A):j,k} = \Phi^{(A):\pi_O(j),\pi_A(k)}.$$

for all $j, j', k, k', s, s', p, p', q, q', \pi_O, \pi_A$ satisfies the condition $[E(gW)]^{(A)}_{j,k} = [gE(W)]^{(A)}_{j,k}$.

**Case 6: Solve the equation $[E(gW)]^{(B)} = [gE(W)]^{(B)}$.** By using the similar argument as Case 5, we also see that:

$$[E(W)]^{(B)}_{j,k} = \sum_{s=1}^{h}\sum_{p=1}^{D}\sum_{q=1}^{D} \Phi^{(B):j,k}_{(QK,s):p,q}[WW]^{(QK,s)}_{p,q}$$

$$+ \sum_{s=1}^{h}\sum_{p=1}^{D}\sum_{q=1}^{D} \Phi^{(B):j,k}_{(VO,s):p,q}[WW]^{(VO,s)}_{p,q}$$

$$+ \sum_{p=1}^{D}\sum_{q=1}^{D_A} \Phi^{(B):j,k}_{(A):p,q}[W]^{(A)}_{p,q} + \sum_{p=1}^{D_A}\sum_{q=1}^{D} \Phi^{(B):j,k}_{(B):p,q}[W]^{(B)}_{p,q}$$

$$+ \sum_{q=1}^{D_A} \Phi^{(B):j,k}_{(A):q}[b]^{(A)}_q + \sum_{q=1}^{D} \Phi^{(B):j,k}_{(B):q}[b]^{(B)}_q + \Phi^{(B):j,k}, \qquad (181)$$

with the constraints

$$\Phi^{(B):j,k}_{(QK,s):p,q} = \Phi^{(B):\pi_A(j),k}_{(QK,\tau(s)):p,q}$$

$$\Phi^{(B):j,k}_{(VO,s):p,q} = \Phi^{(B):\pi_A(j),k}_{(VO,\tau(s)):p,\pi_O(q)}$$

$$\Phi^{(B):j,k}_{(A):p,q} = \Phi^{(B):\pi_A(j),k}_{(A):\pi_O(p),\pi_A(q)}$$

$$\Phi^{(B):j,k}_{(B):p,q} = \Phi^{(B):\pi_A(j),k}_{(B):\pi_A(p),q}$$

$$\Phi^{(B):j,k}_{(A):q} = \Phi^{(B):\pi_A(j),k}_{(A):\pi_A(q)}$$

$$\Phi^{(B):j,k}_{(B):q} = \Phi^{(B):\pi_A(j),k}_{(B):q}$$

$$\Phi^{(B):j,k} = \Phi^{(B):\pi_A(j),k}.$$

for all $j, k, s, p, q, \tau, \pi_O, \pi_A$, satisfies the constraint $[E(gW)]^{(B)}_{j,k} = [gE(W)]^{(B)}_{j,k}$.

**Case 7. Solve the equation $[E(gb)]^{(A)} = [gE(b)]^{(A)}$.** By using the same argument as Case 5, we also see that $[E(b)]^{(A)}_{k}$ defined by

$$[E(b)]^{(A)}_{k} = \sum_{s=1}^{h}\sum_{p=1}^{D}\sum_{q=1}^{D} \Phi^{(A):k}_{(QK,s):p,q}[WW]^{(QK,s)}_{p,q}$$

$$+ \sum_{s=1}^{h}\sum_{p=1}^{D}\sum_{q=1}^{D} \Phi^{(A):k}_{(VO,s):p,q}[WW]^{(VO,s)}_{p,q}$$

$$+ \sum_{p=1}^{D}\sum_{q=1}^{D_A} \phi^{(A):k}_{(A):p,q}[W]^{(A)}_{p,q} + \sum_{p=1}^{D_A}\sum_{q=1}^{D} \Phi^{(A):k}_{(B):p,q}[W]^{(B)}_{p,q}$$

$$+ \sum_{q=1}^{D_A} \Phi^{(A):k}_{(A):q}[b]^{(A)}_q + \sum_{q=1}^{D} \Phi^{(A):k}_{(B):q}[b]^{(B)}_q + \Phi^{(A):k}, \qquad (182)$$

with the constraints

$$\Phi^{(A):k}_{(QK,s):p,q} = \Phi^{(A):\pi_A(k)}_{(QK,\tau(s)):p,q}$$

$$\Phi^{(A):k}_{(VO,s):p,q} = \Phi^{(A):\pi_A(k)}_{(VO,\tau(s)):p,\pi_O(q)}$$

$$\Phi^{(A):k}_{(A):p,q} = \phi^{(A):\pi_A(k)}_{(A):\pi_O(p),\pi_A(q)}$$

$$\Phi^{(A):k}_{(B):p,q} = \Phi^{(A):\pi_A(k)}_{(B):\pi_A(p),q}$$

$$\Phi^{(A):k}_{(A):q} = \Phi^{(A):\pi_A(k)}_{(A):\pi_A(q)}$$

$$\Phi^{(A):k}_{(B):q} = \Phi^{(A):\pi_A(k)}_{(B):q}$$

$$\Phi^{(A):k} = \Phi^{(A):\pi_A(k)}.$$

for all $j, k, s, p, q, \tau, \pi_O, \pi_A$, satisfies the constraint $[E(gW)]^{(A)}_{j,k} = [gE(W)]^{(A)}_{j,k}$.

**Case 8. Solve the equation** $[E(gb)]^{(B)} = [gE(b)]^{(B)}$**.** By using the same argument as in Case 5, we see that $[E(b)]^{(B)}_q$ defined by

$$[E(b)]^{(B)}_k = \sum_{s=1}^{h} \sum_{p=1}^{D} \sum_{q=1}^{D} \Phi^{(B):k}_{(QK,s):p,q}[WW]^{(QK,s)}_{p,q}$$

$$+ \sum_{s=1}^{h} \sum_{p=1}^{D} \sum_{q=1}^{D} \Phi^{(B):k}_{(VO,s):p,q}[WW]^{(VO,s)}_{p,q}$$

$$+ \sum_{p=1}^{D} \sum_{q=1}^{D_A} \Phi^{(B):k}_{(A):p,q}[W]^{(A)}_{p,q} + \sum_{p=1}^{D_A} \sum_{q=1}^{D} \Phi^{(B):k}_{(B):p,q}[W]^{(B)}_{p,q}$$

$$+ \sum_{q=1}^{D_A} \Phi^{(B):k}_{(A):q}[b]^{(A)}_q + \sum_{q=1}^{D} \Phi^{(B):k}_{(B):q}[b]^{(B)}_q + \Phi^{(B):k}, \qquad (183)$$

with the constraints

$$\Phi^{(B):k}_{(QK,s):p,q} = \Phi^{(B):k}_{(QK,\tau(s)):p,q}$$

$$\Phi^{(B):k}_{(VO,s):p,q} = \Phi^{(B):k}_{(VO,\tau(s)):p,\pi_O(q)}$$

$$\Phi^{(B):k}_{(A):p,q} = \Phi^{(B):k}_{(A):\pi_O(p),\pi_A(q)}$$

$$\Phi^{(B):k}_{(B):p,q} = \Phi^{(B):k}_{(B):\pi_A(p),q}$$

$$\Phi^{(B):k}_{(A):q} = \Phi^{(B):k}_{(A):\pi_A(q)}$$

$$\Phi^{(B):k}_{(B):q} = \Phi^{(B):k}_{(B):q}$$

$$\Phi^{(B):k} = \Phi^{(B):k},$$

for all $j, k, s, p, q, \tau, \pi_A, \pi_O$, satisfies the constraint $[E(gW)]^{(A)}_{j,k} = [gE(W)]^{(A)}_{j,k}$.

### D.4 FINAL FORMULA FOR THE EQUIVARIANT POLYNOMIAL LAYER

We gather the results of the computations of the equivariant layer in the previous section here for further implementation.

**Theorem D.4.** *The map* $E: \mathcal{U} \to \mathcal{U}$ *with*

$$E(U) = \left( \left( [E(W)]^{(Q,i)}, [E(W)]^{(K,i)}, [E(W)]^{(V,i)}, [E(W)]^{(O,i)} \right)_{i=1,\ldots,h}, \right.$$

$$\left. \left( [E(W)]^{(A)}, [E(W)]^{(B)} \right), \left( [E(b)]^{(A)}, [E(b)]^{(B)} \right) \right), \qquad (184)$$

*is $\mathcal{G}_{\mathcal{U}}$-equivariant.*

*Here, the components of $E(U)$ are given below.*

1. $[E(W)]^{(Q,i)}$ *is defined as*

$$[E(W)]_{j,k}^{(Q,i)} = \sum_{p=1}^{D} \Phi_{(Q,i):p,k}^{(Q,i):j,k}[W]_{p,k}^{(Q,i)},$$

*with the constraints*

$$\Phi_{(Q,i):p,k}^{(Q,i):j,k} = \Phi_{(Q,\tau(i)):p,k'}^{(Q,\tau(i)):j,k'},$$

2. $[E(W)]^{(K,i)}$ *is defined as*

$$[E(W)]_{j,k}^{(K,i)} = \sum_{p=1}^{D} \Phi_{(K,i):p,k}^{(K,i):j,k}[W]_{p,k}^{(K,i)},$$

*with the constraints*

$$\Phi_{(K,i):p,k}^{(K,i):j,k} = \Phi_{(K,\tau(i)):p,k'}^{(K,\tau(i)):j,k'},$$

3. $[E(W)]^{(V,i)}$ *is defined by*

$$[E(W)]_{j,k}^{(V,i)} = \sum_{p=1}^{D} \Phi_{(V,i):p,k}^{(V,i):j,k}[W]_{p,k}^{(V,i)},$$

*with the constraints*

$$\Phi_{(V,i):p,k}^{(V,i):j,k} = \Phi_{(V,\tau(i)):p,k'}^{(V,\tau(i)):j,k'},$$

4. $[E(W)]^{(O,i)}$ *is defined by*

$$[E(W)]_{j,k}^{(O,i)} = \sum_{q=1}^{D} \Phi_{(O,i):j,q}^{(O,i):j,k}[W]_{j,q}^{(O,i)},$$

*with the constraints*

$$\Phi_{(O,i):j,q}^{(O,i):j,k} = \Phi_{(O,\tau(i)):j',\pi_O(q)}^{(O,\tau(i)):j',\pi_O(k)}$$

5. $[E(W)]^{(A)}$ *is defined by*

$$[E(W)]_{j,k}^{(A)} = \sum_{s=1}^{h}\sum_{p=1}^{D}\sum_{q=1}^{D} \Phi_{(QK,s):p,q}^{(A):j,k}[WW]_{p,q}^{(QK,s)}$$

$$+ \sum_{s=1}^{h}\sum_{p=1}^{D}\sum_{q=1}^{D} \Phi_{(VO,s):p,q}^{(A):j,k}[WW]_{p,q}^{(VO,s)}$$

$$+ \sum_{p=1}^{D}\sum_{q=1}^{D_A} \Phi_{(A):p,q}^{(A):j,k}[W]_{p,q}^{(A)} + \sum_{p=1}^{D_A}\sum_{q=1}^{D} \Phi_{(B):p,q}^{(A):j,k}[W]_{p,q}^{(B)}$$

$$+ \sum_{q=1}^{D_A} \Phi_{(A):q}^{(A):j,k}[b]_q^{(A)} + \sum_{q=1}^{D} \Phi_{(B):q}^{(A):j,k}[b]_q^{(B)} + \Phi^{(A):j,k},$$

*with the constraints*

$$\Phi_{(QK,s):p,q}^{(A):j,k} = \Phi_{(QK,\tau(s)):p,q}^{(A):\pi_O(j),\pi_A(k)},$$

$$\Phi_{(VO,s):p,q}^{(A):j,k} = \Phi_{(VO,\tau(s)):p,\pi_O(q)}^{(A):\pi_O(j),\pi_A(k)},$$

$$\Phi_{(A):p,q}^{(A):j,k} = \Phi_{(A):\pi_O(p),\pi_A(q)}^{(A):\pi_O(j),\pi_A(k)},$$

$$\Phi_{(B):p,q}^{(A):j,k} = \Phi_{(B):\pi_A(p),q}^{(A):\pi_O(j),\pi_A(k)},$$

$$\Phi_{(A):q}^{(A):j,k} = \Phi_{(A):\pi_A(q)}^{(A):\pi_O(j),\pi_A(k)},$$

$$\Phi_{(B):q}^{(A):j,k} = \Phi_{(B):q}^{(A):\pi_O(j),\pi_A(k)},$$

$$\Phi^{(A):j,k} = \Phi^{(A):\pi_O(j),\pi_A(k)}.$$

6. $[E(W)]^{(B)}$ *is defined by*

$$[E(W)]_{j,k}^{(B)} = \sum_{s=1}^{h}\sum_{p=1}^{D}\sum_{q=1}^{D} \Phi_{(QK,s):p,q}^{(B):j,k}[WW]_{p,q}^{(QK,s)}$$

$$+ \sum_{s=1}^{h}\sum_{p=1}^{D}\sum_{q=1}^{D} \Phi_{(VO,s):p,q}^{(B):j,k}[WW]_{p,q}^{(VO,s)}$$

$$+ \sum_{p=1}^{D}\sum_{q=1}^{D_A} \Phi_{(A):p,q}^{(B):j,k}[W]_{p,q}^{(A)} + \sum_{p=1}^{D_A}\sum_{q=1}^{D} \Phi_{(B):p,q}^{(B):j,k}[W]_{p,q}^{(B)}$$

$$+ \sum_{q=1}^{D_A} \Phi_{(A):q}^{(B):j,k}[b]_{q}^{(A)} + \sum_{q=1}^{D} \Phi_{(B):q}^{(B):j,k}[b]_{q}^{(B)} + \Phi^{(B):j,k},$$

*with the constraints*

$$\Phi_{(QK,s):p,q}^{(B):j,k} = \Phi_{(QK,\tau(s)):p,q}^{(B):\pi_A(j),k}$$

$$\Phi_{(VO,s):p,q}^{(B):j,k} = \Phi_{(VO,\tau(s)):p,\pi_O(q)}^{(B):\pi_A(j),k}$$

$$\Phi_{(A):p,q}^{(B):j,k} = \Phi_{(A):\pi_O(p),\pi_A(q)}^{(B):\pi_A(j),k}$$

$$\Phi_{(B):p,q}^{(B):j,k} = \Phi_{(B):\pi_A(p),q}^{(B):\pi_A(j),k}$$

$$\Phi_{(A):q}^{(B):j,k} = \Phi_{(A):\pi_A(q)}^{(B):\pi_A(j),k}$$

$$\Phi_{(B):q}^{(B):j,k} = \Phi_{(B):q}^{(B):\pi_A(j),k}$$

$$\Phi^{(B):j,k} = \Phi^{(B):\pi_A(j),k}.$$

7. $[E(b)]^{(A)}$ *is defined by*

$$[E(b)]_{k}^{(A)} = \sum_{s=1}^{h}\sum_{p=1}^{D}\sum_{q=1}^{D} \Phi_{(QK,s):p,q}^{(A):k}[WW]_{p,q}^{(QK,s)}$$

$$+ \sum_{s=1}^{h}\sum_{p=1}^{D}\sum_{q=1}^{D} \Phi_{(VO,s):p,q}^{(A):k}[WW]_{p,q}^{(VO,s)}$$

$$+ \sum_{p=1}^{D}\sum_{q=1}^{D_A} \phi_{(A):p,q}^{(A):k}[W]_{p,q}^{(A)} + \sum_{p=1}^{D_A}\sum_{q=1}^{D} \Phi_{(B):p,q}^{(A):k}[W]_{p,q}^{(B)}$$

$$+ \sum_{q=1}^{D_A} \Phi_{(A):q}^{(A):k}[b]_{q}^{(A)} + \sum_{q=1}^{D} \Phi_{(B):q}^{(A):k}[b]_{q}^{(B)} + \Phi^{(A):k},$$

*with the constraints*

$$\Phi_{(QK,s):p,q}^{(A):k} = \Phi_{(QK,\tau(s)):p,q}^{(A):\pi_A(k)}$$

$$\Phi_{(VO,s):p,q}^{(A):k} = \Phi_{(VO,\tau(s)):p,\pi_O(q)}^{(A):\pi_A(k)}$$

$$\Phi_{(A):p,q}^{(A):k} = \phi_{(A):\pi_O(p),\pi_A(q)}^{(A):\pi_A(k)}$$

$$\Phi_{(B):p,q}^{(A):k} = \Phi_{(B):\pi_A(p),q}^{(A):\pi_A(k)}$$

$$\Phi_{(A):q}^{(A):k} = \Phi_{(A):\pi_A(q)}^{(A):\pi_A(k)}$$

$$\Phi_{(B):q}^{(A):k} = \Phi_{(B):q}^{(A):\pi_A(k)}$$

$$\Phi^{(A):k} = \Phi^{(A):\pi_A(k)}.$$

8. $[E(b)]^{(B)}$ *is defined by*

$$[E(b)]_{k}^{(B)} = \sum_{s=1}^{h}\sum_{p=1}^{D}\sum_{q=1}^{D} \Phi_{(QK,s):p,q}^{(B):k}[WW]_{p,q}^{(QK,s)}$$

$$+ \sum_{s=1}^{h} \sum_{p=1}^{D} \sum_{q=1}^{D} \Phi_{(VO,s):p,q}^{(B):k} [WW]_{p,q}^{(VO,s)}$$

$$+ \sum_{p=1}^{D} \sum_{q=1}^{D_A} \Phi_{(A):p,q}^{(B):k} [W]_{p,q}^{(A)} + \sum_{p=1}^{D_A} \sum_{q=1}^{D} \Phi_{(B):p,q}^{(B):k} [W]_{p,q}^{(B)}$$

$$+ \sum_{q=1}^{D_A} \Phi_{(A):q}^{(B):k} [b]_q^{(A)} + \sum_{q=1}^{D} \Phi_{(B):q}^{(B):k} [b]_q^{(B)} + \Phi^{(B):k},$$

*with the constraints*

$$\Phi_{(QK,s):p,q}^{(B):k} = \Phi_{(QK,\tau(s)):p,q}^{(B):k}$$

$$\Phi_{(VO,s):p,q}^{(B):k} = \Phi_{(VO,\tau(s)):p,\pi_O(q)}^{(B):k}$$

$$\Phi_{(A):p,q}^{(B):k} = \Phi_{(A):\pi_O(p),\pi_A(q)}^{(B):k}$$

$$\Phi_{(B):p,q}^{(B):k} = \Phi_{(B):\pi_A(p),q}^{(B):k}$$

$$\Phi_{(A):q}^{(B):k} = \Phi_{(A):\pi_A(q)}^{(B):k}$$

$$\Phi_{(B):q}^{(B):k} = \Phi_{(B):q}^{(B):k}$$

$$\Phi^{(B):k} = \Phi^{(B):k},$$

*for all $j, j', k, k', s, s', p, p', q, q'$ and $\tau, \pi_O, \pi_A$.*

# E   AN INVARIANT POLYNOMIAL LAYER

In this section, we will construct a map $I \colon \mathcal{U} \to \mathbb{R}^{D'}$ such that $I$ is equivariant with respect to $\mathcal{G}_{\mathcal{U}}$, i.e.

$$I(gU) = I(U), \tag{185}$$

for all $U \in \mathcal{U}$ and $g \in \mathcal{G}_{\mathcal{U}}$.

## E.1   A GENERAL FORM WITH UNKNOWN COEFFICIENTS

We select $I(U)$ to be a linear combination of entries of the matrices $[W]^{(Q,s)}$, $[W]^{(K,s)}$, $[W]^{(V,s)}$, $[W]^{(O,s)}$, $[W]^{(A)}$, $[W]^{(B)}$, $[b]^{(A)}$, $[b]^{(B)}$ as well as the matrices $[WW]^{(QK,s)}$ and $[WW]^{(VO,s)}$, i.e

$$I(U) = \sum_{s=1}^{h} \sum_{p=1}^{D} \sum_{q=1}^{D} \Phi_{(QK,s):p,q} \cdot [WW]_{p,q}^{(QK,s)}$$

$$+ \sum_{s=1}^{h} \sum_{p=1}^{D} \sum_{q=1}^{D} \Phi_{(VO,s):p,q} \cdot [WW]_{p,q}^{(VO,s)}$$

$$+ \sum_{s=1}^{h} \sum_{p=1}^{D} \sum_{q=1}^{D_k} \Phi_{(Q,s):p,q} \cdot [W]_{p,q}^{(Q,s)} + \sum_{s=1}^{h} \sum_{p=1}^{D} \sum_{q=1}^{D_k} \Phi_{(K,s):p,q} \cdot [W]_{p,q}^{(K,s)}$$

$$+ \sum_{s=1}^{h} \sum_{p=1}^{D} \sum_{q=1}^{D_v} \Phi_{(V,s):p,q} \cdot [W]_{p,q}^{(V,s)} + \sum_{s=1}^{h} \sum_{p=1}^{D_v} \sum_{q=1}^{D} \Phi_{(O,s):p,q} \cdot [W]_{p,q}^{(O,s)}$$

$$+ \sum_{p=1}^{D} \sum_{q=1}^{D_A} \Phi_{(A):p,q} \cdot [W]_{p,q}^{(A)} + \sum_{p=1}^{D_A} \sum_{q=1}^{D} \Phi_{(B):p,q} \cdot [W]_{p,q}^{(B)}$$

$$+ \sum_{q=1}^{D_A} \Phi_{(A):q} \cdot [b]_q^{(A)} + \sum_{q=1}^{D} \Phi_{(B):q} \cdot [b]_q^{(B)} + \Phi_1, \tag{186}$$

where the coefficients $\Phi_\_$s are matrices of size $D' \times 1$.

In the following, we will determine constraints on the coefficients such that $I$ will be $\mathcal{G}_{\mathcal{U}}$-invariant under these constraints.

### E.2 COMPUTE $I(gU)$

By using the same arguments in the computation of $E(gU)$, we have

$$
\begin{aligned}
I(gW) &= \sum_{s=1}^{h}\sum_{p=1}^{D}\sum_{q=1}^{D}\Phi_{(QK,\tau^{-1}(s)):p,q}[WW]_{p,q}^{(QK,s)} \\
&+ \sum_{s=1}^{h}\sum_{p=1}^{D}\sum_{q=1}^{D}\Phi_{(VO,\tau^{-1}(s)):p,\pi_O^{-1}(q)}[WW]_{p,q}^{(VO,s)} \\
&+ \sum_{s=1}^{h}\sum_{p=1}^{D}\sum_{q=1}^{D_k}\Phi_{(Q,\tau^{-1}(s)):p,q}\left[[W]^{(Q,s)}\cdot\left(M^{(s)}\right)^{\top}\right]_{p,q} \\
&+ \sum_{s=1}^{h}\sum_{p=1}^{D}\sum_{q=1}^{D_k}\Phi_{(K,\tau^{-1}(s)):p,q}\left[[W]^{(K,s)}\cdot\left(M^{(s)}\right)^{-1}\right]_{p,q} \\
&+ \sum_{s=1}^{h}\sum_{p=1}^{D}\sum_{q=1}^{D_v}\Phi_{(V,\tau^{-1}(s)):p,q}\left[[W]^{(V,s)}\cdot N^{(s)}\right]_{p,q} \\
&+ \sum_{s=1}^{h}\sum_{p=1}^{D_v}\sum_{q=1}^{D}\Phi_{(O,\tau^{-1}(s)):p,\pi_O^{-1}(q)}\left[\left(N^{(s)}\right)^{-1}\cdot[W]^{(O,s)}\right]_{p,q} \\
&+ \sum_{p=1}^{D}\sum_{q=1}^{D_A}\Phi_{(A):\pi_O^{-1}(p),\pi_A^{-1}(q)}[W]_{p,q}^{(A)} + \sum_{p=1}^{D_A}\sum_{q=1}^{D}\Phi_{(B):\pi_A^{-1}(p),q}[W]_{p,q}^{(B)} \\
&+ \sum_{q=1}^{D_A}\Phi_{(A):\pi_A^{-1}(q)}[b]_q^{(A)} + \sum_{q=1}^{D}\Phi_{(B):q}[b]_q^{(B)} + \Phi.
\end{aligned}
\tag{187}
$$

### E.3 COMPARE $I(gU)$ AND $I(U)$

According to Lemma D.1, in order to have $I(gU) = I(U)$, the corresponding coefficients of $I(gU)$ and $I(U)$ must be equal. As a consequence, we have

$$
\begin{aligned}
\Phi_{(QK,\tau^{-1}(s)):p,q} &= \Phi_{(QK,s):p,q} \\
\Phi_{(VO,\tau^{-1}(s)):p,\pi_O^{-1}(q)} &= \Phi_{(VO,s):p,q} \\
\Phi_{(Q,\tau^{-1}(s)):p,q} &= 0 \\
\Phi_{(K,\tau^{-1}(s)):p,q} &= 0 \\
\Phi_{(V,\tau^{-1}(s)):p,q} &= 0 \\
\Phi_{(O,\tau^{-1}(s)):p,\pi_O^{-1}(q)} &= 0 \\
\Phi_{(A):\pi_O^{-1}(p),\pi_A^{-1}(q)} &= \Phi_{(A):p,q} \\
\Phi_{(B):\pi_A^{-1}(p),q} &= \Phi_{(B):p,q} \\
\Phi_{(A):\pi_A^{-1}(q)} &= \Phi_{(A):q} \\
\Phi_{(B):q} &= \Phi_{(B):q} \\
\Phi_1 &= \Phi_1,
\end{aligned}
$$

or equivalently,

$$
\begin{aligned}
\Phi_{(QK,s):p,q} &= \Phi_{(QK,\tau(s)):p,q} \\
\Phi_{(VO,s):p,q} &= \Phi_{(VO,\tau(s)):p,\pi_O(q)} \\
\Phi_{(Q,s):p,q} &= 0 \\
\Phi_{(K,s):p,q} &= 0 \\
\Phi_{(V,s):p,q} &= 0
\end{aligned}
$$

$$\Phi_{(O,s):p,q} = 0$$
$$\Phi_{(A):p,q} = \Phi_{(A):\pi_O(p),\pi_A(q)}$$
$$\Phi_{(B):p,q} = \Phi_{(B):\pi_A(p),q}$$
$$\Phi_{(A):q} = \Phi_{(A):\pi_A(q)}$$
$$\Phi_{(B):q} = \Phi_{(B):q}$$
$$\Phi_1 = \Phi_1.$$

### E.4   FINAL FORMULA FOR THE INVARIANT POLYNOMIAL LAYER

We summarize the above computation here for implementation.

**Theorem E.1.** *The map $I : \mathcal{U} \to \mathbb{R}^{D'}$ defined by*

$$
\begin{aligned}
I(U) = &\sum_{s=1}^{h}\sum_{p=1}^{D}\sum_{q=1}^{D} \Phi_{(QK,s):p,q} \cdot [WW]_{p,q}^{(QK,s)} \\
&+ \sum_{s=1}^{h}\sum_{p=1}^{D}\sum_{q=1}^{D} \Phi_{(VO,s):p,q} \cdot [WW]_{p,q}^{(VO,s)} \\
&+ \sum_{p=1}^{D}\sum_{q=1}^{D_A} \Phi_{(A):p,q} \cdot [W]_{p,q}^{(A)} + \sum_{p=1}^{D_A}\sum_{q=1}^{D} \Phi_{(B):p,q} \cdot [W]_{p,q}^{(B)} \\
&+ \sum_{q=1}^{D_A} \Phi_{(A):q} \cdot [b]_q^{(A)} + \sum_{q=1}^{D} \Phi_{(B):q} \cdot [b]_q^{(B)} + \Phi_1,
\end{aligned}
\tag{188}
$$

*with the constraints*

$$\Phi_{(QK,s):p,q} = \Phi_{(QK,\tau(s)):p,q}$$
$$\Phi_{(VO,s):p,q} = \Phi_{(VO,\tau(s)):p,\pi_O(q)}$$
$$\Phi_{(A):p,q} = \Phi_{(A):\pi_O(p),\pi_A(q)}$$
$$\Phi_{(B):p,q} = \Phi_{(B):\pi_A(p),q}$$
$$\Phi_{(A):q} = \Phi_{(A):\pi_A(q)}$$
$$\Phi_{(B):q} = \Phi_{(B):q}$$
$$\Phi_1 = \Phi_1.$$

*is $\mathcal{G}_{\mathcal{U}}$-equivariant.*

## F   COMPUTATION COMPLEXITY OF EQUIVARIANT AND INVARIANT LAYERS

The computational complexity of the Transformer-NFN is derived from its invariant and equivariant layers as outlined in their pseudocode (Appendix I.2, I.3):

- **Equivariant Layer Complexity:** $\mathcal{O}(d \cdot e \cdot h \cdot D^2 \cdot \max(D_q, D_k, D_v))$
- **Invariant Layer Complexity:** $\mathcal{O}(d \cdot e \cdot D' \cdot D \cdot \max(D \cdot h, D_A))$

Where the parameters follow our notation in Table 8.

The Transformer-NFN implementation leverages optimized tensor contraction operations (e.g., `einsum`), enabling efficient and highly parallelizable computations on modern GPUs. This ensures computational practicality while delivering significant performance improvements.

## G   ADDITIONAL DATASET DETAILS

To create a wide range of transformer model, we opt to vary six hyperparameters in our experiments: train fraction, optimizer (SGD, SGDm, Adam, or RMSprop), learning rate, L2 regularization coefficient, weight initialization standard deviation, and dropout probability. The train fraction determines the proportion of the original training dataset used, while the optimizer dictates the algorithm for parameter updates. Learning rate, L2 regularization, and weight initialization standard deviation

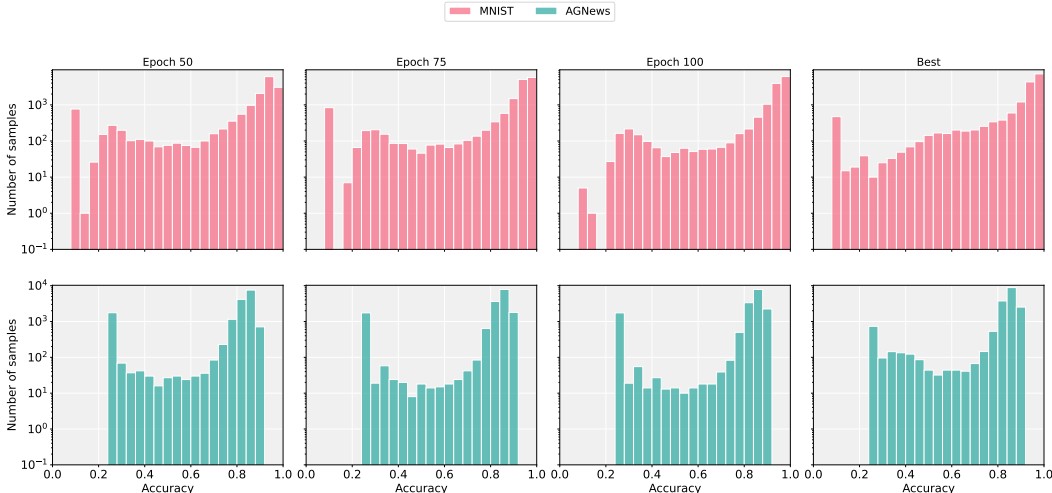

Figure 1: Accuracy histogram of MNIST task and AGNews task in the Small Transformer Zoo. The number of samples is showed in log scale for improved visibility.

control various aspects of the training process, and dropout probability helps in preventing overfitting.

We select a range of typical values for each hyperparameter independently, then generate all possible combinations to form our set of hyperparameter configurations. During our preliminary experiments, we observed that the optimal range of hyperparameters varies significantly between optimizer types. Consequently, we divided our settings into two categories: Adam-RMSprop and SGD-SGDm. Table 4 provides a detailed overview of our hyperparameter configurations. Overall, there are 8000 configurations for each category, resulting in 16000 configurations in total. These configurations are consistently applied across both tasks to ensure comparability. All models are trained for 100 epochs, with checkpoints and accuracy measurements recorded at epochs 50, 75, 100, and at the epoch with the best accuracy. During the process, we eliminate any runs that crash.

Table 4: Hyperparameter configurations of the Small Transformer Zoo Dataset

| Hyperparameter | SGD-SGDm | Adam-RMSprop |
|---|---|---|
| Train Fraction | [1.0, 0.9, 0.8, 0.7] | [1.0, 0.9, 0.8, 0.7] |
| Dropout | [0.2, 0.15, 0.1, 0.05, 0] | [0.2, 0.15, 0.1, 0.05, 0] |
| Learning Rate | [1e-3, 3e-3, 5e-3, 7e-3, 1e-2, 3e-2, 5e-2, 7e-2] | [3e-4, 5e-4, 1e-3, 3e-3, 5e-3, 1e-2, 3e-2, 5e-2] |
| Weight Init Standard Deviation | [0.1, 0.15, 0.2, 0.25, 0.3] | [0.1, 0.2, 0.3, 0.4, 0.5] |
| L2 Regularization | [1e-8, 1e-7, 1e-6, 1e-4, 1e-2] | [1e-8, 1e-7, 1e-6, 1e-4, 1e-2] |

**MNIST-Transformers.** The MNIST dataset (LeCun & Cortes, 2005) is a widely-used benchmark in computer vision, consisting of $28 \times 28$ pixel grayscale images of handwritten digits (0-9). For this vision task, the model objective is to perform digit classification. The embedding component first applies a 2D convolution to encode image patches and then adds a fixed positional encoding to provide spatial information. The encoder, comprising two stacked transformer blocks, processes these embeddings to capture complex relationships between different parts of the image. The classifier applies global average pooling to the encoder's output and then passes it through two fully connected layers with a ReLU activation in between, finally outputting probabilities for ten classes corresponding to the digits 0-9. Using our hyperparameter configurations, we generate a total of 62756 model samples for the MNIST task, including 15689 checkpoints at epochs 50, 75, 100, and the best-performing epoch, each trained with a distinct combination of hyperparameters. The accuracy distribution for MNIST in Figure 1 displays a strong concentration in the 80% to 100% range

Table 5: Ablation study on varying the hidden dimension and number of layers in Transformer-NFN, trained on the AGNews-Transformers dataset. Dimensions of all equivariant layers are indicated inside square brackets.

| Transformer-NFN dimensions | [3] | [5] | [10] | [15] | [3, 3] | [5, 5] | [10, 10] | [15, 15] |
|---|---|---|---|---|---|---|---|---|
| Kendall's $\tau$ | $0.907 \pm 0.002$ | $0.909 \pm 0.002$ | $0.909 \pm 0.001$ | $0.913 \pm 0.001$ | $0.905 \pm 0.003$ | $0.911 \pm 0.002$ | $0.913 \pm 0.001$ | $0.913 \pm 0.002$ |
| Num. params | 0.491M | 0.840M | 1.793M | 2.857M | 1.901M | 4.757M | 17.457M | 38.101M |

Table 6: Number of parameters for all models

| Model | MNIST | AGNews |
|---|---|---|
| Transformer-NFN | 1.812M | 1.804M |
| MLP | 0.933M | 0.896M |
| STATNN | 0.203M | 0.168M |

while remaining models distribute almost uniformly across lower accuracies, with a slight increase around $10\%$.

**AGNews-Transformers.** The AG's News Topic Classification Dataset (Zhang et al., 2015) is a collection of news articles from the AG's corpus of news articles on the web, categorized into four classes: World, Sports, Business, and Sci/Tech. We take the description of the articles input and train models to predict its corresponding topic. Our transformer-based model is adapted for this task as follows: The embedding component uses a pre-trained Word2Vec model to map each token to an embedding vector. These embeddings are combined with fixed positional encodings to retain sequential information. The encoder, consisting of two stacked transformer blocks, processes these embeddings to capture contextual relationships within the text. The classifier applies global average pooling to the encoder's output, then passes it through two fully connected layers with a ReLU activation in between, finally outputting probabilities for the four categories. Our experiments on this task yield a diverse set of 63796 model checkpoints, derived from 15949 unique model configurations. These checkpoints are collected at four key points during training: epochs 50, 75, 100, and at the epoch of peak performance. The accuracy distribution for AGNews in Figure 1 shows a notable concentration in the $50\%$ to $90\%$ range, with a peak around $80\%$ while also exhibiting a smaller cluster of models around $25\%$.

## H    ADDITIONAL EXPERIMENTAL DETAILS

### H.1    GENERAL DETAILS

**Training details** The models were trained for a total of 50 epochs, using a batch size of 16. We employed the Adam optimizer with a maximum learning rate of $10^{-3}$. A linear warmup strategy was applied to the learning rate, spanning the initial 10 epochs for gradual warmup. We utilize Binary Cross Entropy for the loss function.

**Number of parameters** Table 6 summarizes the total parameter count for all models. The architectural details and hyperparameter configurations for each model are provided in H.2 and H.3. Importantly, we optimized the baseline models to their best configurations, and further increasing the number of parameters would likely result in overfitting.

### H.2    ARCHIECTURE AND HYPERPARAMETERS OF TRANSFORMER-NFN

The Transformer-NFN model is composed of three main components designed to handle the input weights of a transformer network. For the embedding and classifier components, both of which are MLP-based, we employ a standard MLP with ReLU activation to process individual component. The transformer block itself is handled by an invariant architecture, incorporating several equivariant polynomial layers of Transformer-NFN. These layers utilize ReLU activation, which specifically operates on the two MLP components within the transformer block. Following this, the output is passed through an invariant polynomial layer of Transformer-NFN. The outputs of each of these components are represented as vectors, which are concatenated and passed through a final MLP head with Sigmoid activation for prediction.

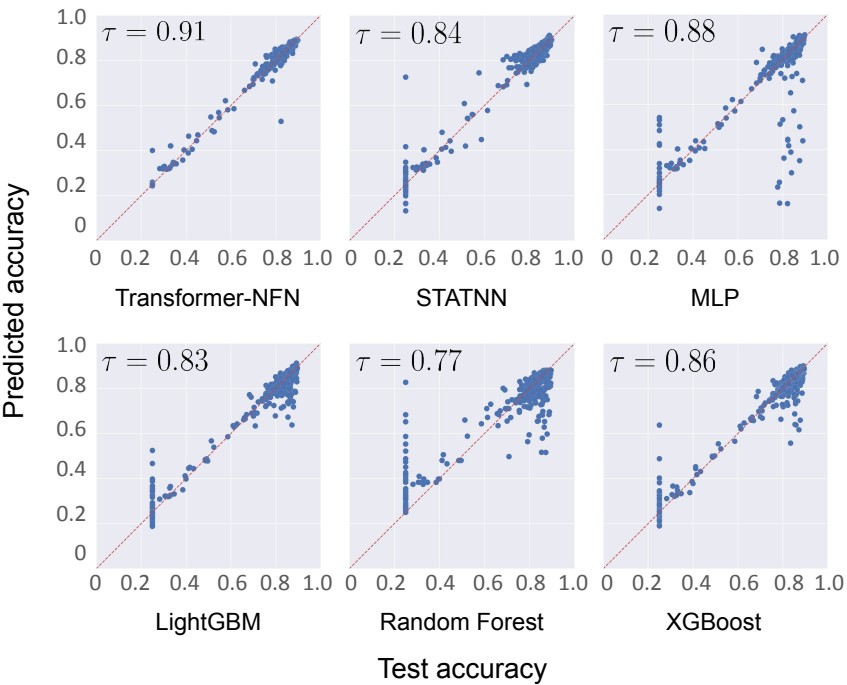

Figure 2: Visualization of all models on test set of AGNews-Transformers dataset.

In our experimental setup, the embedding component is modeled using a single-layer MLP with 10 hidden neurons, while the classifier component is a two-layer MLP, each layer containing 10 hidden neurons. For the invariant architecture of Transformer-NFN, we apply an equivariant polynomial layer of Transformer-NFN with 10 hidden channels to process the weights, followed by an invariant polynomial layer of Transformer-NFN and an MLP that outputs a 10-dimensional vector. The resulting vectors from these components are concatenated and passed through a single classification head to generate the final prediction.

### H.3 ARCHITECTURE AND HYPERPARAMETERS FOR OTHER BASELINES

Here we describe the architecture of all baselines:

- **MLP** In the MLP model, the weights of all components are first flattened and processed through a separate MLP for each component. Specifically, the MLP handling the transformer block and embedding components consists of a single layer with 50 hidden neurons, while the classifier component is modeled using a two-layer MLP, each layer containing 50 neurons. The outputs from all components are concatenated and passed through a final MLP to produce the prediction.

- **STATNN** (Unterthiner et al., 2020) For the STATNN model, we adapt the original approach to work with the transformer block. In this case, we compute statistical features from the weights of the query, key, value, output, as well as the weights and biases of the two linear layers. These features are concatenated and passed through a single-layer MLP with 256 hidden neurons. The classifier component uses the original STATNN model to extract features, which are then processed through an MLP with 256 hidden neurons. For the embedding component, a single-layer MLP with 64 hidden neurons is employed. The outputs from all components are concatenated and passed through a single-layer MLP for the final prediction.

- **XGBoost** (Chen & Guestrin, 2016), **LightGBM** (Ke et al., 2017), **Random Forest** (Breiman, 2001): We flattened the weights of all components and directly employed popular gradient boosting libraries for regression. The hyperparameters for all three tree-based models were set to a maximum depth of 10, a minimum child weight of 50, and a maximum of 256 leaves.

### H.4 EXPERIMENT ON AUGMENTED AGNEWS-TRANSFORMERS DATASET

**Experiment Setup**    In this experiment, we evaluate the performance of Transformer-NFN on the AGNews-Transformers dataset augmented using the group action $\mathcal{G}_{\mathcal{U}}$. We perform a 2-fold augmentation for both the train and test sets. The original weights are retained, and additional weights are constructed by applying permutations and scaling transformations to transformer modules. The elements in $M$ and $N$ (see Section 4.3) are uniformly sampled across $[-1, 1]$, $[-10, 10]$, and $[-100, 100]$.

**Results**    The results for all models are presented in Table 7. The findings demonstrate that Transformer-NFN maintains stable Kendall's $\tau$ across different ranges of scale operators. Notably, as the weights are augmented, the performance of Transformer-NFN improves, whereas the performance of other baseline methods declines significantly. This performance disparity results in a widening gap between Transformer-NFN and the second-best model, increasing from 0.031 to 0.082.

Table 7: Performance measured by Kendall's $\tau$ of all models on augmented AGNews-Transformers dataset using the group action $\mathcal{G}_{\mathcal{U}}$. Uncertainties indicate standard error over 5 runs.

| | Original | $[-1, 1]$ | $[-10, 10]$ | $[-100, 100]$ |
|---|---|---|---|---|
| XGBoost | $0.859 \pm 0.001$ | $0.799 \pm 0.003$ | $0.800 \pm 0.001$ | $0.802 \pm 0.003$ |
| LightGBM | $0.835 \pm 0.001$ | $0.785 \pm 0.003$ | $0.784 \pm 0.003$ | $0.786 \pm 0.004$ |
| Random Forest | $0.774 \pm 0.003$ | $0.714 \pm 0.001$ | $0.715 \pm 0.002$ | $0.716 \pm 0.002$ |
| MLP | $\underline{0.879 \pm 0.006}$ | $\underline{0.830 \pm 0.002}$ | $\underline{0.833 \pm 0.002}$ | $\underline{0.833 \pm 0.005}$ |
| STATNN | $0.841 \pm 0.002$ | $0.793 \pm 0.003$ | $0.791 \pm 0.003$ | $0.771 \pm 0.013$ |
| Transformer-NFN | $\mathbf{0.910 \pm 0.001}$ | $\mathbf{0.912 \pm 0.001}$ | $\mathbf{0.912 \pm 0.002}$ | $\mathbf{0.913 \pm 0.001}$ |
| Gap | 0.031 | 0.082 | 0.079 | 0.080 |

The general decline observed in baseline models highlights their lack of symmetry. In contrast, Transformer-NFN's equivariance to $\mathcal{G}_{\mathcal{U}}$ ensures stability and even slight performance improvements under augmentation. This underscores the importance of defining the maximal symmetric group, to which Transformer-NFN is equivariant, in overcoming the limitations of baseline methods.

## I    IMPLEMENTATION OF EQUIVARIANT AND INVARIANT LAYERS

We present the multi-channel implementations of the $\mathcal{G}_{\mathcal{U}}$-equivariant map $E : \mathcal{U}^d \to \mathcal{U}^e$ and the $\mathcal{G}_{\mathcal{U}}$-invariant map $I : \mathcal{U}^d \to \mathbb{R}^{e \times D'}$. We summarize the Equivariant and Invariant Layers with the bullet notation $\bullet$ and adopt einops-like pseudocode to maintain consistency and standardization in transformer weight space manipulations.

To facilitate understanding of the implementation, we summarize the key dimensions involved in Table 8 and define the shapes of the input terms in Table 9.

### I.1    SUMMARY OF EQUIVARIANT AND INVARIANT LAYERS

We summarize our derived Equivariant and Invariant Layers from Appendix D and Appendix E. We use the bullet notation $\bullet$ to simlify the notation, making it easier to implementation. Roughly speaking, the bullet stands for "the parameter is unchanged when the bullet varies across all possible index values".

#### I.1.1    EQUIVARIANT LAYERS WITH BULLET NOTATION

$$[E(W)]_{j,k}^{(Q:i)} = \Phi_{(Q:\bullet):p,\bullet}^{(Q:\bullet):j,\bullet} \cdot [W]_{p,q}^{(Q:i)},$$

$$[E(W)]_{j,k}^{(K:i)} = \Phi_{(K:\bullet):p,\bullet}^{(K:\bullet):j,\bullet} \cdot [W]_{p,q}^{(K:i)},$$

$$[E(W)]_{j,k}^{(V:i)} = \Phi_{(V:\bullet):p,\bullet}^{(V:\bullet):j,\bullet} \cdot [W]_{p,q}^{(V:i)},$$

$$[E(W)]_{j,k}^{(O:i)} = \left(\Phi_{(O:\bullet):\bullet,\bullet}^{(O:\bullet):\bullet,\bullet}\right)_1 \cdot \sum_{k=1}^{D} [W]_{j,k}^{(O:i)} + \left(\Phi_{(O:\bullet):\bullet,\bullet}^{(O:\bullet):\bullet,\bullet}\right)_2 \cdot [W]_{j,k}^{(O:i)},$$

Table 8: Summary of key dimensions involved in the implementation

| Symbol | Description |
|---|---|
| $d$ | Number of input channels for the equivariant and invariant layer |
| $e$ | Number of output channels for the equivariant and invariant layer |
| $D$ | Embedding dimension of the input and output sequences of the transformer block |
| $D_k = D_q$ | Embedding dimension for key and query vectors in the transformer block |
| $D_v$ | Embedding dimension for value vectors in the transformer block |
| $D_A$ | Embedding dimension for the linear projection step in the transformer block |
| $h$ | Number of attention heads in the transformer block |
| $b$ | Batch size |
| $D'$ | Embedding dimension of the invariant layer's output |

Table 9: Shapes of input terms used in the implementation

| Term | Shape |
|---|---|
| $[W]_{p,q}^{(Q,i)}$ | $[b, d, h, D, D_q]$ |
| $[W]_{p,q}^{(K,i)}$ | $[b, d, h, D, D_k]$ |
| $[W]_{p,q}^{(V,i)}$ | $[b, d, h, D, D_v]$ |
| $[W]_{p,q}^{(O,i)}$ | $[b, d, h, D_v, D]$ |
| $[WW]_{p,q}^{(QK,i)}$ | $[b, d, h, D, D]$ |
| $[WW]_{p,q}^{(VO,i)}$ | $[b, d, h, D, D]$ |
| $[W]_{p,q}^{A}$ | $[b, d, D, D_A]$ |
| $[b]_{q}^{A}$ | $[b, d, D_A]$ |
| $[W]_{p,q}^{B}$ | $[b, d, D_A, D]$ |
| $[b]_{q}^{B}$ | $[b, d, D]$ |

$$
\begin{aligned}
[E(W)]_{j,k}^{(A)} &= \sum_{p=1}^{D}\sum_{q=1}^{D} \Phi_{(QK,\bullet):p,q}^{(A):\bullet,\bullet}\left(\sum_{s=1}^{h}[WW]_{p,q}^{(QK,s)}\right) \\
&+ \sum_{p=1}^{D}\sum_{q=1}^{D}\left(\Phi_{(VO,\bullet):p,\bullet}^{(A):\bullet,\bullet}\right)_1\left(\sum_{s=1}^{h}[WW]_{p,q}^{(VO,s)}\right) \\
&+ \sum_{p=1}^{D}\left(\Phi_{(VO,\bullet):p,\bullet}^{(A):\bullet,\bullet}\right)_2\left(\sum_{s=1}^{h}[WW]_{p,j}^{(VO,s)}\right) \\
&+ \sum_{p=1}^{D}\sum_{q=1}^{D_A}\left(\Phi_{(A):\bullet,\bullet}^{(A):\bullet,\bullet}\right)_1[W]_{p,q}^{(A)} + \sum_{q=1}^{D_A}\left(\Phi_{(A):\bullet,\bullet}^{(A):\bullet,\bullet}\right)_2[W]_{j,q}^{(A)} \\
&+ \sum_{p=1}^{D}\left(\Phi_{(A):\bullet,\bullet}^{(A):\bullet,\bullet}\right)_3[W]_{p,k}^{(A)} + \left(\Phi_{(A):\bullet,\bullet}^{(A):\bullet,\bullet}\right)_4[W]_{j,k}^{(A)} \\
&+ \sum_{p=1}^{D_A}\sum_{q=1}^{D}\left(\Phi_{(B):\bullet,q}^{(A):\bullet,\bullet}\right)_1[W]_{p,q}^{(B)} + \sum_{q=1}^{D}\left(\Phi_{(B):\bullet,q}^{(A):\bullet,\bullet}\right)_2[W]_{k,q}^{(B)} \\
&+ \sum_{q=1}^{D_A}\left(\Phi_{(A):\bullet}^{(A):\bullet,\bullet}\right)_1[b]_q^{(A)} + \left(\Phi_{(A):\bullet}^{(A):\bullet,\bullet}\right)_2[b]_k^{(A)} \\
&+ \sum_{q=1}^{D}\Phi_{(B):q}^{(A):\bullet,\bullet}[b]_q^{(B)} + \Phi^{(A):\bullet,\bullet},
\end{aligned}
$$

$$[E(W)]_{j,k}^{(B)} = \sum_{p=1}^{D}\sum_{q=1}^{D}\sum_{s=1}^{h} \Phi_{(QK,\bullet):p,q}^{(B):\bullet,k}\left([WW]_{p,q}^{(QK,s)}\right)$$

$$+ \sum_{p=1}^{D}\sum_{q=1}^{D}\sum_{s=1}^{h} \Phi_{(VO,\bullet):p,\bullet}^{(B):\bullet,k}\left([WW]_{p,q}^{(VO,s)}\right)$$

$$+ \sum_{p=1}^{D}\sum_{q=1}^{D_A} \left(\Phi_{(A):\bullet,\bullet}^{(B):\bullet,k}\right)_1 [W]_{p,q}^{(A)} + \sum_{p=1}^{D}\left(\Phi_{(A):\bullet,\bullet}^{(B):\bullet,k}\right)_2 [W]_{p,j}^{(A)}$$

$$+ \sum_{p=1}^{D_A}\sum_{q=1}^{D} \left(\Phi_{(B):\bullet,q}^{(B):\bullet,k}\right)_1 [W]_{p,q}^{(B)} + \sum_{q=1}^{D}\left(\Phi_{(B):\bullet,q}^{(B):\bullet,k}\right)_2 [W]_{j,q}^{(B)}$$

$$+ \sum_{q=1}^{D_A} \left(\Phi_{(A):\bullet}^{(B):\bullet,k}\right)_1 [b]_q^{(A)} + \left(\Phi_{(A):\bullet}^{(B):\bullet,k}\right)_2 [b]_j^{(A)}$$

$$+ \sum_{q=1}^{D} \Phi_{(B):q}^{(B):\bullet,k}[b]_q^{(B)} + \Phi^{(B):\bullet,k},$$

$$[E(b)]_k^{(A)} = \sum_{p=1}^{D}\sum_{q=1}^{D} \Phi_{(QK,\bullet):p,q}^{(A):\bullet}\left(\sum_{s=1}^{h}[WW]_{p,q}^{(QK,s)}\right)$$

$$+ \sum_{p=1}^{D}\sum_{q=1}^{D} \Phi_{(VO,\bullet):p,\bullet}^{(A):\bullet}\left(\sum_{s=1}^{h}[WW]_{p,q}^{(VO,s)}\right)$$

$$+ \sum_{p=1}^{D}\sum_{q=1}^{D_A} \left(\phi_{(A):\bullet,\bullet}^{(A):\bullet}\right)_1 [W]_{p,q}^{(A)} + \sum_{p=1}^{D}\left(\phi_{(A):\bullet,\bullet}^{(A):\bullet}\right)_2 [W]_{p,k}^{(A)}$$

$$+ \sum_{p=1}^{D_A}\sum_{q=1}^{D} \left(\Phi_{(B):\bullet,q}^{(A):\bullet}\right)_1 [W]_{p,q}^{(B)} + \sum_{q=1}^{D}\left(\Phi_{(B):\bullet,q}^{(A):\bullet}\right)_2 [W]_{k,q}^{(B)}$$

$$+ \sum_{q=1}^{D_A} \left(\Phi_{(A):\bullet}^{(A):\bullet}\right)_1 [b]_q^{(A)} + \left(\Phi_{(A):\bullet}^{(A):\bullet}\right)_2 [b]_k^{(A)}$$

$$+ \sum_{q=1}^{D} \Phi_{(B):q}^{(A):\bullet}[b]_q^{(B)} + \Phi^{(A):\bullet},$$

$$[E(b)]_k^{(B)} = \sum_{p=1}^{D}\sum_{q=1}^{D}\sum_{s=1}^{h} \Phi_{(QK,\bullet):p,q}^{(B):k}\left([WW]_{p,q}^{(QK,s)}\right)$$

$$+ \sum_{p=1}^{D}\sum_{q=1}^{D}\sum_{s=1}^{h} \Phi_{(VO,\bullet):p,\bullet}^{(B):k}\left([WW]_{p,q}^{(VO,s)}\right)$$

$$+ \sum_{p=1}^{D}\sum_{q=1}^{D_A} \Phi_{(A):\bullet,\bullet}^{(B):k}[W]_{p,q}^{(A)} + \sum_{p=1}^{D_A}\sum_{q=1}^{D} \Phi_{(B):\bullet,q}^{(B):k}[W]_{p,q}^{(B)}$$

$$+ \sum_{q=1}^{D_A} \Phi_{(A):\bullet}^{(B):k}[b]_q^{(A)} + \sum_{q=1}^{D} \Phi_{(B):q}^{(B):k}[b]_q^{(B)} + \Phi^{(B):k}.$$

## I.1.2 INVARIANT LAYERS WITH BULLET NOTATION

$$I(U) = \sum_{p=1}^{D}\sum_{q=1}^{D} \Phi_{(QK,s):p,q} \cdot \left(\sum_{s=1}^{h}[WW]_{p,q}^{(QK,s)}\right)$$

$$+ \sum_{s=1}^{h} \sum_{p=1}^{D} \sum_{q=1}^{D} \Phi_{(VO,s):p,\bullet} \cdot \left( \sum_{s=1}^{h} [WW]_{p,q}^{(VO,s)} \right)$$

$$+ \sum_{p=1}^{D} \sum_{q=1}^{D_A} \Phi_{(A):\bullet,\bullet} \cdot [W]_{p,q}^{(A)} + \sum_{p=1}^{D_A} \sum_{q=1}^{D} \Phi_{(B):\bullet,q} \cdot [W]_{p,q}^{(B)}$$

$$+ \sum_{q=1}^{D_A} \Phi_{(A):\bullet} \cdot [b]_q^{(A)} + \sum_{q=1}^{D} \Phi_{(B):q} \cdot [b]_q^{(B)} + \Phi_1$$

## I.2 EQUIVARIANT LAYERS PSEUDOCODE

### I.2.1 $[E(W)]_{j,k}^{(Q:i)}$ PSEUDOCODE

From the formula:

$$[E(W)]_{j,k}^{(Q:i)} = \Phi_{(Q:\bullet):p,\bullet}^{(Q:\bullet):j,\bullet} \cdot [W]_{p,q}^{(Q:i)}$$

We define the pseudocode for each term:

For $\Phi_{(Q:\bullet):p,\bullet}^{(Q:\bullet):j,\bullet} \cdot [W]_{p,q}^{(Q:i)}$, with $[W]_{p,q}^{(Q:i)}$ of shape $[b, d, h, D, D_q]$ and $\Phi_{(Q:\bullet):p,\bullet}^{(Q:\bullet):j,\bullet}$ of shape $[e, d, D, D]$

    Corresponding pseudocode: `einsum`$(bdhpk, edjp \rightarrow behjk)$

### I.2.2 $[E(W)]_{j,k}^{(K:i)}$ PSEUDOCODE

From the formula:

$$[E(W)]_{j,k}^{(K:i)} = \Phi_{(K:\bullet):p,\bullet}^{(K:\bullet):j,\bullet} \cdot [W]_{p,q}^{(K:i)}$$

We define the pseudocode for each term:

For $\Phi_{(K:\bullet):p,\bullet}^{(K:\bullet):j,\bullet} \cdot [W]_{p,q}^{(K:i)}$, with $[W]_{p,q}^{(K:i)}$ of shape $[b, d, h, D, D_k]$ and $\Phi_{(K:\bullet):p,\bullet}^{(K:\bullet):j,\bullet}$ of shape $[e, d, D, D]$

    Corresponding pseudocode: `einsum`$(bdhpk, edjp \rightarrow behjk)$

### I.2.3 $[E(W)]_{j,k}^{(V:i)}$ PSEUDOCODE

From the formula:

$$[E(W)]_{j,k}^{(V:i)} = \Phi_{(V:\bullet):p,\bullet}^{(V:\bullet):j,\bullet} \cdot [W]_{p,q}^{(V:i)}$$

We define the pseudocode for each term:

For $\Phi_{(V:\bullet):p,\bullet}^{(V:\bullet):j,\bullet} \cdot [W]_{p,q}^{(V:i)}$, with $[W]_{p,q}^{(V:i)}$ of shape $[b, d, h, D, D_v]$ and $\Phi_{(V:\bullet):p,\bullet}^{(V:\bullet):j,\bullet}$ of shape $[e, d, D, D]$

    Corresponding pseudocode: `einsum`$(bdhpk, edjp \rightarrow behjk)$

### I.2.4 $[E(W)]_{j,k}^{(O:i)}$ PSEUDOCODE

From the formula:

$$[E(W)]_{j,k}^{(O:i)} = \left( \Phi_{(O:\bullet):\bullet,\bullet}^{(O:\bullet):\bullet,\bullet} \right)_1 \cdot \sum_{k=1}^{D} [W]_{j,k}^{(O:i)} + \left( \Phi_{(O:\bullet):\bullet,\bullet}^{(O:\bullet):\bullet,\bullet} \right)_2 \cdot [W]_{j,k}^{(O:i)}$$

We define the pseudocode for each term:

For $\left( \Phi_{(O:\bullet):\bullet,\bullet}^{(O:\bullet):\bullet,\bullet} \right)_1 \cdot \sum_{k=1}^{D} [W]_{j,k}^{(O:i)}$, with $[W]_{j,k}^{(O:i)}$ of shape $[b, d, h, D_v, D]$ and $\left( \Phi_{(O:\bullet):\bullet,\bullet}^{(O:\bullet):\bullet,\bullet} \right)_1$ of shape $[e, d]$

    Corresponding pseudocode: `einsum`$(bdhjk, ed \rightarrow behj)$`.unsqueeze`$(-1)$

For $\left( \Phi_{(O:\bullet):\bullet,\bullet}^{(O:\bullet):\bullet,\bullet} \right)_2 \cdot [W]_{j,k}^{(O:i)}$, with $[W]_{j,k}^{(O:i)}$ of shape $[b, d, h, D_v, D]$ and $\left( \Phi_{(O:\bullet):\bullet,\bullet}^{(O:\bullet):\bullet,\bullet} \right)_2$ of shape $[e, d]$

    Corresponding pseudocode: `einsum`$(bdhjk, ed \rightarrow behjk)$

### I.2.5 $[E(W)]_{j,k}^{(A)}$ PSEUDOCODE

From the formula:

$$
\begin{aligned}
[E(W)]_{j,k}^{(A)} ={}& \sum_{p=1}^{D}\sum_{q=1}^{D}\Phi_{(QK,\bullet):p,q}^{(A):\bullet,\bullet}\left(\sum_{s=1}^{h}[WW]_{p,q}^{(QK,s)}\right) \\
&+ \sum_{p=1}^{D}\sum_{q=1}^{D}\left(\Phi_{(VO,\bullet):p,\bullet}^{(A):\bullet,\bullet}\right)_{1}\left(\sum_{s=1}^{h}[WW]_{p,q}^{(VO,s)}\right) \\
&+ \sum_{p=1}^{D}\left(\Phi_{(VO,\bullet):p,\bullet}^{(A):\bullet,\bullet}\right)_{2}\left(\sum_{s=1}^{h}[WW]_{p,j}^{(VO,s)}\right) \\
&+ \sum_{p=1}^{D}\sum_{q=1}^{D_A}\left(\Phi_{(A):\bullet,\bullet}^{(A):\bullet,\bullet}\right)_{1}[W]_{p,q}^{(A)} + \sum_{q=1}^{D_A}\left(\Phi_{(A):\bullet,\bullet}^{(A):\bullet,\bullet}\right)_{2}[W]_{j,q}^{(A)} \\
&+ \sum_{p=1}^{D}\left(\Phi_{(A):\bullet,\bullet}^{(A):\bullet,\bullet}\right)_{3}[W]_{p,k}^{(A)} + \left(\Phi_{(A):\bullet,\bullet}^{(A):\bullet,\bullet}\right)_{4}[W]_{j,k}^{(A)} \\
&+ \sum_{p=1}^{D_A}\sum_{q=1}^{D}\left(\Phi_{(B):\bullet,q}^{(A):\bullet,\bullet}\right)_{1}[W]_{p,q}^{(B)} + \sum_{q=1}^{D}\left(\Phi_{(B):\bullet,q}^{(A):\bullet,\bullet}\right)_{2}[W]_{k,q}^{(B)} \\
&+ \sum_{q=1}^{D_A}\left(\Phi_{(A):\bullet}^{(A):\bullet,\bullet}\right)_{1}[b]_{q}^{(A)} + \left(\Phi_{(A):\bullet}^{(A):\bullet,\bullet}\right)_{2}[b]_{k}^{(A)} \\
&+ \sum_{q=1}^{D}\Phi_{(B):q}^{(A):\bullet,\bullet}[b]_{q}^{(B)} + \Phi^{(A):\bullet,\bullet}
\end{aligned}
$$

We define the pseudocode for each term:

For $\Phi_{(QK,\bullet):p,q}^{(A):\bullet,\bullet}\cdot\sum_{s=1}^{h}[WW]_{p,q}^{(QK,s)}$, with $[WW]_{p,q}^{(QK,s)}$ of shape $[b,d,h,D,D]$ and $\Phi_{(QK,\bullet):p,q}^{(A):\bullet,\bullet}$

of shape $[e,d,D,D]$

Corresponding pseudocode: $\texttt{einsum}(bdhpq, edpq \to be).unsqueeze(-1).unsqueeze(-1)$

For $\left(\Phi_{(VO,\bullet):p,\bullet}^{(A):\bullet,\bullet}\right)_{1}\cdot\sum_{s=1}^{h}[WW]_{p,q}^{(VO,s)}$, with $[WW]_{p,q}^{(VO,s)}$ of shape $[b,d,h,D,D]$ and

$\left(\Phi_{(VO,\bullet):p,\bullet}^{(A):\bullet,\bullet}\right)_{1}$ of shape $[e,d,D]$

Corresponding pseudocode: $\texttt{einsum}(bdhpq, edp \to be).unsqueeze(-1).unsqueeze(-1)$

For $\left(\Phi_{(VO,\bullet):p,\bullet}^{(A):\bullet,\bullet}\right)_{2}\cdot\sum_{s=1}^{h}[WW]_{p,j}^{(VO,s)}$, with $[WW]_{p,j}^{(VO,s)}$ of shape $[b,d,h,D,D]$ and

$\left(\Phi_{(VO,\bullet):p,\bullet}^{(A):\bullet,\bullet}\right)_{2}$ of shape $[e,d,D]$

Corresponding pseudocode: $\texttt{einsum}(bdhpj, edp \to bej).unsqueeze(-1)$

For $\left(\Phi_{(A):\bullet,\bullet}^{(A):\bullet,\bullet}\right)_{1}\cdot[W]_{p,q}^{(A)}$, with $[W]_{p,q}^{(A)}$ of shape $[b,d,D,D_A]$ and $\left(\Phi_{(A):\bullet,\bullet}^{(A):\bullet,\bullet}\right)_{1}$ of shape $[e,d]$

Corresponding pseudocode: $\texttt{einsum}(bdpq, ed \to be).unsqueeze(-1).unsqueeze(-1)$

For $\left(\Phi_{(A):\bullet,\bullet}^{(A):\bullet,\bullet}\right)_{2}\cdot[W]_{j,q}^{(A)}$, with $[W]_{j,q}^{(A)}$ of shape $[b,d,D,D_A]$ and $\left(\Phi_{(A):\bullet,\bullet}^{(A):\bullet,\bullet}\right)_{2}$ of shape $[e,d]$

Corresponding pseudocode: $\texttt{einsum}(bdjq, ed \to bej).unsqueeze(-1)$

For $\left(\Phi_{(A):\bullet,\bullet}^{(A):\bullet,\bullet}\right)_{3}\cdot[W]_{p,k}^{(A)}$, with $[W]_{p,k}^{(A)}$ of shape $[b,d,D,D_A]$ and $\left(\Phi_{(A):\bullet,\bullet}^{(A):\bullet,\bullet}\right)_{3}$ of shape $[e,d]$

Corresponding pseudocode: $\texttt{einsum}(bdpk, ed \to bek).unsqueeze(-2)$

For $\left(\Phi_{(A):\bullet,\bullet}^{(A):\bullet,\bullet}\right)_4 \cdot [W]_{j,k}^{(A)}$, with $[W]_{j,k}^{(A)}$ of shape $[b,d,D,D_A]$ and $\left(\Phi_{(A):\bullet,\bullet}^{(A):\bullet,\bullet}\right)_4$ of shape $[e,d]$

    Corresponding pseudocode: $\texttt{einsum}(bdjk, ed \rightarrow bejk)$

For $\left(\Phi_{(B):\bullet,q}^{(A):\bullet,\bullet}\right)_1 \cdot [W]_{p,q}^{(B)}$, with $[W]_{p,q}^{(B)}$ of shape $[b,d,D_A,D]$ and $\left(\Phi_{(B):\bullet,q}^{(A):\bullet,\bullet}\right)_1$ of shape $[e,d,D]$

    Corresponding pseudocode: $\texttt{einsum}(bdpq, edq \rightarrow be).unsqueeze(-1).unsqueeze(-1)$

For $\left(\Phi_{(B):\bullet,q}^{(A):\bullet,\bullet}\right)_2 \cdot [W]_{k,q}^{(B)}$, with $[W]_{k,q}^{(B)}$ of shape $[b,d,D_A,D]$ and $\left(\Phi_{(B):\bullet,q}^{(A):\bullet,\bullet}\right)_2$ of shape $[e,d,D]$

    Corresponding pseudocode: $\texttt{einsum}(bdkq, edq \rightarrow bek).unsqueeze(-2)$

For $\left(\Phi_{(A):\bullet}^{(A):\bullet,\bullet}\right)_1 \cdot [b]_q^{(A)}$, with $[b]_q^{(A)}$ of shape $[b,d,D_A]$ and $\left(\Phi_{(A):\bullet}^{(A):\bullet,\bullet}\right)_1$ of shape $[e,d]$

    Corresponding pseudocode: $\texttt{einsum}(bdq, ed \rightarrow be).unsqueeze(-1).unsqueeze(-1)$

For $\left(\Phi_{(A):\bullet}^{(A):\bullet,\bullet}\right)_2 \cdot [b]_k^{(A)}$, with $[b]_k^{(A)}$ of shape $[b,d,D_A]$ and $\left(\Phi_{(A):\bullet}^{(A):\bullet,\bullet}\right)_2$ of shape $[e,d]$

    Corresponding pseudocode: $\texttt{einsum}(bdk, ed \rightarrow bek).unsqueeze(-2)$

For $\Phi_{(B):q}^{(A):\bullet,\bullet} \cdot [b]_q^{(B)}$, with $[b]_q^{(B)}$ of shape $[b,d,D]$ and $\Phi_{(B):q}^{(A):\bullet,\bullet}$ of shape $[e,d,D]$

    Corresponding pseudocode: $\texttt{einsum}(bdq, edq \rightarrow be).unsqueeze(-1).unsqueeze(-1)$

For $\Phi^{(A):\bullet,\bullet}$ with shape $[e]$

    Corresponding pseudocode: $\texttt{einsum}(e \rightarrow e).unsqueeze(0).unsqueeze(-1).unsqueeze(-1)$

### I.2.6   $[E(b)]_k^{(A)}$ PSEUDOCODE

From the formula:

$$
\begin{aligned}
[E(b)]_k^{(A)} =& \sum_{p=1}^{D}\sum_{q=1}^{D} \Phi_{(QK,\bullet):p,q}^{(A):\bullet} \left(\sum_{s=1}^{h}[WW]_{p,q}^{(QK,s)}\right) \\
&+ \sum_{p=1}^{D}\sum_{q=1}^{D} \Phi_{(VO,\bullet):p,\bullet}^{(A):\bullet} \left(\sum_{s=1}^{h}[WW]_{p,q}^{(VO,s)}\right) \\
&+ \sum_{p=1}^{D}\sum_{q=1}^{D_A} \left(\phi_{(A):\bullet,\bullet}^{(A):\bullet}\right)_1 [W]_{p,q}^{(A)} + \sum_{p=1}^{D} \left(\phi_{(A):\bullet,\bullet}^{(A):\bullet}\right)_2 [W]_{p,k}^{(A)} \\
&+ \sum_{p=1}^{D_A}\sum_{q=1}^{D} \left(\Phi_{(B):\bullet,q}^{(A):\bullet}\right)_1 [W]_{p,q}^{(B)} + \sum_{q=1}^{D} \left(\Phi_{(B):\bullet,q}^{(A):\bullet}\right)_2 [W]_{k,q}^{(B)} \\
&+ \sum_{q=1}^{D_A} \left(\Phi_{(A):\bullet}^{(A):\bullet}\right)_1 [b]_q^{(A)} + \left(\Phi_{(A):\bullet}^{(A):\bullet}\right)_2 [b]_k^{(A)} \\
&+ \sum_{q=1}^{D} \Phi_{(B):q}^{(A):\bullet}[b]_q^{(B)} + \Phi^{(A):\bullet}.
\end{aligned}
$$

We define the pseudocode for each term:

For $\Phi_{(QK,\bullet):p,q}^{(A):\bullet} \cdot \left(\sum_{s=1}^{h}[WW]_{p,q}^{(QK,s)}\right)$, with $[WW]_{p,q}^{(QK,s)}$ of shape $[b,d,h,D,D]$ and $\Phi_{(QK,\bullet):p,q}^{(A):\bullet}$ of shape $[e,d,D,D]$

    Corresponding pseudocode: $\texttt{einsum}(bdhpq, edpq \rightarrow be) \, \texttt{.unsqueeze}(-1)$

For $\Phi_{(VO,\bullet):p,\bullet}^{(A):\bullet} \cdot \left(\sum_{s=1}^{h}[WW]_{p,q}^{(VO,s)}\right)$, with $[WW]_{p,q}^{(VO,s)}$ of shape $[b,d,h,D,D]$ and $\Phi_{(VO,\bullet):p,\bullet}^{(A):\bullet}$ of shape $[e,d,D]$

    Corresponding pseudocode: $\texttt{einsum}(bdhpq, edp \rightarrow be) \, \texttt{.unsqueeze}(-1)$

For $\left(\phi_{(A):\bullet,\bullet}^{(A):\bullet}\right)_1 \cdot [W]_{p,q}^{(A)}$, with $[W]_{p,q}^{(A)}$ of shape $[b,d,D,D_A]$ and $\left(\phi_{(A):\bullet,\bullet}^{(A):\bullet}\right)_1$ of shape $[e,d]$

Corresponding pseudocode: $\texttt{einsum}(bdpq, ed \rightarrow be)\texttt{.unsqueeze}(-1)$

For $\left(\phi_{(A):\bullet,\bullet}^{(A):\bullet}\right)_2 \cdot [W]_{p,k}^{(A)}$, with $[W]_{p,k}^{(A)}$ of shape $[b,d,D,D_A]$ and $\left(\phi_{(A):\bullet,\bullet}^{(A):\bullet}\right)_2$ of shape $[e,d]$

Corresponding pseudocode: $\texttt{einsum}(bdpk, ed \rightarrow bek)$

For $\left(\Phi_{(B):\bullet,q}^{(A):\bullet}\right)_1 \cdot [W]_{p,q}^{(B)}$, with $[W]_{p,q}^{(B)}$ of shape $[b,d,D_A,D]$ and $\left(\Phi_{(B):\bullet,q}^{(A):\bullet}\right)_1$ of shape $[e,d,D]$

Corresponding pseudocode: $\texttt{einsum}(bdpq, edq \rightarrow be)\texttt{.unsqueeze}(-1)$

For $\left(\Phi_{(B):\bullet,q}^{(A):\bullet}\right)_2 \cdot [W]_{k,q}^{(B)}$, with $[W]_{k,q}^{(B)}$ of shape $[b,d,D_A,D]$ and $\left(\Phi_{(B):\bullet,q}^{(A):\bullet}\right)_2$ of shape $[e,d,D]$

Corresponding pseudocode: $\texttt{einsum}(bdkq, edq \rightarrow bek)$

For $\left(\Phi_{(A):\bullet}^{(A):\bullet}\right)_1 \cdot [b]_q^{(A)}$, with $[b]_q^{(A)}$ of shape $[b,d,D_A]$ and $\left(\Phi_{(A):\bullet}^{(A):\bullet}\right)_1$ of shape $[e,d]$

Corresponding pseudocode: $\texttt{einsum}(bdq, ed \rightarrow be)\texttt{.unsqueeze}(-1)$

For $\left(\Phi_{(A):\bullet}^{(A):\bullet}\right)_2 \cdot [b]_k^{(A)}$, with $[b]_k^{(A)}$ of shape $[b,d,D_A]$ and $\left(\Phi_{(A):\bullet}^{(A):k}\right)_2$ of shape $[e,d]$

Corresponding pseudocode: $\texttt{einsum}(bdk, ed \rightarrow bek)$

For $\Phi_{(B):q}^{(A):\bullet} \cdot [b]_q^{(B)}$, with $[b]_q^{(B)}$ of shape $[b,d,D]$ and $\Phi_{(B):q}^{(A):\bullet}$ of shape $[e,d,D]$

Corresponding pseudocode: $\texttt{einsum}(bdq, edq \rightarrow be)\texttt{.unsqueeze}(-1)$

For $\Phi^{(A):\bullet}$ of shape $[e]$,

Corresponding pseudocode: $\texttt{einsum}(e \rightarrow e)\texttt{.unsqueeze}(0)\texttt{.unsqueeze}(-1)$

### I.2.7 $[E(W)]_{j,k}^{(B)}$ PSEUDOCODE

From the formula:

$$[E(W)]_{j,k}^{(B)} = \sum_{p=1}^{D}\sum_{q=1}^{D}\sum_{s=1}^{h} \Phi_{(QK,\bullet):p,q}^{(B):\bullet,k}\left([WW]_{p,q}^{(QK,s)}\right)$$

$$+ \sum_{p=1}^{D}\sum_{q=1}^{D}\sum_{s=1}^{h} \Phi_{(VO,\bullet):p,\bullet}^{(B):\bullet,k}\left([WW]_{p,q}^{(VO,s)}\right)$$

$$+ \sum_{p=1}^{D}\sum_{q=1}^{D_A} \left(\Phi_{(A):\bullet,\bullet}^{(B):\bullet,k}\right)_1 [W]_{p,q}^{(A)} + \sum_{p=1}^{D}\left(\Phi_{(A):\bullet,\bullet}^{(B):\bullet,k}\right)_2 [W]_{p,j}^{(A)}$$

$$+ \sum_{p=1}^{D_A}\sum_{q=1}^{D} \left(\Phi_{(B):\bullet,q}^{(B):\bullet,k}\right)_1 [W]_{p,q}^{(B)} + \sum_{q=1}^{D}\left(\Phi_{(B):\bullet,q}^{(B):\bullet,k}\right)_2 [W]_{j,q}^{(B)}$$

$$+ \sum_{q=1}^{D_A} \left(\Phi_{(A):\bullet}^{(B):\bullet,k}\right)_1 [b]_q^{(A)} + \left(\Phi_{(A):\bullet}^{(B):\bullet,k}\right)_2 [b]_j^{(A)}$$

$$+ \sum_{q=1}^{D} \Phi_{(B):q}^{(B):\bullet,k}[b]_q^{(B)} + \Phi^{(B):\bullet,k}$$

We define the pseudocode for each term:

For $\Phi_{(QK,\bullet):p,q}^{(B):\bullet,k} \cdot [WW]_{p,q}^{(QK,s)}$, with $[WW]_{p,q}^{(QK,s)}$ of shape $[b,d,h,D,D]$ and $\Phi_{(QK,\bullet):p,q}^{(B):\bullet,k}$

of shape $[e,d,D,D,D]$

Corresponding pseudocode: $\texttt{einsum}(bdhpq, edkpq \rightarrow bek)\texttt{.unsqueeze}(-2)$

For $\Phi_{(VO,\bullet):p,\bullet}^{(B):\bullet,k} \cdot [WW]_{p,q}^{(VO,s)}$, with $[WW]_{p,q}^{(VO,s)}$ of shape $[b,d,h,D,D]$ and $\Phi_{(VO,\bullet):p,\bullet}^{(B):\bullet,k}$

of shape $[e,d,D,D]$

Corresponding pseudocode: $\texttt{einsum}(bdhpq, edkp \rightarrow bek)\texttt{.unsqueeze}(-2)$

For $\left(\Phi_{(A):\bullet,\bullet}^{(B):\bullet,k}\right)_1 \cdot [W]_{p,q}^{(A)}$, with $[W]_{p,q}^{(A)}$ of shape $[b,d,D,D_A]$ and $\left(\Phi_{(A):\bullet,\bullet}^{(B):\bullet,k}\right)_1$ of shape $[e,d,D]$

    Corresponding pseudocode: $\mathtt{einsum}(bdpq, edk \rightarrow bek)\mathtt{.unsqueeze}(-2)$

For $\left(\Phi_{(A):\bullet,\bullet}^{(B):\bullet,k}\right)_2 \cdot [W]_{p,j}^{(A)}$, with $[W]_{p,j}^{(A)}$ of shape $[b,d,D,D_A]$ and $\left(\Phi_{(A):\bullet,\bullet}^{(B):\bullet,k}\right)_2$ of shape $[e,d,D]$

    Corresponding pseudocode: $\mathtt{einsum}(bdpj, edk \rightarrow bejk)$

For $\left(\Phi_{(B):\bullet,q}^{(B):\bullet,k}\right)_1 \cdot [W]_{p,q}^{(B)}$, with $[W]_{p,q}^{(B)}$ of shape $[b,d,D_A,D]$ and $\left(\Phi_{(B):\bullet,q}^{(B):\bullet,k}\right)_1$ of shape $[e,d,D,D]$

    Corresponding pseudocode: $\mathtt{einsum}(bdpq, edkq \rightarrow bek)\mathtt{.unsqueeze}(-2)$

For $\left(\Phi_{(B):\bullet,q}^{(B):\bullet,k}\right)_2 \cdot [W]_{j,q}^{(B)}$, with $[W]_{j,q}^{(B)}$ of shape $[b,d,D_A,D]$ and $\left(\Phi_{(B):\bullet,q}^{(B):\bullet,k}\right)_2$ of shape $[e,d,D,D]$

    Corresponding pseudocode: $\mathtt{einsum}(bdjq, edkq \rightarrow bejk)$

For $\left(\Phi_{(A):\bullet}^{(B):\bullet,k}\right)_1 \cdot [b]_q^{(A)}$, with $[b]_q^{(A)}$ of shape $[b,d,D_A]$ and $\left(\Phi_{(A):\bullet}^{(B):\bullet,k}\right)_1$ of shape $[e,d,D]$

    Corresponding pseudocode: $\mathtt{einsum}(bdq, edk \rightarrow bek)\mathtt{.unsqueeze}(-2)$

For $\left(\Phi_{(A):\bullet}^{(B):\bullet,k}\right)_2 \cdot [b]_j^{(A)}$, with $[b]_j^{(A)}$ of shape $[b,d,D_A]$ and $\left(\Phi_{(A):\bullet}^{(B):\bullet,k}\right)_2$ of shape $[e,d,D]$

    Corresponding pseudocode: $\mathtt{einsum}(bdj, edk \rightarrow bejk)$

For $\Phi_{(B):q}^{(B):\bullet,k} \cdot [b]_q^{(B)}$, with $[b]_q^{(B)}$ of shape $[b,d,D]$ and $\Phi_{(B):q}^{(B):\bullet,k}$ of shape $[e,d,D,D]$

    Corresponding pseudocode: $\mathtt{einsum}(bdq, edkq \rightarrow bek)\mathtt{.unsqueeze}(-2)$

For $\Phi^{(B):\bullet,k}$ of shape $[e,D]$,

    Corresponding pseudocode: $\mathtt{einsum}(ek \rightarrow ek)\mathtt{.unsqueeze}(0)$

### I.2.8   $[E(b)]_k^{(B)}$ PSEUDOCODE

From the formula:

$$[E(b)]_k^{(B)} = \sum_{p=1}^{D}\sum_{q=1}^{D}\sum_{s=1}^{h} \Phi_{(QK,\bullet):p,q}^{(B):k}\left([WW]_{p,q}^{(QK,s)}\right)$$

$$+ \sum_{p=1}^{D}\sum_{q=1}^{D}\sum_{s=1}^{h} \Phi_{(VO,\bullet):p,\bullet}^{(B):k}\left([WW]_{p,q}^{(VO,s)}\right)$$

$$+ \sum_{p=1}^{D}\sum_{q=1}^{D_A} \Phi_{(A):\bullet,\bullet}^{(B):k}[W]_{p,q}^{(A)} + \sum_{p=1}^{D_A}\sum_{q=1}^{D} \Phi_{(B):\bullet,q}^{(B):k}[W]_{p,q}^{(B)}$$

$$+ \sum_{q=1}^{D_A} \Phi_{(A):\bullet}^{(B):k}[b]_q^{(A)} + \sum_{q=1}^{D} \Phi_{(B):q}^{(B):k}[b]_q^{(B)} + \Phi^{(B):k}$$

We define the pseudocode for each term:

    For $\Phi_{(QK,\bullet):p,q}^{(B):k} \cdot [WW]_{p,q}^{(QK,s)}$, with $[WW]_{p,q}^{(QK,s)}$ of shape $[b,d,h,D,D]$ and $\Phi_{(QK,\bullet):p,q}^{(B):k}$
       of shape $[e,d,D,D,D]$

      Corresponding pseudocode: $\mathtt{einsum}(bdhpq, edkpq \rightarrow bek)$

    For $\Phi_{(VO,\bullet):p,\bullet}^{(B):k} \cdot [WW]_{p,q}^{(VO,s)}$, with $[WW]_{p,q}^{(VO,s)}$ of shape $[b,d,h,D,D]$ and $\Phi_{(VO,\bullet):p,\bullet}^{(B):k}$
       of shape $[e,d,D,D]$

      Corresponding pseudocode: $\mathtt{einsum}(bdhpq, edkp \rightarrow bek)$

    For $\Phi_{(A):\bullet,\bullet}^{(B):k} \cdot [W]_{p,q}^{(A)}$, with $[W]_{p,q}^{(A)}$ of shape $[b,d,D,D_A]$ and $\Phi_{(A):\bullet,\bullet}^{(B):k}$ of shape $[e,d,D]$

      Corresponding pseudocode: $\mathtt{einsum}(bdpq, edk \rightarrow bek)$

    For $\Phi_{(B):\bullet,q}^{(B):k} \cdot [W]_{p,q}^{(B)}$, with $[W]_{p,q}^{(B)}$ of shape $[b,d,D_A,D]$ and $\Phi_{(B):\bullet,q}^{(B):k}$ of shape $[e,d,D,D]$

      Corresponding pseudocode: $\mathtt{einsum}(bdpq, edkq \rightarrow bek)$

For $\Phi_{(A):\bullet}^{(B):k} \cdot [b]_q^{(A)}$, with $[b]_q^{(A)}$ of shape $[b, d, D_A]$ and $\Phi_{(A):\bullet}^{(B):k}$ of shape $[e, d, D]$

    Corresponding pseudocode: $\texttt{einsum}(bdq, edk \rightarrow bek)$

For $\Phi_{(B):q}^{(B):k} \cdot [b]_q^{(B)}$, with $[b]_q^{(B)}$ of shape $[b, d, D]$ and $\Phi_{(B):q}^{(B):k}$ of shape $[e, d, D, D]$

    Corresponding pseudocode: $\texttt{einsum}(bdq, edkq \rightarrow bek)$

For $\Phi^{(B):k}$ of shape $[e, D]$,

    Corresponding pseudocode: $\texttt{einsum}(ek \rightarrow ek)\texttt{.unsqueeze}(0)$

## I.3 INVARIANT LAYERS PSEUDOCODE

From the formula:

$$
\begin{aligned}
I(U) = &\sum_{p=1}^{D} \sum_{q=1}^{D} \Phi_{(QK,\bullet):p,q} \cdot \left( \sum_{s=1}^{h} [WW]_{p,q}^{(QK,s)} \right) \\
&+ \sum_{p=1}^{D} \sum_{q=1}^{D} \Phi_{(VO,\bullet):p,\bullet} \cdot \left( \sum_{s=1}^{h} [WW]_{p,q}^{(VO,s)} \right) \\
&+ \sum_{p=1}^{D} \sum_{q=1}^{D_A} \Phi_{(A):\bullet,\bullet} \cdot [W]_{p,q}^{(A)} + \sum_{p=1}^{D_A} \sum_{q=1}^{D} \Phi_{(B):\bullet,q} \cdot [W]_{p,q}^{(B)} \\
&+ \sum_{q=1}^{D_A} \Phi_{(A):\bullet} \cdot [b]_q^{(A)} + \sum_{q=1}^{D} \Phi_{(B):q} \cdot [b]_q^{(B)} + \Phi_1
\end{aligned}
$$

We define the pseudocode for each term:

For $\Phi_{(QK,\bullet):p,q} \cdot [WW]_{p,q}^{(QK,s)}$, with $[WW]_{p,q}^{(QK,s)}$ of shape $[b, d, h, D, D]$ and $\Phi_{(QK,\bullet):p,q}$ of shape $[e, d, D, D, D']$

    Corresponding pseudocode: $\texttt{einsum}(bdhpq, edpqk \rightarrow bek)$

For $\Phi_{(VO,\bullet):p,\bullet} \cdot [WW]_{p,q}^{(VO,s)}$, with $[WW]_{p,q}^{(VO,s)}$ of shape $[b, d, h, D, D]$ and $\Phi_{(VO,\bullet):p,\bullet}$ of shape $[e, d, D, D']$

    Corresponding pseudocode: $\texttt{einsum}(bdhpq, edpk \rightarrow bek)$

For $\Phi_{(A):\bullet,\bullet} \cdot [W]_{p,q}^{(A)}$, with $[W]_{p,q}^{(A)}$ of shape $[b, d, D, D_A]$ and $\Phi_{(A):\bullet,\bullet}$ of shape $[e, d, D']$

    Corresponding pseudocode: $\texttt{einsum}(bdpq, edk \rightarrow bek)$

For $\Phi_{(B):\bullet,q} \cdot [W]_{p,q}^{(B)}$, with $[W]_{p,q}^{(B)}$ of shape $[b, d, D_A, D]$ and $\Phi_{(B):\bullet,q}$ of shape $[e, d, D, D']$

    Corresponding pseudocode: $\texttt{einsum}(bdpq, edqk \rightarrow bek)$

For $\Phi_{(A):\bullet} \cdot [b]_q^{(A)}$, with $[b]_q^{(A)}$ of shape $[b, d, D_A]$ and $\Phi_{(A):\bullet}$ of shape $[e, d, D']$

    Corresponding pseudocode: $\texttt{einsum}(bdq, edk \rightarrow bek)$

For $\Phi_{(B):q} \cdot [b]_q^{(B)}$, with $[b]_q^{(B)}$ of shape $[b, d, D]$ and $\Phi_{(B):q}$ of shape $[e, d, D, D']$

    Corresponding pseudocode: $\texttt{einsum}(bdq, edqk \rightarrow bek)$

For $\Phi_1$ of shape $[e, D']$,

    Corresponding pseudocode: $\texttt{einsum}(ek \rightarrow ek)\texttt{.unsqueeze}(0)$

