# OpenReview forum: "Equivariant Neural Functional Networks for Transformers"
_ICLR.cc/2025/Conference — ICLR 2025 Poster_

### Official Review · Reviewer_ynVN · 2024-11-02

**Soundness:** 3
**Presentation:** 3
**Contribution:** 3
**Rating:** 8
**Confidence:** 2

**Summary:**

This submission provides a theoretical study of neural functional networks (NFNs) for transformers. By formally defining the weight space of the standard transformer block and the corresponding group action of such a space, the authors propose several design principles for NFNs and introduce Transformer-NFN to predict the test performance with a transformer checkpoint and the corresponding training configurations. Additionally, a dataset with 125k transformer checkpoints trained on MINST and AGNews will be released for reproduction and further analysis.

**Strengths:**

- This topic is quite interesting and do have some foreground. In the foreseeable future, a plenty of pre-trained transformer checkpoints would be released and there is no golden rule to find out which checkpoint is appropriate for a specific task. And the NFN may be useful and draw some inspiration for future research.
- For all I know, this may be one of the first works to develop NFN for transformers.
- The theoretical analysis is detailed enough.

**Weaknesses:**

- According to the main presentation of this submission, no obvious weaknesses are found.

**Questions:**

- How about the performance on language modeling? Is Transformer-NFN able to predict?
- How about the performance of the larger transformer
- From my perspective, the performance of a checkpoint is related to the training data, the architecture of the network, the training algorithm, hyperparameter configuration, etc. So why does NFN only take the weights and the configuration into account? Is it a sufficient condition for performance prediction?
- The evaluated dataset is far away from the present scale of transformer. The weight of modern transformers is huge (bert-base 110M parameters), is Transformer-NFN able to predict the performance of it?

---

> ### Author Response · Authors · 2024-11-18
>
> We appreciate the reviewer’s feedback and have provided the following responses to address the concerns raised about our paper. Below, we summarize the weaknesses and questions highlighted by the reviewer and provide our answers accordingly.
>
> ---
>
> **Q1+Q2. How about the performance on language modeling? Is Transformer-NFN able to predict?**
>
> **How about the performance of the larger transformer?**
>
> **Answer.** We greatly appreciate your observation regarding the importance of evaluating Transformer-NFN on tasks such as language modeling. While this would enhance understanding of NFN's applicability, generating such a dataset within the rebuttal period is challenging due to computational demands. For context, both our MNIST-Transformers and AGNews-Transformer datasets each consist of approximately 63k trained weights each. Extending this effort to larger-scale models, such as training thousands of 110M-parameter transformers, exceeds the computational resources of our academic lab. Nevertheless, we recognize the importance of this direction and plan to address it in future work.
>
> Currently, the closest evaluation we conducted to language modeling is text classification on the AGNews dataset. In this setting, Transformer-NFN demonstrated significant performance improvements over other baseline methods, as shown in **Q2** of the General Response.
>
> We recall the details of the experiment from Q2 in General Response as follows.
>
> ***Experiment Setup.*** In this experiment, we evaluate the performance of Transformer-NFN on the AGNews-Transformers dataset augmented using the group action $\mathcal{G}_\mathcal{U}$. We perform a 2-fold augmentation for both the train and test sets. The original weights are retained, and additional weights are constructed by applying permutations and scaling transformations to transformer modules. The elements in $M$ and $N$ (as defined in Section 4.3) are uniformly sampled across $[-1, 1]$, $[-10, 10]$, and $[-100, 100]$.
>
> ***Results.*** The results for all models are presented in Table 1. The findings demonstrate that Transformer-NFN maintains stable Kendall’s $\tau$ across different ranges of scale operators. Notably, as the weights are augmented, the performance of Transformer-NFN improves, whereas the performance of other baseline methods declines significantly. This performance disparity results in a widening gap between Transformer-NFN and the second-best model, increasing from 0.031 to 0.082.
>
> The general decline observed in baseline models highlights their lack of symmetry. In contrast, Transformer-NFN's equivariance to $\mathcal{G}_\mathcal{U}$ ensures stability and even slight performance improvements under augmentation. This underscores the importance of defining the maximal symmetric group, to which Transformer-NFN is equivariant, in overcoming the limitations of baseline methods.
>
> *Table 1: Performance measured by Kendall's $\tau$ of all models on **Augmented AGNews-Transformers** dataset. Uncertainties indicate standard error over 5 runs.*
> |                 |           Original            |                $[-1, 1]$                |               $[-10, 10]$                |               $[-100,100]$               |
> |:--------------- |:-----------------------------:|:-------------------------------:|:-------------------------------:|:-------------------------------:|
> | XGBoost         |       $0.859 \pm 0.001$       |       $0.799 \pm 0.003$       |       $0.800 \pm 0.001$       |       $0.802 \pm 0.003$       |
> | LightGBM        |       $0.835 \pm 0.001$       |       $0.785 \pm 0.003$       |       $0.784 \pm 0.003$       |       $0.786 \pm 0.004$       |
> | Random Forest   |       $0.774 \pm 0.003$       |       $0.714 \pm 0.001$       |       $0.715 \pm 0.002$       |       $0.716 \pm 0.002$       |
> | MLP             | $\underline{0.879 \pm 0.006}$ | $\underline{0.830 \pm 0.002}$ | $\underline{0.833 \pm 0.002}$ | $\underline{0.833 \pm 0.005}$ |
> | STATNN          |       $0.841 \pm 0.002$       |       $0.793 \pm 0.003$       |       $0.791 \pm 0.003$       |        $0.771 \pm 0.013$        |
> | **Transformer-NFN** |  $\mathbf{0.910 \pm 0.001}$   |  $\mathbf{0.912 \pm 0.001}$   |  $\mathbf{0.912 \pm 0.002}$   |  $\mathbf{0.913 \pm 0.001}$   |
> | Gap             |              $0.031$              |               $0.082$               |              $0.079$               |               $0.080$               |

---

> ### Author Response · Authors · 2024-11-18
>
> **Q3. From my perspective, the performance of a checkpoint is related to the training data, the architecture of the network, the training algorithm, hyperparameter configuration, etc. So why does NFN only take the weights and the configuration into account? Is it a sufficient condition for performance prediction?**
>
> **Answer.** In our study, we evaluate Transformer-NFN on a test set comprising transformer weights trained on the same task as those in the training set. Our findings demonstrate that Transformer-NFN, by learning from a diverse set of weights trained under varying hyperparameter configurations and training algorithms, can effectively predict the performance of unseen transformer weights trained in different settings.
>
> This highlights that transformer weights themselves contain substantial information, making them sufficient for predicting a model's generalization capability without relying on additional features. As shown in Table 3, the encoder weights hold the most relevant information, enabling accurate predictions of the generalization of pretrained weights. This aligns with intuition, as the transformer's encoder performs the majority of the computational work and is the most critical component driving the performance of the trained model.
>
> **Q4. The evaluated dataset is far away from the present scale of transformer. The weight of modern transformers is huge (bert-base 110M parameters), is Transformer-NFN able to predict the performance of it?**
>
> **Answer.** We agree that evaluating Transformer-NFN on larger models and datasets would provide a more comprehensive demonstration of its strengths. However, generating such datasets requires training thousands of large-scale transformers, which exceeds the computational resources available in our academic lab. For context, both our MNIST-Transformers and AGNews-Transformer datasets each consist of approximately 63k trained weights each. While our current experiments are limited to smaller-scale models due to these constraints, the core principles of Transformer-NFN are model-agnostic and specifically designed to scale with increasing model size.
>
> We appreciate the suggestion to include small-scale experiments with larger models. We recognize its importance and will prioritize this direction in future iterations of our work.
>
> ---
>
> Once again, we sincerely thank the reviewer for their feedback. Please let us know if there are any additional concerns or questions from the reviewer regarding our paper.

---

> ### Comment · Reviewer_ynVN · 2024-11-19
>
> Well done, all my concerns have been addressed. I still recommend the authors pay some attention to the larger and more practical scenario about the transformer in their future works. I will keep my score.

---

### Official Review · Reviewer_YffW · 2024-11-03

**Soundness:** 3
**Presentation:** 2
**Contribution:** 3
**Rating:** 6
**Confidence:** 2

**Summary:**

This paper investigates Equivariant Neural Functional Networks for Transformer architectures by developing a specialized model, Transformer-NFN, designed to process the weights, gradients, or sparsity patterns of Transformers as input data. The paper defines a maximal symmetric group for multi-head attention modules and establishes a group action on the weight space of Transformer architectures, creating a systematic framework for NFNs with equivariance properties. Besides, Transformer-NFN, an NFN specifically tailored to Transformer structures, is evaluated on a new dataset containing 125,000 Transformer checkpoints. Experimental results indicate that Transformer-NFN outperforms other models in predicting Transformer model performance.

**Strengths:**

1. The paper presents a novel theoretical framework for designing Neural Functional Networks that respects the inherent symmetries of Transformer architectures, representing an innovative approach previously unexplored in this domain.
2. The release of a dataset with 125,000 Transformer model checkpoints offers a valuable resource for future research on Transformer performance prediction and NFNs.
3. The paper provides a detailed methodological development, including thorough derivations and design principles, establishing the theoretical foundation for NFNs tailored to Transformer architectures, which may inspire future studies in this area.

**Weaknesses:**

1. While the use of equivariant layers in Transformer-NFN is theoretically compelling, empirical results suggest limited practical impact. Specifically, the marginal gains in predictive accuracy on the benchmark dataset raise questions about the method’s utility in real-world applications.
2. The architecture's intricate structure, requiring a specialized understanding of group actions and equivariant layers, could hinder adoption. The dense theoretical foundation, coupled with limited practical resources, may present obstacles to reproducibility and broader implementation in practical settings.

**Questions:**

1. Could the authors clarify the expected computational cost of Transformer-NFN compared to traditional NFNs, considering the added design complexity?
2. Will the paper include code to facilitate reproducibility?

---

> ### Author Response · Authors · 2024-11-18
>
> We appreciate the reviewer’s feedback and have provided the following responses to address the concerns raised about our paper. Below, we summarize the weaknesses and questions highlighted by the reviewer and provide our answers accordingly.
>
> ---
>
> **W1. While the use of equivariant layers in Transformer-NFN is theoretically compelling, empirical results suggest limited practical impact. Specifically, the marginal gains in predictive accuracy on the benchmark dataset raise questions about the method’s utility in real-world applications.**
>
> **Answer.** To further evaluate the performance of our Transformer-NFN, we conducted experiments testing all methods under variations in the maximal symmetric group (refer to **Q2** in General Response). The dataset used was a set of Transformers trained on text classification for AGNews, representing a natural language application. Our model demonstrated consistent performance, while other baselines experienced a significant drop. This underscores our model's ability to process diverse inputs in the weight space, which could be advantageous for larger models trained on extensive datasets in real-world applications.
>
> We recall the details of the experiment from **Q2** in General Response as follows.
>
> *Experiment Setup.* In this experiment, we evaluate the performance of Transformer-NFN on the AGNews-Transformers dataset augmented using the group action $\mathcal{G}_\mathcal{U}$. We perform a 2-fold augmentation for both the train and test sets. The original weights are retained, and additional weights are constructed by applying permutations and scaling transformations to transformer modules. The elements in $M$ and $N$ (as defined in Section 4.3) are uniformly sampled across $[-1, 1]$, $[-10, 10]$, and $[-100, 100]$.
>
> *Results.* The results for all models are presented in Table 1. The findings demonstrate that Transformer-NFN maintains stable Kendall’s $\tau$ across different ranges of scale operators. Notably, as the weights are augmented, the performance of Transformer-NFN improves, whereas the performance of other baseline methods declines significantly. This performance disparity results in a widening gap between Transformer-NFN and the second-best model, increasing from 0.031 to 0.082.
>
> The general decline observed in baseline models highlights their lack of symmetry. In contrast, Transformer-NFN's equivariance to $\mathcal{G}_\mathcal{U}$ ensures stability and even slight performance improvements under augmentation. This underscores the importance of defining the maximal symmetric group, to which Transformer-NFN is equivariant, in overcoming the limitations of baseline methods.
>
> *Table 1: Performance measured by Kendall's $\tau$ of all models on **Augmented AGNews-Transformers** dataset. Uncertainties indicate standard error over 5 runs.*
> |                 |           Original            |                $[-1, 1]$                |               $[-10, 10]$                |               $[-100,100]$               |
> |:--------------- |:-----------------------------:|:-------------------------------:|:-------------------------------:|:-------------------------------:|
> | XGBoost         |       $0.859 \pm 0.001$       |       $0.799 \pm 0.003$       |       $0.800 \pm 0.001$       |       $0.802 \pm 0.003$       |
> | LightGBM        |       $0.835 \pm 0.001$       |       $0.785 \pm 0.003$       |       $0.784 \pm 0.003$       |       $0.786 \pm 0.004$       |
> | Random Forest   |       $0.774 \pm 0.003$       |       $0.714 \pm 0.001$       |       $0.715 \pm 0.002$       |       $0.716 \pm 0.002$       |
> | MLP             | $\underline{0.879 \pm 0.006}$ | $\underline{0.830 \pm 0.002}$ | $\underline{0.833 \pm 0.002}$ | $\underline{0.833 \pm 0.005}$ |
> | STATNN          |       $0.841 \pm 0.002$       |       $0.793 \pm 0.003$       |       $0.791 \pm 0.003$       |        $0.771 \pm 0.013$        |
> | **Transformer-NFN** |  $\mathbf{0.910 \pm 0.001}$   |  $\mathbf{0.912 \pm 0.001}$   |  $\mathbf{0.912 \pm 0.002}$   |  $\mathbf{0.913 \pm 0.001}$   |
> | Gap             |              $0.031$              |               $0.082$               |              $0.079$               |               $0.080$               |

---

> ### Author Response · Authors · 2024-11-18
>
> **W2+Q2. The architecture's intricate structure, requiring a specialized understanding of group actions and equivariant layers, could hinder adoption. The dense theoretical foundation, coupled with limited practical resources, may present obstacles to reproducibility and broader implementation in practical settings. Will the paper include code to facilitate reproducibility?**
>
> **Answer.** We recognize the challenges in understanding the implementation. To address this, our paper already includes an Implementation section in the Appendix H.2 and H.3 with pseudocode for clarity. The network consists of multiple components, represented by elegant einsum operators that are easy to read and adapt. Additionally, the supplementary materials provide code for dataset generation and Transformer-NFN layers, accompanied by detailed documentation to facilitate result reproduction.
>
>
> **Q1. Could the authors clarify the expected computational cost of Transformer-NFN compared to traditional NFNs, considering the added design complexity?**
>
> **Answer.** The computational complexity of the Transformer-NFN is derived from its invariant and equivariant layers as outlined in our pseudocode (Appendix H):
>
> - **Equivariant Layer Complexity:**  $\mathcal{O}(d \cdot e \cdot h \cdot D^2 \cdot \max(D_q, D_k, D_v))$.
> - **Invariant Layer Complexity:**  $\mathcal{O}(d \cdot e \cdot D' \cdot D \cdot \max(D \cdot h, D_A))$.
>
> Where the parameters are:
> - $d, e$: Input and output channels of the Transformer-NFN layer, respectively.
> - $D$: Embedding dimension of the transformer.
> - $D_q, D_k, D_v$: Dimensions for query, key, and value vectors of the transformer.
> - $D_A$: Dimension of the linear projection of the transformer.
> - $h$: Number of attention heads of the transformer.
> - $D'$: Dimension of the output of the Transformer-NFN Invariant Layer.
>
> The Transformer-NFN implementation leverages optimized tensor contraction operations (e.g., $\texttt{einsum}$), enabling efficient and highly parallelizable computations on modern GPUs. This ensures computational practicality while delivering significant performance improvements.
>
> ---
>
> We sincerely thank the reviewer for the valuable feedback. If our responses adequately address all the concerns raised, we kindly hope the reviewer will consider raising the score of our paper.

---

> ### Author Response · Authors · 2024-11-21
> **Any Questions from Reviewer YffW on Our Rebuttal?**
>
> We would like to thank the reviewer again for your thoughtful reviews and valuable feedback.
>
> We would appreciate it if you could let us know if our responses have addressed your concerns and whether you still have any other questions about our rebuttal.
>
> We would be happy to do any follow-up discussion or address any additional comments.

---

> > ### Comment · Reviewer_YffW · 2024-11-21
> >
> > The author fully addressed my concerns. Thank you! I am happy to maintain my positive score and feedback.

---

> > > ### Author Response · Authors · 2024-11-21
> > > **Thanks for your endorsement!**
> > >
> > > Thanks for your response, and we appreciate your endorsement.

---

### Official Review · Reviewer_TUKj · 2024-11-03

**Soundness:** 3
**Presentation:** 2
**Contribution:** 3
**Rating:** 8
**Confidence:** 4

**Summary:**

The paper presents a formulation of NFNs for transformers beyond MLPs and CNNs. The paper provides thorough analysis of how to establish a weight space around multihead attention and a way to formalize NFNs through a parameter sharing strategy. The paper also shows the application on multiple transformer models across a basic vision and language task. The paper is very detailed in it's approach and provides for a good approach to understanding how the weights of transformer blocks are affected by hyperparameters and data

**Strengths:**

The paper provides a very structured approach to a the solution of defining NFNs for transformers and is rigorous in it's approach.Some of the mathematical formulation around weight spaces of a multihead attention and the solution for parameter sharing of the NFN is good.

**Weaknesses:**

The paper is a bit dense to read and understand. Though this can be considered fine given the topic and detailed approach. There could've been a better balance at analyzing the different theorems presented vs covering the mathematical proof.

The experimentation is basic and doesn't necessarily capture the strengths of the method or even transformers. Transformers are known to generalize over large model sizes and large data and it is the biggest weakness of the paper on how this method would hold for larger models. Even if a small scale experiment was included, it would bolster the through mathematical approach much better.

**Questions:**

How does this generalize to models with more parameters and trained on larger datasets

---

> ### Author Response · Authors · 2024-11-18
>
> We appreciate the reviewer’s feedback and have provided the following responses to address the concerns raised about our paper. Below, we summarize the weaknesses and questions highlighted by the reviewer and provide our answers accordingly.
>
> ---
>
> **W1. The paper is a bit dense to read and understand. Though this can be considered fine given the topic and detailed approach. There could've been a better balance at analyzing the different theorems presented vs covering the mathematical proof.**
>
> **Answer.** We appreciate the reviewer's suggestion and are in the process of restructuring the paper to enhance its flow. The revised version will be updated in a few days.
>
> **W2 + Q1. The experimentation is basic and doesn't necessarily capture the strengths of the method or even transformers. Transformers are known to generalize over large model sizes and large data and it is the biggest weakness of the paper on how this method would hold for larger models. Even if a small scale experiment was included, it would bolster the through mathematical approach much better.**
>
> How does this generalize to models with more parameters and trained on larger datasets?
>
> **Answer.** We agree that evaluating Transformer-NFN on larger models and datasets would provide a more comprehensive demonstration of its strengths. However, generating such datasets requires training thousands of large-scale transformers, which exceeds the computational resources available in our academic lab. For context, both our MNIST-Transformers and AGNews-Transformer datasets each consist of approximately 63k trained weights each. While our current experiments are limited to smaller-scale models due to these constraints, the core principles of Transformer-NFN are model-agnostic and specifically designed to scale with increasing model size.
>
> We appreciate the suggestion to include small-scale experiments with larger models. We recognize its importance and will prioritize this direction in future iterations of our work.
>
> ---
>
> Once again, we sincerely thank the reviewer for their feedback. Please let us know if there are any additional concerns or questions from the reviewer regarding our paper.

---

> ### Author Response · Authors · 2024-11-21
> **Any Questions from Reviewer TUKj on Our Rebuttal?**
>
> We would like to thank the reviewer again for your thoughtful reviews and valuable feedback.
>
> We would appreciate it if you could let us know if our responses have addressed your concerns and whether you still have any other questions about our rebuttal.
>
> We would be happy to do any follow-up discussion or address any additional comments.

---

> ### Author Response · Authors · 2024-11-26
>
> Thanks for your response, and we appreciate your endorsement. We acknowledge the importance of applicability to larger models and consider this an intriguing challenge for future research.

---

### Official Review · Reviewer_wnMM · 2024-11-11

**Soundness:** 3
**Presentation:** 3
**Contribution:** 3
**Rating:** 6
**Confidence:** 4

**Summary:**

The paper introduces a novel concept of Transformer Neural Functional Networks (Transformer-NFN), designed as equivariant neural functional networks specifically for transformer architectures.The proposed Transformer-NFN addresses the challenge of applying neural functional networks (NFNs) to transformers by systematically defining weight spaces, symmetry groups, and group actions for multi-head attention and ReLU-MLP components. Experimental results demonstrate the effectiveness of Transformer-NFN in predicting transformer generalization and performance, surpassing traditional models across multiple metrics.

**Strengths:**

- The is the first paper to my knowledge expore neural functional networks (NFN) in the context of transformer, where NFN treat the weights, gradients, or sparsity patterns of a deep neural network (DNN) as input data.
- The proposed Small Transformer Zoo dataset is a valuable resource for benchmarking and studying transformer-based networks, promoting reproducibility and further research.

**Weaknesses:**

- The paper did not discuss the computational overhead associated with equivariant polynomial layers and whether this could affect scalability for larger transformer models.

**Questions:**

- I'm curious how sensitive is the maximal symmetric group of the weights in a MHA module, would affect the peformance of final peformance of NFN.

---

> ### Author Response · Authors · 2024-11-18
>
> We appreciate the reviewer’s feedback and have provided the following responses to address the concerns raised about our paper. Below, we summarize the weaknesses and questions highlighted by the reviewer and provide our answers accordingly.
>
> ---
>
> **W1. The paper did not discuss the computational overhead associated with equivariant polynomial layers and whether this could affect scalability for larger transformer models.**
>
> **Answer.** The computational complexity of the Transformer-NFN is derived from its invariant and equivariant layers as outlined in our pseudocode (Appendix H):
>
> - **Equivariant Layer Complexity:**  $\mathcal{O}(d \cdot e \cdot h \cdot D^2 \cdot \max(D_q, D_k, D_v))$.
> - **Invariant Layer Complexity:**  $\mathcal{O}(d \cdot e \cdot D' \cdot D \cdot \max(D \cdot h, D_A))$.
>
> Where the parameters are:
> - $d, e$: Input and output channels of the Transformer-NFN layer, respectively.
> - $D$: Embedding dimension of the transformer.
> - $D_q, D_k, D_v$: Dimensions for query, key, and value vectors of the transformer.
> - $D_A$: Dimension of the linear projection of the transformer.
> - $h$: Number of attention heads of the transformer.
> - $D'$: Dimension of the output of the Transformer-NFN Invariant Layer.
>
> The Transformer-NFN implementation leverages optimized tensor contraction operations (e.g., $\texttt{einsum}$), enabling efficient and highly parallelizable computations on modern GPUs. This ensures computational practicality while delivering significant performance improvements.

---

> ### Author Response · Authors · 2024-11-18
>
> **Q2. I'm curious how sensitive is the maximal symmetric group of the weights in a MHA module, would affect the peformance of final peformance of NFN.**
>
> **Answer.** We show that Transformer-NFN’s equivariance to the maximal symmetric group action $\mathcal{G}_\mathcal{U}$ is crucial to its performance, by conducting an experiment on augmented dataset. The experiment is presented in **Q2** of General Response as below.
>
>
>
> ***Experiment Setup.*** In this experiment, we evaluate the performance of Transformer-NFN on the AGNews-Transformers dataset augmented using the group action $\mathcal{G}_\mathcal{U}$. We perform a 2-fold augmentation for both the train and test sets. The original weights are retained, and additional weights are constructed by applying permutations and scaling transformations to transformer modules. The elements in $M$ and $N$ (as defined in Section 4.3) are uniformly sampled across $[-1, 1]$, $[-10, 10]$, and $[-100, 100]$.
>
> ***Results.*** The results for all models are presented in Table 1. The findings demonstrate that Transformer-NFN maintains stable Kendall’s $\tau$ across different ranges of scale operators. Notably, as the weights are augmented, the performance of Transformer-NFN improves, whereas the performance of other baseline methods declines significantly. This performance disparity results in a widening gap between Transformer-NFN and the second-best model, increasing from 0.031 to 0.082.
>
> The general decline observed in baseline models highlights their lack of symmetry. In contrast, Transformer-NFN's equivariance to $\mathcal{G}_\mathcal{U}$ ensures stability and even slight performance improvements under augmentation. This underscores the importance of defining the maximal symmetric group, to which Transformer-NFN is equivariant, in overcoming the limitations of baseline methods.
>
> *Table 1: Performance measured by Kendall's $\tau$ of all models on **Augmented AGNews-Transformers** dataset. Uncertainties indicate standard error over 5 runs.*
> |                 |           Original            |                $[-1, 1]$                |               $[-10, 10]$                |               $[-100,100]$               |
> |:--------------- |:-----------------------------:|:-------------------------------:|:-------------------------------:|:-------------------------------:|
> | XGBoost         |       $0.859 \pm 0.001$       |       $0.799 \pm 0.003$       |       $0.800 \pm 0.001$       |       $0.802 \pm 0.003$       |
> | LightGBM        |       $0.835 \pm 0.001$       |       $0.785 \pm 0.003$       |       $0.784 \pm 0.003$       |       $0.786 \pm 0.004$       |
> | Random Forest   |       $0.774 \pm 0.003$       |       $0.714 \pm 0.001$       |       $0.715 \pm 0.002$       |       $0.716 \pm 0.002$       |
> | MLP             | $\underline{0.879 \pm 0.006}$ | $\underline{0.830 \pm 0.002}$ | $\underline{0.833 \pm 0.002}$ | $\underline{0.833 \pm 0.005}$ |
> | STATNN          |       $0.841 \pm 0.002$       |       $0.793 \pm 0.003$       |       $0.791 \pm 0.003$       |        $0.771 \pm 0.013$        |
> | **Transformer-NFN** |  $\mathbf{0.910 \pm 0.001}$   |  $\mathbf{0.912 \pm 0.001}$   |  $\mathbf{0.912 \pm 0.002}$   |  $\mathbf{0.913 \pm 0.001}$   |
> | Gap             |              $0.031$              |               $0.082$               |              $0.079$               |               $0.080$               |
>
> Unlike baseline methods, which lack such symmetry considerations and exhibit significant performance declines under augmentation, Transformer-NFN maintains stability and even demonstrates improved performance. This highlights the importance of leveraging equivariant group actions to achieve robustness and enhance final performance.
>
> This is supported by existing research [1, 2, 3] showing that incorporating symmetric groups into NFN for MLP improves performance compared to those without symmetry consideration.
>
> ---
>
> **Reference.**
>
> [1] Zhou et al., Permutation Equivariant Neural Functionals. NeurIPS 2023.
>
> [2] Kalogeropoulos et al., Scale Equivariant Graph Metanetworks. NeurIPS 2024.
>
> [3] Tran et al., Monomial Matrix Group Equivariant Neural Functional Networks. NeurIPS 2024.
>
> ---
>
> We sincerely thank the reviewer for the valuable feedback. If our responses adequately address all the concerns raised, we kindly hope the reviewer will consider raising the score of our paper.

---

> ### Author Response · Authors · 2024-11-21
> **Any Questions from Reviewer wnMM on Our Rebuttal?**
>
> We would like to thank the reviewer again for your thoughtful reviews and valuable feedback.
>
> We would appreciate it if you could let us know if our responses have addressed your concerns and whether you still have any other questions about our rebuttal.
>
> We would be happy to do any follow-up discussion or address any additional comments.

---

### Author Response · Authors · 2024-11-18
**General Response (1/2)**

Dear AC and reviewers,

Thanks for your thoughtful reviews and valuable comments, which have helped us improve the paper significantly.

We sincerely thank the reviewers for their valuable feedback and constructive suggestions. We are encouraged by the endorsements that:

1. Our Transformer-NFN is the first equivariant neural functional networks for Transformer models (Reviewer wnMM, YffW, ynVN), which may inspire and prove useful for future research (Reviewer YffW, ynVN).
2. The theoretical analysis is detailed (Reviewer ynVN), well-structured, and supported by rigorous mathematical derivations and design principles (Reviewer TUKj, YffW).
3. Our proposed Small Transformer Zoo dataset provides a valuable resource for advancing future research on Transformers (Reviewer wnMM, YffW).

---

Below, we address two common points raised in the reviews:

**Q1. Computational complexity of Transformer-NFN.**

The computational complexity of the Transformer-NFN is derived from its invariant and equivariant layers as outlined in our pseudocode (Appendix H):

- **Equivariant Layer Complexity:**  $\mathcal{O}(d \cdot e \cdot h \cdot D^2 \cdot \max(D_q, D_k, D_v))$
- **Invariant Layer Complexity:**  $\mathcal{O}(d \cdot e \cdot D' \cdot D \cdot \max(D \cdot h, D_A))$

Where the parameters are:
- $d, e$: Input and output channels of the Transformer-NFN layer, respectively.
- $D$: Embedding dimension of the transformer.
- $D_q, D_k, D_v$: Dimensions for query, key, and value vectors of the transformer.
- $D_A$: Dimension of the linear projection of the transformer.
- $h$: Number of attention heads of the transformer.
- $D'$: Dimension of the output of the Transformer-NFN Invariant Layer.

The Transformer-NFN implementation leverages optimized tensor contraction operations (e.g., $\texttt{einsum}$), enabling efficient and highly parallelizable computations on modern GPUs. This ensures computational practicality while delivering significant performance improvements.

---

> ### Author Response · Authors · 2024-11-18
> **General Response (2/2)**
>
> **Q2. Additional experiment to show the strength of Transformer-NFN.**
>
> *Experiment Setup.* In this experiment, we evaluate the performance of Transformer-NFN on the AGNews-Transformers dataset augmented using the group action $\mathcal{G}_\mathcal{U}$. We perform a 2-fold augmentation for both the train and test sets. The original weights are retained, and additional weights are constructed by applying permutations and scaling transformations to transformer modules. The elements in $M$ and $N$ (as defined in Section 4.3) are uniformly sampled across $[-1, 1]$, $[-10, 10]$, and $[-100, 100]$.
>
> *Results.* The results for all models are presented in Table 1. The findings demonstrate that Transformer-NFN maintains stable Kendall’s $\tau$ across different ranges of scale operators. Notably, as the weights are augmented, the performance of Transformer-NFN improves, whereas the performance of other baseline methods declines significantly. This performance disparity results in a widening gap between Transformer-NFN and the second-best model, increasing from 0.031 to 0.082.
>
> The general decline observed in baseline models highlights their lack of symmetry. In contrast, Transformer-NFN's equivariance to $\mathcal{G}_\mathcal{U}$ ensures stability and even slight performance improvements under augmentation. This underscores the importance of defining the maximal symmetric group, to which Transformer-NFN is equivariant, in overcoming the limitations of baseline methods.
>
> *Table 1: Performance measured by Kendall's $\tau$ of all models on **Augmented AGNews-Transformers** dataset. Uncertainties indicate standard error over 5 runs.*
> |                 |           Original            |                $[-1, 1]$                |               $[-10, 10]$                |               $[-100,100]$               |
> |:--------------- |:-----------------------------:|:-------------------------------:|:-------------------------------:|:-------------------------------:|
> | XGBoost         |       $0.859 \pm 0.001$       |       $0.799 \pm 0.003$       |       $0.800 \pm 0.001$       |       $0.802 \pm 0.003$       |
> | LightGBM        |       $0.835 \pm 0.001$       |       $0.785 \pm 0.003$       |       $0.784 \pm 0.003$       |       $0.786 \pm 0.004$       |
> | Random Forest   |       $0.774 \pm 0.003$       |       $0.714 \pm 0.001$       |       $0.715 \pm 0.002$       |       $0.716 \pm 0.002$       |
> | MLP             | $\underline{0.879 \pm 0.006}$ | $\underline{0.830 \pm 0.002}$ | $\underline{0.833 \pm 0.002}$ | $\underline{0.833 \pm 0.005}$ |
> | STATNN          |       $0.841 \pm 0.002$       |       $0.793 \pm 0.003$       |       $0.791 \pm 0.003$       |        $0.771 \pm 0.013$        |
> | **Transformer-NFN** |  $\mathbf{0.910 \pm 0.001}$   |  $\mathbf{0.912 \pm 0.001}$   |  $\mathbf{0.912 \pm 0.002}$   |  $\mathbf{0.913 \pm 0.001}$   |
> | Gap             |              $0.031$              |               $0.082$               |              $0.079$               |               $0.080$               |
>
> ---
>
> We hope that our rebuttal has helped to clear concerns about our work. We are glad to answer any further questions you have on our submission and we would appreciate it if we could get your further feedback at your earliest convenience.

---

### Author Response · Authors · 2024-11-21
**Summary of Revisions**

Incoporating comments and suggestions from reviewers, as well as some further empirical studies we believe informative, we summarize here the main changes in the revised paper:
1. We add an experiment that shows Transformer-NFN’s equivariance to the maximal symmetric group action $\mathcal{G}\_\mathcal{U}$ is crucial for its consistent performance compared to other baselines (Table 9 in Appendix I). The results show that, unlike other non-equivariant baselines, Transformer-NFN maintains stable Kendall’s $\tau$ values across different ranges of scale operators, highlighting its robustness in when the input is transformed by maximal symmetric group action $\mathcal{G}\_\mathcal{U}$.

2. We add the computational complexity of our Equivariant Layer and Invariant Layer in Appendix J of the revised paper. The implementation uses the $\texttt{einsum}$ operation, which enables efficient and highly parallelized computations on modern GPUs. This design ensures computational feasibility while delivering significant performance gains.

---

### Meta-Review · Area_Chair_UUSX · 2024-12-21

**Metareview:**

The paper proposes an algorithm for equivariant transformer architectures of neural fields. Reviewers are unanimously in favor of accepting the paper, and initially had questions about the computational cost, as well as the significance of improvements. Authors answered with additional experiments, showing that the computational cost is reasonable, and improvements are also non-trivial. All in all, this is a good paper.

**Additional Comments On Reviewer Discussion:**

There were no significant comments or changes during the reviewer discussion.

---

### Decision · Program_Chairs · 2025-01-22

Accept (Poster)